



**Effects of surface roughness and light-absorbing impurities on glacier surface albedo, August-one ice cap, Qilian Mountains, China.**

Junfeng LIU[1], Rensheng CHEN[1], Yongjian Ding[1], Chuntan HAN[1], Yong Yang[1], Zhangwen Liu[1], Xiqiang
Wang[1], Shuhai Guo[1], Yaoxuan Song[1], Wenwu QING[2]

1. Qilian Alpine Ecology and Hydrology Research Station, Key Laboratory of Ecohydrology of Inland River Basin, Northwest Institute of Eco-Environment and Resources, Chinese Academy of Sciences, Lanzhou, China.
2. Lanzhou University, Lanzhou, China.

Correspondence to: Rensheng Chen (crs2008@lzb.ac.cn), Yongjian Ding (dyj@lzb.ac.cn)

**Abstract:** Surface albedo is the main influence on surface melt for Qilian mountain glaciers. Fluctuations in surface albedo are due primarily to variations in micro scale surface roughness ($\xi$) and light-absorbing impurities (LAIs) in this region. However, combined $\xi$ and LAIs effects over glacier surface albedo are rarely studied and surface roughness rarely considered in the albedo parameterization methods in this region. The present study was conducted in tandem with an intensive photogrammetric survey of glacier surface roughness, LAIs samples and albedo observations along the main flow-line of August-one ice cap during the 2018 melt season. Automatic photogrammetry of surface roughness and automatic observation of glacier surface albedo was conducted at middle of the ice cap in 2018. Detailed analysis indicates a negative power function and positive linear relationship exist between $\xi$ and albedo for snow and ice surface, respectively. $\xi$ could explain 68% of snow surface albedo and 38% of ice surface albedo variation in melt season. Effective LAIs concentration ($C_\xi$) calculated by consider $\xi$ effect over LAIs deposition account for more than 63% of albedo variation at ice surface. Using either $\xi$ or $C_\xi$ to estimate ice surface albedo would be a great improvement over some current parameterization methods, such as assuming a constant mean ice surface albedo. A finer resolution of above 50mm and above 100mm is recommended for ice and snow $\xi$ calculations, which explain more albedo variation than coarse resolutions below it. With advances in topographic surveys to improve the resolution, extent and availability of topographic datasets and surface roughness, appropriate parameterizations of albedo based on $\xi$ have exciting potential



to be applied over large scale snow cover and glacier.

**Keywords**: Albedo; snow; ice; surface roughness; Light-absorbing impurities; August-one ice cap; Qilian Mountains.

## 1. Introduction

The energy balance at the glacier-atmosphere interface is controlled by meteorological conditions above the glacier and physical properties of the glacier itself, which determine glacier melt process (Hock, 2005).

In general, the melting of glacier is dominated by air temperature and net shortwave radiation in the glacier surface. Shortwave radiation is the main energy input causing snow and ice melt (Gardner and Sharp, 2010; Male and Granger, 1981). According to Hock (2005), on average it accounts for over 70% of the net energy input to glacier surfaces. Recent surface energy budget (SEB) studies undertaken on Qilian mountain glaciers show that in this region, melt season net radiation accounts for 80–92% of glacier

melt in ablation zones (Chen et al., 2007; Sun et al., 2012; Qing et al., 2018; Sun et al., 2018) and over 70% of glacier melt in accumulation zones (Sun et al., 2014; Wu et al., 2016). Hence the rate of glacier melt is largely determined by the surface albedo in melting season. It is of great importance to study albedo if we are to improve the accuracy of our energy- and mass-balance models and meltwater runoff in this region.

Glacier albedo varies much more dramatically than other land covers, within the range of 0.1 for dirty wet ice to 0.9 for fresh dry snow (Warren and Wiscombe, 1980; Bøggild et al., 2010). Its variability is attributable to constantly changing surface characteristics as well as clouds and solar incidence angle (Mellor, 1977; Carroll and Fitch, 1981; Konzelmann and Ohmura, 1995; Cutler and Munro, 1996; Oerlemans and Knap, 1998; Brock et al., 2000; Kylling et al., 2000; Gardner and Sharp, 2010; Zhuravleva

and Kokhanovsky, 2011; Goelles and Bøggild, 2017). Light-absorbing impurities (LAIs) at the glacier surface, consisting of wind-blown dust particles, carbonaceous particles and colored organic matter produced by glacial organisms, can accelerates snow aging process, and reduce glacier albedo greatly in its visible spectrum (Oerlemans, 1993; Brock et al., 2000; Bøggild et al., 2010; Hadley and Kirchstetter, 2012; Cook et al., 2017), which enhance the melting snow and ice (Bond et al., 2013; Flanner et al., 2007;

Oerlemans et al., 2009; Xu et al., 2009; Gabbi et al., 2015). Glaciers in the Tibetan Plateau are greatly



affected by LAIs (Takeuchi and Li, 2008; Jiang et al., 2010; Yang et al., 2011; Wang et al., 2012; Zhang et al., 2013; Liu et al., 2014; Ji, 2016; Li et al., 2018; Qing et al., 2018; Sun et al., 2018). Simulation of the effect of LAIs on the albedo of Tibetan glaciers showed that LAIs had a contribution of 34% to the albedo reduction during the late spring time (Ming et al., 2012). For the Qilian mountain glaciers, where

the measured daily mean albedo decreased to the lowest of 0.13±0.06 due to the effect of LAIs for four glaciers observed during melting season (Jiang et al., 2010; Liu et al., 2014; Qing et al., 2018; Sun et al., 2018).

As the snow melts, insoluble LAIs are retained at the snow surface, so concentrations of LAIs in surface snow increase with snow melt, further reducing snow albedo (Doherty et al., 2013). Snow surface

roughness is a measurement of the variability of surface microtopographic features (Fassnacht et al., 2009a). It is a function of crystal type, deposition conditions, and metamorphism and temperature history (Fassnacht et al., 2009a). Snow surface roughness is an important factor for the scattering of light and thereby related to the surface albedo (Warren, 1982; Leroux and Fily, 1998; Warren et al., 1998; Mishchenko et al., 1999; König et al., 2001; Arnold and Rees, 2003; Zhuravleva and Kokhanovsky, 2011;

Larue et al., 2019). The importance of surface roughness for snow was recognized by Kuhn (1974, 1985), who pioneered the study of its effect on bidirectional reflectance distribution function. The influence of snow roughness on albedo has been advanced by several measurement campaigns (Grenfell et al., 1994; Warren et al., 1998; Zhuravleva and Kokhanovsky, 2011). Several theoretical models have also been developed in order to quantify the snow roughness effect on radiative characteristics (Roujean et al., 1992;

O'Rawe, 1991; Larue et al., 2019). Studies have indicates that inclusion of surface roughness in radiative transfer equation has improved the agreement with measurements for macro scale snow surface features such as suncups, penitents, sastrugi. In contrast, the relationship between snow surface roughness and its albedo has been poorly investigated, and snow surface albedo parameterization methods based on surface roughness are rarely reported.

As the ice melts, the distribution LAIs across ice surfaces is heterogeneous (Hodson et al., 2008; Li et al., 2018), leading to differential absorbing of shortwave radiation at a range of spatial scales. The heterogeneous distribution of LAIs results in the roughening of the ice surface: a process that enhances



turbulent heat exchange across the atmospheric boundary layer-ice interface. Furthermore, there may be a feedback process whereby a roughening surface hiding LAIs from direct sun light and increasing ice

surface albedo. Surface roughness structures developed during melt season such as crevasses, cyroconite holes, can increase ice surface albedo by hiding LAIs from direct sunlight have been widely reported (Lliboutry, 1964; Oerlemans, 1993; Bøggild et al., 2010; Takeuchi et al., 2014; Chandler et al., 2015; Takeuchi et al., 2018). During the past 5 years the August-one ice cap has become darker due to the accumulation of LAIs (Figure1). Larger melt rates may have caused an increase in the melt-out of LAIs

contained in the ice. A study at the August-one ice cap has indicate that spatial and temporal surface roughness is variable during melting season (Liu et al., 2020). This surface roughness could directly affect the concentrations of LAIs over ice surface roughness and further more affect the ice surface albedo. Although there have been extensive research focusing on quantifying the impact of LAIs in ice and snow to understand the relationship between LAIs and albedo reduction (Aoki et al., 2011; Painter et al., 2012;

Ginot et al., 2014; Kaspari et al., 2014), combined surface roughness and LAIs effects over dirty ice surface albedo are rarely investigated.

In this study, we try to investigate the spatial and temporal variability of albedo, micro scale surface roughness, and LAIs, with the objective to better understanding and simulating surface albedo variability over snow and dirty ice surface at the August-one ice cap in Qilian Mountain. We used manual

photogrammetry to precisely measure surface roughness at different altitudes and times. Combine with sampled LAIs and albedo observations over different altitudes. We try to investigate the effects of micro scale surface roughness and LAIs on glacier snow and bare ice albedo along the main flow-line of August-one ice cap during the 2018 melt season. Automatic photogrammetry of surface roughness and automatic observation of glacier surface albedo was conducted at middle of the ice cap in melting season and

accumulation season of 2018. During this time, snow cover gave place to ice and then returned to snow. The effect of micro scale surface roughness on glacier snow and ice albedo was also analyzed on the daily scale in melting and accumulation season separately. Snow surface albedo parameterization methods are established based on either surface roughness or both LAIs and surface roughness in melting season. Bare ice surface albedo parameterization methods are also established based on either surface roughness or



effective LAIs concentration.

## 2. Data and methods

### 2.1 Study area


The August-one ice cap is located in the middle of Qilian Mountains on the northeastern edge of the
Tibetan Plateau (Fig 1a, 1b). The glacier is a flat-topped ice cap that is approximately 2.3 km long and
2.4 km$^2$ in area. It ranges in elevation from 4550 to 4820m a.s.l. (Guo et al., 2015). The ice cap experiences
westerly winds, and is characterized by a typical continental climate with dominant precipitation from
May through September. Summer is short, and mean monthly air temperature is $> 0$ $^o$C from June through
August at 4800 m a.s.l. Moreover, the Badain Jaran and Tengger deserts are located to the north; the
Qaidam Basin lies to the south of the Qilian Mountains. Prevailing winds send enormous amounts of dust
particles to the glacial surface in this region (Kreutz, 2007; Dong et al., 2014a; Dong et al., 2014b). Annual
dust deposition in the western mountain areas spans a range of 143.8–207.6 µg cm$^{-2}$yr$^{-1}$ (Dong et al.,
2014b). Influenced by climate warming and continuous accumulation of LAIs, the whole August-one ice
cap surface darkens during melting season and no accumulation area is observed for the last 6 years.
**Figure 1. Locations of the automatic and manual photogrammetry plots, LAIs samples and shortwave observation platforms. Background dark ice surface image of the August-one ice cap was captured by unmanned aerial vehicle on July 29 of 2016.**

[Figure 1]

## 2.2 Data collection

Photogrammetry has been widely used to measure micro scale surface topography (Fassnacht et al., 2009a; Fassnacht et al., 2009b; Manninen et al., 2012; Irvine-Fynn et al., 2014; Smith et al., 2016; Miles et al.,



2017; Quincey et al., 2017; Fitzpatrick et al., 2019). In this study, manual close-range photogrammetry was performed using a portable aluminum square frame delineating a 1.1m×1.1m plot. Researchers made high-definition photographs of the glacier surface within the frame. Photos were taken at ~1.6 m distances, covering an area of ~1.75 m². Seven to fifteen such pairs were taken at each survey site by surrounding

the target area of snow or ice surface. The camera used was an EOS 6D 50mm, with fixed focal lens and an image size of 5472×3648 pixels. The f-stop was fixed at f 22 with an exposure time from 1/25 to 1/125 s. This was done for several different locations that ranged from the glacier terminals to top of the ice cap. These transits were performed on 12 and 25 July, and 28 August, during the 2018 melt season.

These photographs allowed researchers to calculate micro scale topographic over different altitude.

Physical surface roughness was estimated based on these micro scale topography data. At the same time, measurements of up and downward shortwave radiation were made by putting Kipp & Zonen CMP11 1 m above the aluminum plot. Glacier surface albedo was calculated from measurements of up and downward shortwave radiation. Surface LAIs were sampled in a smaller 20cm×20cm plot inside the aluminum frame (Figure 1s), after manual photogrammetry and albedo measurements had been taken.

The samples were collected using a stainless steel spoon from the upper 0 - 9 cm surface at each site and stored in a transparent plastic zip lock bag. All the collected samples were subsequently transported in the field laboratory and filtered. The filters were dried in the oven at 50 ℃ for 24 h to eliminate the water vapor. The mass of insoluble LAIs in the snow or ice samples were calculated by weighing the filter using a microbalance (accuracy: 0.01g).

A station for automatic close-range photogrammetry was set up on a relatively flat surface in the middle of the ice cap (4700m, 98° 53.4′ E, 39° 1.1′ N. See Figure 1), and began operation on 12 July 2018. A 1.5m×2m plot was marked off by a wooden frame and served as a geo-reference control field. A Canon EOS 1300D camera moved along a 1.5 m long slider rail and took seven pictures of the control field at different locations of the slider rail over a period of 10 minutes. The photography was repeated at three-

hour intervals from 9:00 AM to 18:00 AM, UTC+7 time. Photos were later merged to produce a 1mm×1mm resolution surface topography. A Kipp & Zonen CNR4 was set up to record incoming and reflected solar radiation of the control field surface. Samples were taken every 15 seconds; 10-minute



means were stored on a data logger (CR800, Campbell, USA) located at a height of 1.5m.These measurements were taken over three month period from July 12 to October 15 of 2018.

For manual photogrammetry, we put the aluminum frame horizontally over the ice surface, the plot is detrended by setting the control points at z axis of the same values. For automatic photogrammetry, the control field of wooden frame was also laid horizontally over the ice surface that lowered as the ice melted and maintained a horizontal position between the control field and ice surface. Both manual and automatic photographs were imported into the Agisoft Photoscan Professional 1.4.0 application, which produced

plot-surface point clouds and generated a detrended micro scale DEM of 1 mm resolution at plot scale. More detailed information about the current research program's use of manual and automatic photogrammetry can be found in Liu et al. (2020).

## 2.3 Calculation of surface roughness and effective LAIs concentrations



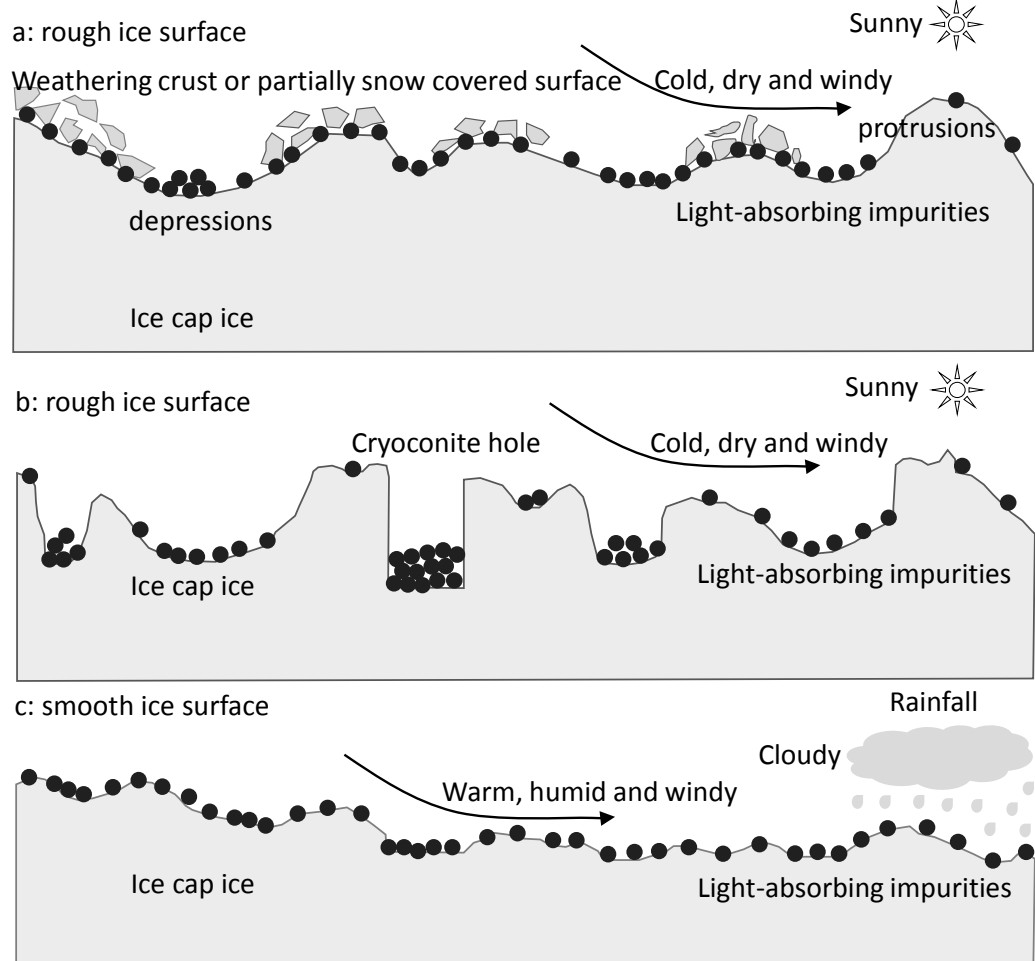

**Figure 2 Schematic images of cross section of ice surface LAIs distribution over rough and smooth ice surface. Panel (a) shows LAIs hidden beneath weathering crust or snow cover, depressions have more LAIs than protrusions; Panel (b) shows LAIs hidden in cryoconite holes and protrusions have less concentration of LAIs. Panel (c) shows LAIs distributed over relatively smooth ice surface without hidden and protrusions effects.**

[Figure 2]

There are dozens of surface roughness describing parameters (Dong et al., 1992; Dong et al., 1994; Smith, 2014). Metrics such as the random roughness (RR) or root mean square height deviation ($\sigma$), the sum of the absolute slopes ($\Sigma$S), the microrelief index (MI), rugosity (Brasington et al., 2012) and the peak




frequency (the number of elevation peaks per unit transect length) were commonly used. Ice surface

albedo is very sensitive to the LAIs concentration over ice surface. Micro scale features such as cryoconite

holes, pits, peaks, weathering crust and macro scale surface fluctuation directly affect LAIs concentrations

through hidden effect, protrusion and depression effects which induced uneven distribution of LAIs over

rough ice surface, or more surface area effect which dilute LAIs concentration over rough ice (Figure 2).

It means surface roughness could indirectly affect albedo by direct affect LAIs concentration. In this study,

a 3-D measure of the topographic roughness defined as:

$$\xi = h^* \frac{A_s}{A_p} \quad (2)$$

Where $\xi$ is the surface roughness (cm), $h^*$ represents the average vertical extent of microtopographic

variations (m), $A_s$ is the detrended 3-D surface area (cm$^2$), and $A_p$ is the planimetric or 2-D area (cm$^2$).

The definition of $\xi$ in this study is adopted from Lettau (1969) developed aerodynamic surface roughness

except the silhouette area facing upwind changed to 3-D surface area in equation (2). The $\xi$ is particularly

sensitive to local variability in surface slope and aspect and the presence of topographic singularities. We

adopted equation (1) to calculate surface roughness ($\xi$) by using the automatic and manual

photogrammetry acquired 1×1m plot detrended digital elevation model (DEM) data at 1mm resolution.

Based on the plot scale $\xi$ and sampled LAIs, the LAIs concentration ($C_{LAIs}$) over smooth ($\xi=0$ cm) snow

and ice surface was estimated based on the following equation:

$$C_{LAIs} = \frac{M_s}{V} \qquad (3)$$

Where $C_{LAIs}$ is the LAIs concentration over smooth surface (g/cm$^3$), $M_s$ is weight of dried LAIs (g), V is

volume of collect snow or ice sample at the August-one ice cap (cm$^3$).

For rough ice surface means more surface area and less concentration of LAIs over ice surface, also rough

surface structures such as depressions and cryoconite holes hide LAIs from direct sunlight and reduced

LAIs concentration. In this case, an effective LAIs concentration ($C_\xi$) by consider surface roughness is

defined as:

$$C_\xi = \frac{M_s}{V(1+\xi)} \qquad (4)$$

In equation (4), we assumed effective LAIs concentration ($C_\xi$) can be adjusted by consider the surface





roughness effects. Where $C_\xi$ is the effective LAIs concentration over rough surface (g/cm³). As for very smooth ice surface or snow surface, $\xi$ is close to 0, and the limit of $C_\xi$ is $C_{LAIs}$. For snow surface, insoluble LAIs are retained at the snow surface, so concentrations of LAIs in surface snow increase with snow melt, add with snow surface $\xi$ is small and no rough structures formed in melting season. In this case we expect equation (3) is more appropriate to calculate snow surface LAIs concentrations in melting season. Equation (4) is appropriate to calculate ice surface LAIs concentrations especially for rough ice surface.

## 3. Spatial and temporal variability of albedo, surface roughness and LAIs concentrations

Solar zenith angle has a strong effect on albedo (Gardner and Sharp, 2010). For that reason, we used only half-hour albedo data taken when zenith angles were less than 60°, in an effort to avoid solar zenith angle effects. Manual observations of albedo are also carried out from 11:00 to 17:00 o'clock to avoid zenith angle effect.

### 3.1 Spatial variability of surface roughness, LAIs and albedo

On July 12, field investigation indicated that at the ice cap terminals of 4600m, the surface featured patchy snow cover with many cryoconite holes. As altitude increased, there was less bare ice, more snow, and fewer cryoconite holes. From 4700m to the top of glacier, there was near-complete snow coverage. Manual measurements indicated that surface albedo increased from 0.31±0.06 at the glacier terminals to 0.61±0.04 at top of the ice cap (Figure 3a). Surface roughness decreased from 5.49±1.5cm to 0.5±0.6cm as altitude increased (Figure 3b). LAIs decreased from 0.02±0.013mg/g for patchy snow cover to around 0.0018±0.001g/cm³ for snow cover as altitude increased (Figure 3c). On August 3, the August-one ice cap surface was basically all ice, except at the top of the ice cap, where there was still some patchy snow cover. Measurements indicated that albedo increased from 0.14±0.03 at the terminals to 0.21±0.05 at top of the ice cap (Fig.3d). Glacier surface albedo had decreased greatly when compared with July 12 observations. Surface roughness fluctuated between 1.4±0.4cm to 3.3±1.1cm; An increasing trend of surface roughness could be detected as altitude increased (Figure 3e). LAIs decreased from 0.05±0.01g/





cm$^3$ to around 0.01±0.005g/ cm$^3$ as altitude increased (Figure 3f). On August 28, the ice cap surface was all bare ice. Albedo showed an increasing trend with altitude and fluctuated between 0.14±0.04 at the middle part to 0.19±0.05 at top of the ice cap (Figure 3g). Ice surface roughness fluctuated between 1.50±0.5 cm at middle of the ice cap to 1.45±0.95cm at top of the ice cap, and it did not significantly increase as altitude increased (Figure 3h). LAIs decreased from 0.04±0.03g/ cm$^3$ at middle part to 0.003±0.002g/ cm$^3$ at higher elevations (Figure 3i).

Manual observations indicated that from July to August, surface snow cover gave place to ice. On the snow-covered glacier surface, surface roughness and LAIs decreased as altitude increased. Albedo, conversely, increased as altitude increased. As the melt season progressed, the transient snowline retreat up-glacier as a zone of patchy snow cover. There was a much higher concentration of LAIs on the uncovered ice surface than snow surface. As a consequence, albedo tended to be low on the ice surface and higher on snow-covered surfaces. As the snow-covered surfaces retreated up the glacier, albedo increased with altitude. After the ice cap was fully bare ice, the glacier showed minimal albedo over the whole surface. The surface roughness of the ice was quite variable, but it also did not increase with altitude. *LAI concentrations* were highly variable and increased from 0.04±0.03g/ cm$^3$ at the terminus to 0.003±0.002g/ cm$^3$ at top of the ice cap.



**Figure 3. Albedo, surface roughness and LAIs concentration vs. altitude, (a-c) As observed on 12 July, (d-f) As observed on 3 August, (g-i) As observed on 28 August.**

[Figure 3]



### 3.2 Temporal variability of albedo and surface roughness

The automatic measurement setup in the middle of the ice cap, which recorded daily surface roughness and albedo, operated successfully for the melting season from July 12 to September 15, and one month of accumulation period from middle of September to October of 17 (Figure 4a). Photos captured by automatic Canon 1300D showed that from July 12 to July 20 the glacier was snow-covered (Figure 4b). Intermittent snowfall decreased surface roughness and increased albedo. After the snow had stopped

falling, surface albedo tended to decrease and surface roughness to increase (Figure 4). When snow began to melt, glacier surface roughness kept increasing and albedo kept decreasing. From July 21 to July 24, snow cover became patchy and surface roughness increased from 1.1cm to 2.1cm (Figure 4a and 4c). During this period, surface albedo decreased sharply from 0.72 to 0.30. Micro-scale structure of cryoconite holes formed during this period (Figure 4c). From July 25 to September 3, glacier surface was

mostly bare ice (Figure 4d and 4e). Surface albedo fluctuated between 0.11±0.01 for smooth ice surface and 0.68±0.02 for intermittent snowfall period, and surface roughness fluctuated between 0.8 cm and 1.6 cm. There were LAIs concentrated on the ice surface (Figure 4d and 4e). Intermittent snowfalls covered the LAIs and reduced surface roughness during this period. At such times surface albedo increased sharply above 0.6. When the snow melted, leaving patchy snow cover, surface roughness increased and albedo

decreased quickly to around 0.3±0.05 within two day. As soon as bare ice appeared, ice surface albedo decreased to around 0.14±0.03. At the end of melting season, from September 4 to 15, ice surface roughness fluctuated and increased to 2.63 cm on September 13. Ice surface micro scale structures of cryoconite holes hidden LAIs from direct sun light. Refreezing process also formed thin ice layer and covered LAIs (Figure 4f). Ice surface albedo following an increasing trend during this period and

increased to 0.55 on September 13. On the following two days, snowfall events increased albedo sharply from 0.55±0.02 to 0.83±0.06 (Figure 4g).

After September 15, the snow covered surface basically entered to the accumulation period at the automatic photogrammetry site. Snow surface albedo fluctuated from 0.61 to 0.90. Snow surface roughness fluctuated between 0.30 cm and 1.67 cm. Based on the snow surface patterns captured by

Canon 1300D camera and meteorological data, we could tell surface roughness variation mainly induced





by constant blowing snow events and intermittent snowfall effects during this period (Figure 4h and 4i).

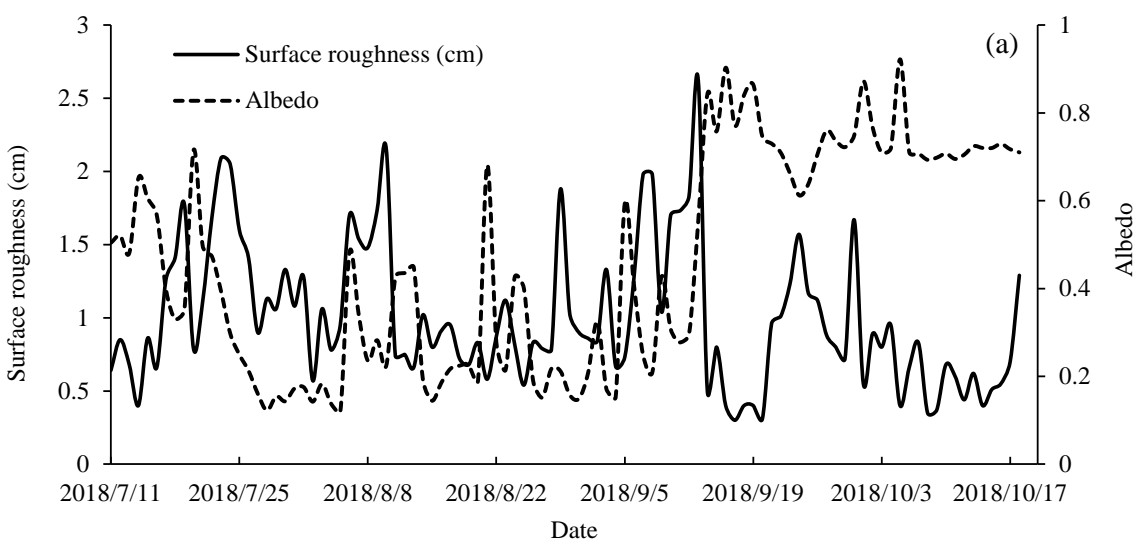

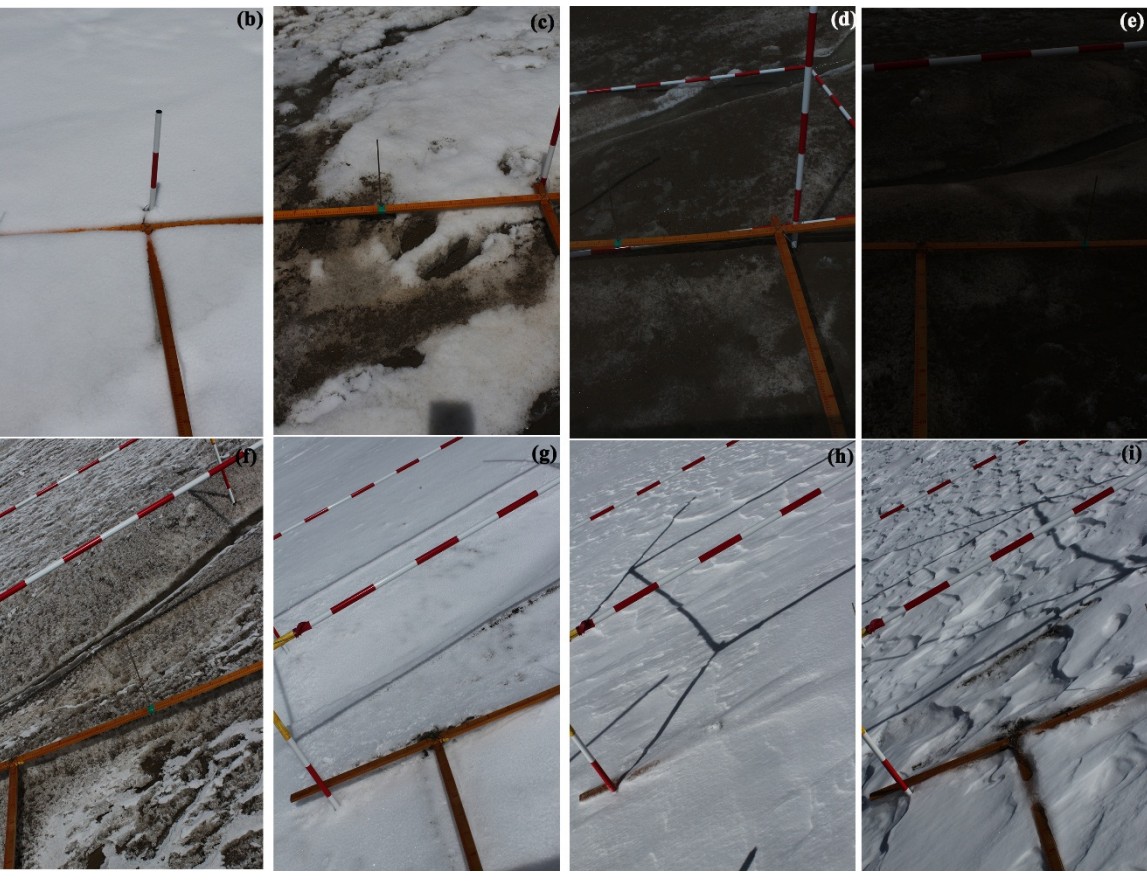



**Figure 4(a) Variations of glacier albedo and surface roughness over time at the automatic photogrammetric surface roughness observation site, photo (b) showed snow covered surface on July 13 at automatic photogrammetry site, photo (c) showed partially snow covered surface on July 23 with cryoconite holes, (d) and (e) showed smooth ice surface on August 1 and August 30, (f) showed rough ice surface on September 13, (g) showed smooth snow surface on September 16, (h) and (i) showed blowing snow events induced the snow surface with sastrugi on September 19 and 25 in 2018.**

[Figure 3]

## 4. Albedo variability, surface roughness and LAIs

### 4.1 Snow surface in melting season

1 m$^2$ plot snow surface albedo and corresponding surface roughness calculated at 1mm resolution are analyzed based on the manual and automatic observations at the August one ice cap. Significant negative power function are established between snow surface roughness and albedo for manual and automatic observation, respectively (Figure 2s). The correlation coefficient reached to 0.77 (Figure 2s, a) for manual and 0.88 (Figure 2s, b) for automatic one. The combined manual and automatic scatter diagrams of Figure 5a display a significant negative power function between snow surface roughness and snow albedo (Figure 5a, r=0.82).

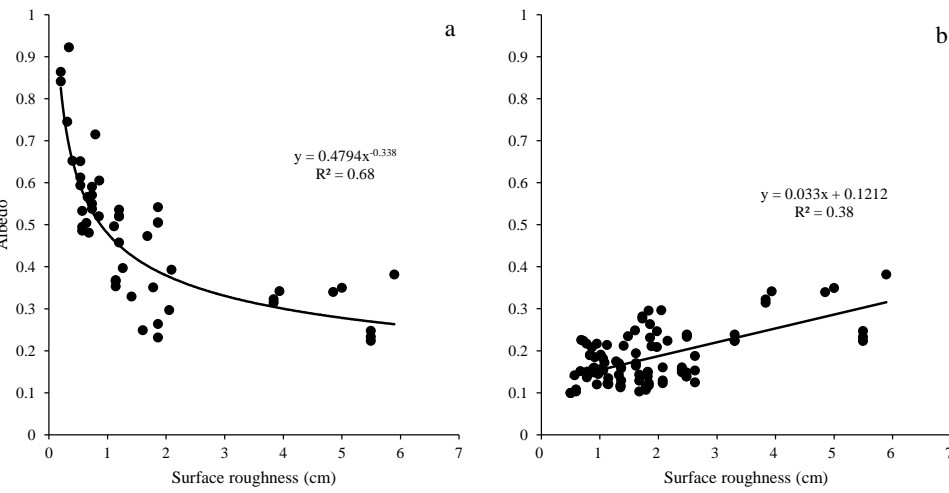





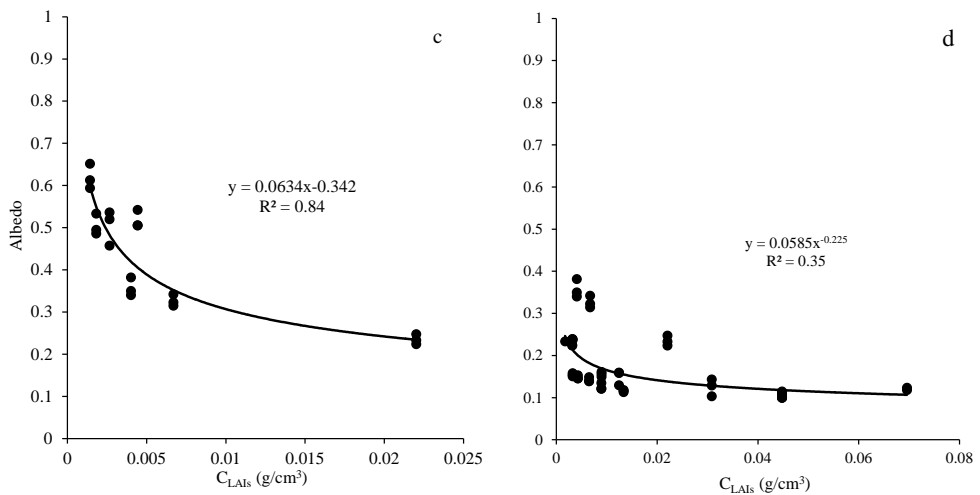

Figure 5(a) Combined manual and automatic measurements of snow surface roughness vs. albedo in 2018 melt season, (b) Combined manual and automatic measurements of ice surface roughness vs. albedo in 2018 melt season, (c) manual measurements of snow surface $C_{LAIs}$ vs. albedo in 2018 melt season, (d) manual measurements of ice surface $C_{LAIs}$ vs. albedo in 2018 melt season

[Figure 5]

## 4.2 Ice surface in melting season

Manual measurements of ice surface albedo and surface roughness at $1m^2$ plot scale with 1mm resolution displayed a significant positive linear relationship (Figure 3s, a, r=0.80). Automatic measurements of ice surface albedo and surface roughness displayed a significant positive linear relationship (Figure 3s, b, r=0.64). Combined manual and automatic measurements of ice surface albedo and surface roughness also displayed a significant positive linear relationship (Figure 5b, r=0.62). Based on Figure 5a and 5b, we could find a negative power function between snow and patchy snow surface roughness and albedo changed to a positive linear relationship between patchy ice and bare ice surface roughness and albedo. A tipping point of negative to positive relationship between surface roughness and albedo appeared when patchy snow and patchy ice cover appeared (Figure 4s). During this period, the large quantity of LAIs hidden beneath snow surface reappeared at depressions or cryoconite holes (Figure 4c). A rough patchy snow surface means a minimum of snow covered surface albedo and maximum of ice surface albedo.

Scatter plots of Figure 5c and 5d shows relationship between $C_{LAIs}$ and albedo without consider surface roughness effect over LAIs concentrations. For snow surface, the LAIs concentration is lower than patchy





snow surface or bare ice, and a significant negative power function relationship established between snow LAIs and albedo (Figure 4c, r=0.91). Scatter diagrams of Figure 4d display a significant power function between manual sampled ice $C_{LAIs}$ and observed ice surface albedo (Figure 4d, r=0.59).

Scatter diagrams of Figure 6a, and 6b shows the effective LAIs concentration of $C_\xi$ and albedo

relationships by considers roughness effect over snow and ice surface LAIs concentration. $C_\xi$ is estimated based on equation (4). Scatter diagrams of Figure 6a display a significant power function between manual sampled snow $C_\xi$ and observed surface albedo (Figure 6a, r=0.61). The correlation coefficient decreased significantly compared with Figure 5c (r=0.91). It means considering surface roughness effect over LAIs concentration by equation (4) do not improve the relationship between LAIs and albedo. It indicate that

the snow surface concentration not affect by surface roughness. Equation (3) is more appropriate to calculate LAIs concentration than equation (4) over snow surface.

Scatter diagrams of Figure 6b display a significant power function between the effective LAIs concentration of $C_\xi$ and observed ice surface albedo (Figure 6b, r=0.79). The correlation coefficient of ice surface LAIs concentration and surface albedo has greatly improved by consider surface roughness effect

over not consider it in Figure 5d (r=0.59), and coefficient of determination has increases from 0.35 to 0.63. It means equation (4) estimate $C_\xi$ could explain more ice surface albedo than equation (3) calculate $C_{LAIs}$.



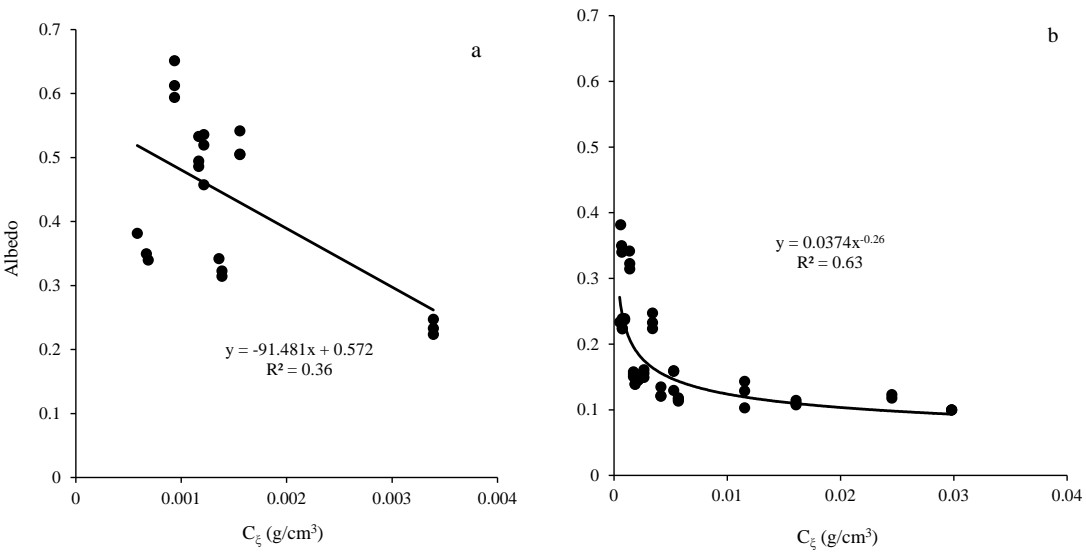

345       Figure 6(a) Snow surface $C_\xi$ vs. albedo in 2018 melt season based on manual observations, (b) Ice surface $C_\xi$ vs. albedo in 2018 melt season based on manual observations. $C_\xi$ is effective LAIs concentration adjusted by considering surface roughness and estimated by equations (4).

[Figure 6]

**4.3 Snow surface in accumulation period**

Glacier surface is basically all covered with snow and LAIs concentration in fresh snow covered ice cap is very low in September and October. Because lack of LAIs samples and the effect of LAIs is not presented here. Scatter diagrams of snow surface roughness and albedo shows negative power function (Figure 7, r=0.49). In accumulation season, except snow particles metamorphism process, constant

blowing snow and intermittent snowfall was the main reasons which induced surface roughness fluctuation (Figure 4a, 4h and 4i). In this case, the correlation coefficient in accumulation period is much lower than in melting season.



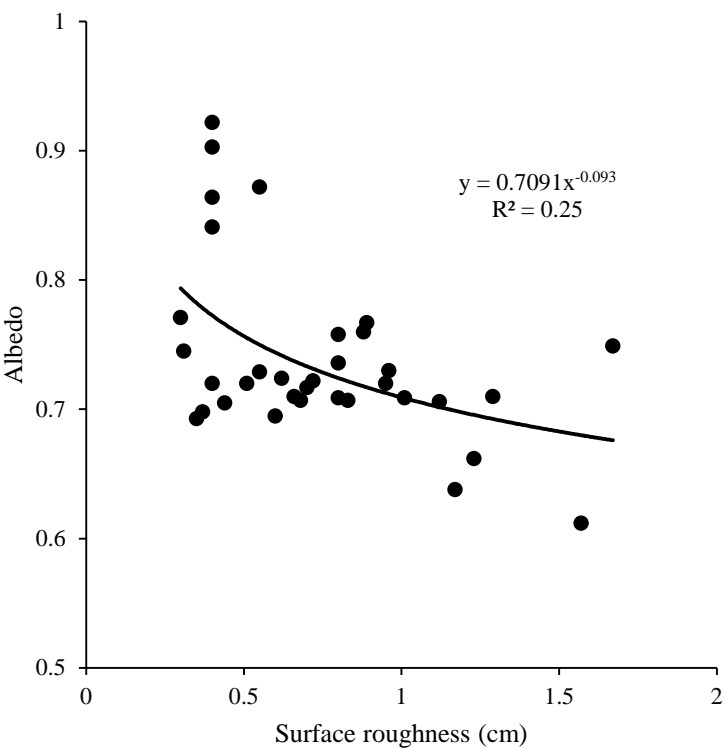

Figure 7 Automatic measurements of snow surface roughness vs. albedo in accumulation period from September 16 to
October 17 in 2018

[Figure 7]

## 7. Discussion

### 7.1 Snow surface albedo variability explanation

Snow surface roughness is a measure of the variability of surface microtopographic features or roughness elements at different scales (Fassnacht et al., 2009b). Fine scale features exist at the millimeter to centimeter size due to variation in grain structure (Munro, 1989; Smeets et al., 1999). At the August-one ice cap, temperatures are above freezing during the day and snow is subject to rapid metamorphism and producing larger, typically rounded grains and clusters. Melt water can refreeze at the surface forming crusts when air temperatures decrease at night. The melt and refreeze pattern are hardened crusts made up of aggregated rounded grains and clusters of grains that have frozen together. The snow surface roughness follows an increasing trend due to snow grain particle metamorphism and melting. Rapid grain





growth actually corresponding with increasing of surface roughness as observed in Figure 4 from July 12 to July 20. There is often a concurrent rise in albedo due to blanket of old snow by fresh snowfall. New

fresh snow surface have smaller surface roughness and corresponding smaller grains that increase the albedo (Figure 4). The surface roughness fluctuation synchronously happened with grain size variation. Although we do not have tandem surface roughness and grain size observations, the evolution of surface roughness calculated at 1mm resolution in snow covered period should quite similar with fluctuation trend of grain size evolution.

The grain size is one of the most critical factors affect snow albedo. As grain size increases, scattering within the snowpack decreases and the absorbing path length within grains increase, thus reducing the broadband albedo (Wiscombe and Warren, 1980; Dozier et al., 1981; Warren, 1982; Flanner and Zender, 2006). Linear or exponential relationships have been established between snow grain size and snow albedo (Brock et al., 2000; Adolph et al., 2016; Skiles and Painter, 2017 ), which is very similar with

scatter plot of surface roughness and snow albedo provided in this study in Figure 5a and Figure 7.  It means surface roughness as substitute of grain size is quite suitable for snow surface albedo explanation and parameterization at millimeter scale.

LAIs is another most critical factors affects snow albedo (e.g. Adolph et al., 2016; Skiles and Painter, 2017). Albedo is most strongly reduced by the presence of LAIs in the visible part (Warren and Wiscombe,

1980), and this effect is enhanced as snow grain size increases (Flanner et al., 2007; Hadley and Kirchstetter, 2012). Higher LAIs content can also impact near infrared range albedo (Hadley and Kirschstetter, 2012). As discussed in the preceding sections, both increasing surface roughness and LAIs concentration tended to decrease albedo. The hidden effect or dilute effect of surface roughness over LAIs concentration is not discerned by application of equation (4) to snow surface. As other studies already

proved that LAIs retained at the snow surface (Doherty et al., 2013), and scavenging by melting process is limited (Skiles and Painter, 2017). We calculate a coefficient of determination value of 0.88 by using a multiple nonlinear regression that includes the surface roughness and $C_{LAIs}$. This indicates that over 85% of the variability in snow albedo can be accounted for using surface roughness and CLAIs. The resulting equation is as follows:



$$\alpha = 0.234 C_{LAIs}^{-0.1415} - 0.02098 \xi^{1.226} \quad (5)$$

The statistical relationship and significance between snow surface roughness and albedo in melting season are also different with accumulation period. During accumulation season, fine scale features develop due to ice grain metamorphism. In accumulation period, large scale features develop due to constant blowing snow events, and snow surface roughness controlled by these macro surface features. In melting season, snow surface roughness increasing due to rapidly melting snow and grain metamorphism, and snow surface features with small scale feature. We expect that different mechanism of surface roughness evolution over melting season and accumulation season are the main reasons of different statistical relationship. There is a general understanding of the physics that control the albedo of snow in melting season and accumulation period, but different factors dominate changes in albedo at various locations (Doherty et al., 2013; Skiles and Painter, 2017). Using extensive measurements in the August-one ice cap, we assess the reducing of snow albedo due to both micro scale surface roughness and LAIs.

**7.2 Ice surface albedo variability explanation**

It has been shown that macro-scale surface roughness greatly reduces surface albedo (Warren et al., 1998; Zhuravleva and Kokhanovsky, 2011; Lhermitte et al., 2014;Larue et al., 2019). When surfaces are rough rather than flat, more incident radiation is absorbed by the slope facing the sun than by the slope facing away from it (Gardner and Sharp, 2010;Cathles et al., 2011). Albedo is decreased on sun-facing slopes. This effect has been shown to hold for sastrugis, crevasses, channels, and penitents (Leroux and Fily, 1998; Warren et al., 1998; Kuchiki et al., 2011; Lhermitte et al., 2014; Rippin et al., 2015; Carroll and Fitch, 1981). Small-scale surface roughness and the resulting variation in the local incidence angle of solar beams also reduce glacier surface albedo and enhance solar absorption (König et al., 2001; Arnold and Rees, 2003; Rees and Arnold, 2006; Cathles et al., 2011).

Contrarily, in this study a significant positive relationship rather than a negative relationship was established over ice surface based on manual and automatic measurements. We expect it is related with abundant LAIs over ice surface at the August one ice cap. Firstly, a rough ice surface means more protection of LAIs from sunlight and smooth ice surface means more expose of LAIs under direct sunlight. This hidden effect such as cryoconite holes could increase surface albedo have been widely reported (e.g.,





Bøggild et al., 2010). Manual observations on August 3 and 28 indicated that micro-scale structures of ice surface cryoconite holes protect LAIs from direct radiation (Figure 4c and 4f, Figure 8d). Secondly, a rough ice surface mean more surface area than smooth ice surface, which could decreased concentration

of LAIs over rough ice surface than smooth ice surface (Figure 8). Additionally, filed investigation over mainly flow line across the August one ice cap find protrusions on the ice surface visually have no or less LAIs concentrations over depressions or flat part of ice at plot scale, which induced higher albedo at these protrusions than depressions even without hidden effect (Figure 8e). Figure 8 displays photos captured during field LAIs measurements which displays LAIs distribution characteristics over snow surface

(Figure 8b), smooth dark bare ice (Figure 8c), cryoconite holes (Figure 8d), protrusions (Figure 8e), and combined hidden and protrusions affect ice surface (Figure 8f). Obviously, cryoconite holes or protrusions all have larger ice surface area and surface roughness over smooth ice, and decrease LAIs effect over absorption of shortwave radiation. A smooth ice surface roughness structures maximized LAIs effect over absorption shortwave radiation and induced the lowest albedo of 0.1 (Figure 8c). The positive linear

relationship between ice surface roughness and albedo in Figure 5b reflect the surface roughness effect over LAIs concentration. Rough surface means lower concentration of LAIs and high albedo over smooth ice surface with heavy loading of LAIs and low ice surface albedo. We expect the ice surface roughness and albedo could be a negative power function similar with snow surface provided if ice surface LAIs concentrations are very low.

As Brock et al. (2000) suggested, to calculate albedo variation accurately in numerical models, parameterizations must be as physically based as possible. Most of the established and widely used snow surface albedo parameterization either based on snowpack age, snow depth, snow density, air temperature (Amaral et al., 2017). The performances of the establishes albedo methods either based on surface roughness, LAIs or effective LAIs concentration shows a great improvement over the assumption of a

constant mean ice albedo or surrogate variables, such as air temperature, accumulated melt and elevation. We expect the new parameterization methods provides in this study are more physically based than some of the studies presented. This induces a great improvement especially for ice surface.





Figure 8(a) Sensitivity of surface roughness under different resolution. (b) is corresponding smooth snow surface with albedo of 0.52±0.02, (c) is corresponding smooth dark and rich in LAIs bare ice with albedo of 0.1±0.01, (d) is corresponding cryoconite holes hidden effect with albedo of 0.36±0.02, (e) is corresponding protrusions effect with albedo of 0.18±0.02, (f) is corresponding protrusions and hidden effects with albedo of 0.23±0.01. (b) and (d) photographed on July 12 2018, (c) and (e) photographed on August 3 2018, and (f) photographed on August 28 2018.




[Figure 8]

## 7.3 Glacier surface albedo parameterization at whole-glacier scales based on surface roughness

For plot scale, estimation of albedo needs fine resolution topography data. Since surface roughness is dependent and sensitive to topography data resolution (Figure 8a), so which resolution is appropriate for snow and ice surface albedo estimation? Based on the manual plot topographic data, Figure 9 shows

correlation coefficients between surface roughness and albedo under different resolutions for 1 m plot scale snow and ice surface. For snow covered surface, the coefficient increases quickly from -0.67 to -0.74 when 1m plot resolution increases from 333.3 mm to 200.0 mm, after then the coefficient stabilized around -0.72±0.04 with increasing of resolution. For ice surface, the correlation coefficient between surface roughness and albedo increases from 0.17 to 0.84 with increasing of plot resolution from 333.3

mm to 0.4 mm. Because the snow surface is flat and smooth, the calculated snow surface roughness is not sensitive to the changes of resolution (Figure 8a). Consequently, the correlation coefficient between albedo and snow surface roughness is not sensitive either. For ice surface, the calculated surface roughness is very sensitive to resolution especially for those rough ice surface such as cryoconite holes and protrusions (Figure 8a). A coarse resolution reduces surface roughness differences of those different ice

surface features (Figure 8a). Consequently, the correlation coefficient reduces as resolution reduces. Figure 8 shows the correlation coefficient increases much slower under finer resolution (<50mm). For albedo studies at plot scale or over whole glacier scale, a resolution of above 50 mm resolution is recommended for topography and surface roughness data during melt season. For snow cover, above 100 mm resolution is recommend for surface roughness calculation in melt season.

For whole glacier scale, recent developments in Terrestrial Laser Scanning, Structure-from-Motion, and Multi-view Stereo may also be able to provide catchment-scale high-resolution topography and surface roughness data (Passalacqua et al., 2015; Smith and Vericat, 2015; Rippin et al., 2015). Such new techniques could provide detailed topographic and surface roughness data. Smith et al. (2016) have suggested a way to extrapolate glacier-scale roughness from plot-scale surface roughness measurements

and glacier-scale DEM statistical relationships. This could be a practicable way to parameterize surface roughness and albedo on a whole-glacier scale. Additionally, LAIs coverage can be captured by high





resolution image (e.g. Takeuchi et al., 2018). LAIs coverage might be a good substitute for LAIs concentration in albedo parameterization, especially for dark and LAIs rich bare ice.

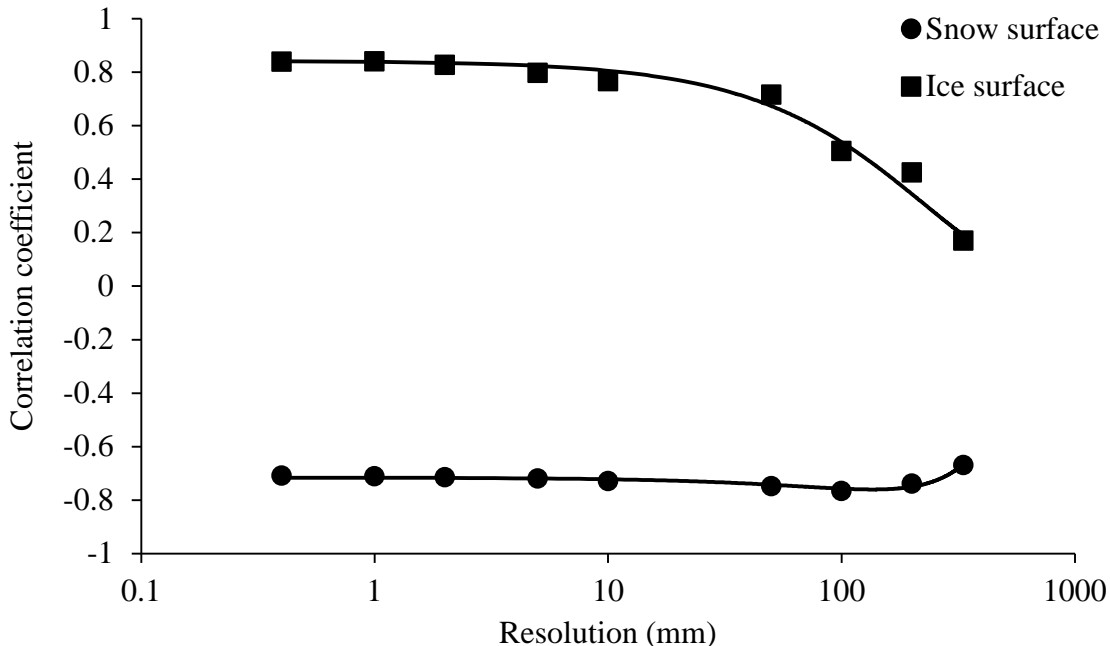


Figure 9 Correlation coefficients between albedo and surface roughness under different resolution. Snow and ice Surface roughness is estimated based on manual photogrammetric data under different resolutions at 1 m plot scale.

[Figure 9]

**8. Conclusion**

By using manual and automatic photogrammetric measurements of surface roughness, manual LAIs samples and measurement of broadband albedo at the August-one ice cap, we have a general understanding of the surface roughness that controls the albedo of snow and ice surface are quite different. This study developed methods of albedo parameterization based on either high resolution surface
roughness or both LAIs and surface roughness. Detailed analysis of albedo, LAIs concentration and surface roughness lead us to conclude that the process of surface roughness is crucially important to the snow and ice albedo estimation and necessity to consider surface roughness in the estimation of the net





shortwave radiation energy budget especially in melting season.

Surface roughness seems play a quite different role in snow surface albedo than it does for the ice surfaces.

For snow-covered surfaces, ice particle metamorphism and surface melting and refreezing induced grain size increasing synchronously happened surface roughness increasing, which induced decreasing snow surface albedo in melting season. For dirty ice surface, the increasing surface roughness increasing its surface albedo through combined hidden LAIs effect, more surface area dilute LAIs effect and protrusions induced uneven distribution of LAIs effect. Surface roughness over LAIs effect can be simplified by

consider surface roughness during calculation LAIs concentrations in equations (4). They will give respectable results even if only surface roughness data. LAIs concentration data are rarely available over whole glacier except sampled manually. Ice surface albedo parameterization method based on $C_\xi$ is not a practicable way. But consider the proportion of LAIs covered area estimated at plot scale based on high resolution images might be a good substitute for LAIs concentration in future. Correlation coefficient

analysis between surface roughness and albedo at different resolution indicate that finer scale above 50 mm and above 100 mm resolution is recommend for ice and snow surface roughness calculation, respectively.

Surface roughness develops due to local melt inhomogeneities in the melting season. Study presented by Liu et al. (2020) indicates that relative high and positive turbulent heat proportion smooth ice surface and

lower or negative turbulent heat proportion induces rough ice surface. In this study, a positive linear relationship established between ice surface roughness and albedo. Based on these two studies, we could conclude that ice surface roughness played a delicate role in turbulent heat flux and net shortwave radiation over ice surface. This smooth ice usually developed during warm and cloudy days and a smooth ice surface means more concentration of LAIs over ice surface and more efficient absorption of inward

shortwave radiation but less efficiency in turbulent heat exchange. Under a sunny and cloudy days, smooth ice surface developed to a rough ice surface because high proportion of shortwave radiation. Consistent shift of weather pattern from cloudy to sunny or vise visa induce a relatively small fluctuation of ice surface roughness during melting season. Ice surface albedo also maintained around $0.15\pm0.05$ except intermittent snowfall which increased ice surface albedo. At the end of melting season, the



roughest ice surface usually developed under consistent cold and sunny days. The rough ice surface not only induce loss of turbulent heat loss from ice surface, but also hide LAIs and increased ice surface albedo to around 0.4. Consequently, the efficiency of net shortwave absorption also reduced to its minimum and ice surface melt shutdown. In this study, we are not intent to present more quantified and detailed analysis about surface roughness role in adjust shortwave ration and turbulent heat flux. More

filed data of LAIs, surface roughness and albedo need to collect to help us to present more detailed analysis and modeling research about surface roughness and LAIs at micro scale over ice or snow surface energy and mass balance process. Our improved understanding of surface roughness and LAIs on snow and ice albedo will allow for better assessment of potential response of glacier to changing climate in future as well as improved modeling of energy and mass balance, not only in the Qilian Mountains glaciers,

but also in other regions of glaciers that experience severe LAIs effect, such as the Central Asia and Himalaya.

***Data availability.*** All of the observation data presented in this study are available upon request to the author (Junfeng Liu, liujfzyou@lzb.ac.cn).

***Author contributions.*** **JL, RC and YD** designed the study and wrote the paper. **JL and CH** carried out field manual photogrammetry observations, **YY, ZL, XW, SG, YS, WQ** sampled snow and ice surface LAIs and measured inward and outward shortwave radiation over the August-one ice cap.

***Competing interests.*** The authors declare that they have no conflict of interest.

***Acknowledgements.*** We thanks the editor Stef Lhermitte for his helpful comments that improved the
paper.

**Financial support.** This study was supported by the National Key Research and Development Project of China (2019YFC1510504) and the National Natural Sciences Foundation of China (41877163, 41671029).

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



Figure 1s LAIs samples over different snow or ice surface roughness types. From left to right, and from top to bottom, snow and ice surface types are: clean snow surface, rich in LAIs snow surface, cryoconite holes, smooth bare ice surface, protrusions, and a very rough ice surface at both macro and micro scales.


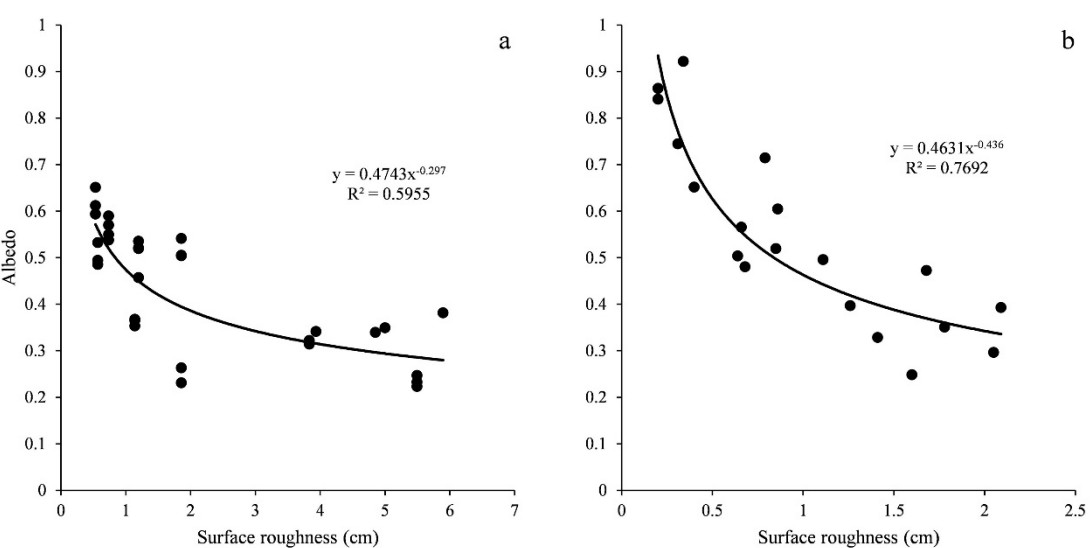

Figure 2s (a) Manual photogrammetry estimated snow surface roughness vs. albedo in the 2018 melt season, (b) Automatic photogrammetry estimated snow surface roughness vs. albedo in the 2018 melt season.


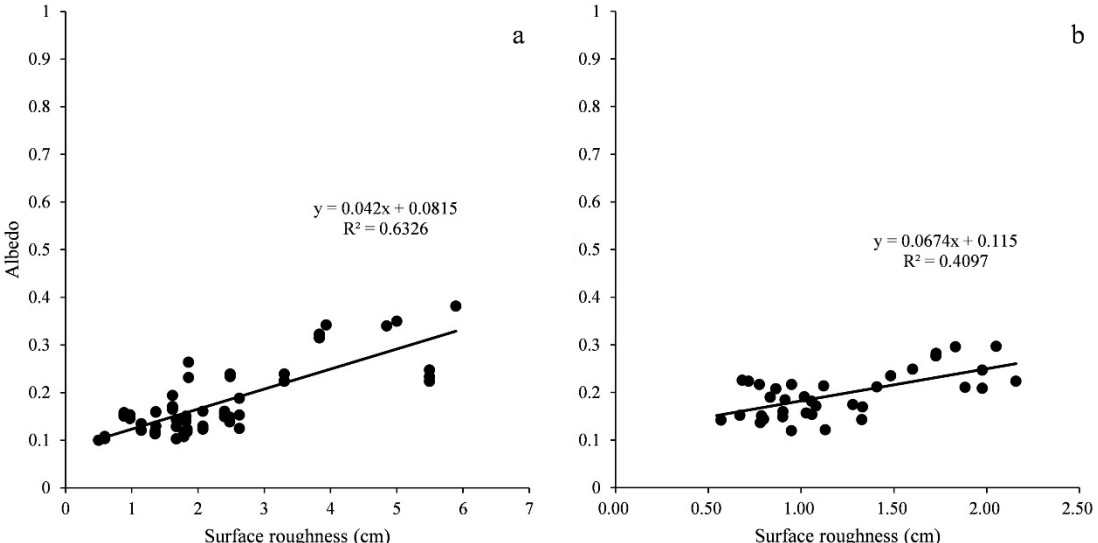

Figure 3s (a) Manual photogrammetry estimated ice surface roughness vs. albedo in the 2018 melt season, (b) Automatic photogrammetry estimated ice surface roughness vs. albedo in the 2018 melt season.



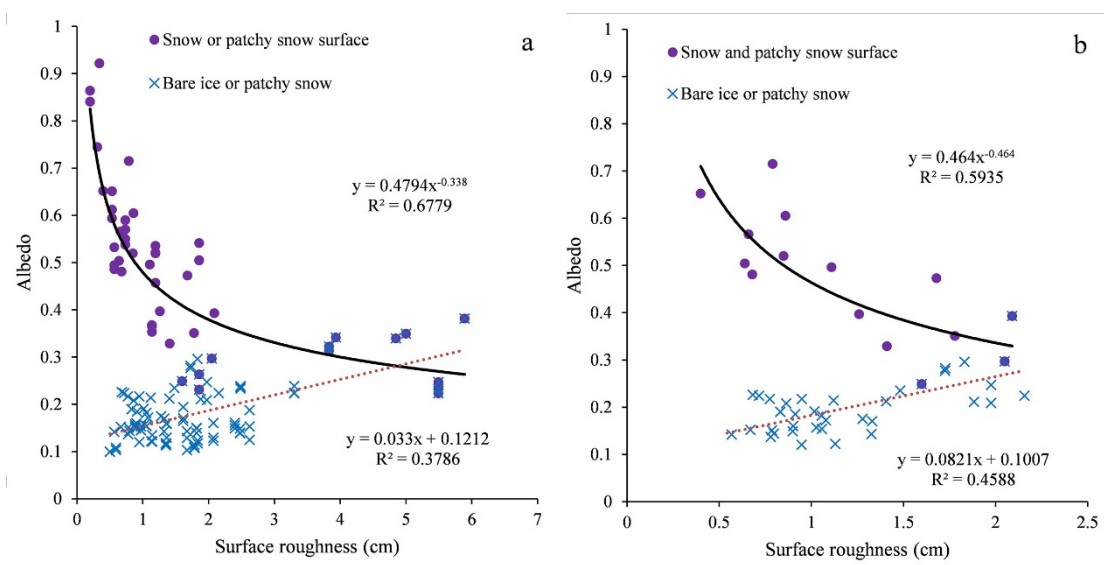

Figure 4s (a) Combined manual and automatic photogrammetry estimated surface roughness vs. albedo for snow, patchy snow and bare ice surface. (b) Automatic photogrammetry estimated surface roughness vs. albedo for snow, patchy snow and bare ice surface. Round purple dots overlapped with blue 'x' shaped points are patchy snow cover. A negative power function for snow surface roughness and albedo changed
to a positive linear relationship for ice surface roughness and albedo. A tipping point of a negative relationship between snow surface roughness and albedo changed to a positive relationship for ice surface appeared after patchy snow cover appeared.