# Peer review of "Effects of surface roughness and light-absorbing impurities on glacier surface albedo, August-one ice cap, Qilian Mountains, China."

_The Cryosphere, 2020_

## Referee Comment (RC1) · Anonymous Referee #1 · 8 May 2020

REVIEWÂă: Liu et al., TC

Journal: TC Title: Effects of surface roughness and light-absorbing impurities on glacier surface albedo, August-one ice cap, Qilian Mountains, China Author(s): Junfeng Liu et al. MS No.: tc-2020-67 MS Type: Research article

General Comments

Authors measured the spatial and temporal surface albedo variations over the Qilian mountain glacier. They analysed the albedo fluctuation considering the presence of small-scale surface roughness and light-absorbing impurities. They proposed an appropriate parametrization of albedo with these two parameters. Reviewer gives a

certain appreciation for the reasons that authors show a strong impact of these contributions on the surface albedo at different location (snow cover, ice cape, bare ice) using an extensive dataset of in situ measurements. However, unfortunately, the organization of the manuscript and presentation of the data and results are too hard to follow. In particular, it is not clear what is the novelty of this paper compared on earlier work of Warren, Cathles, Pfeffer, Lhermitte and many others. The explanations and some results are insufficients. For now, the paper is not a significant advance or contribution, even though it has the potential with interesting measurements. In addition, the structure of the paper can also be much improved and the paper can be condensed. Substantial revisions are needed before having a merit for the publication in the TC. Reviewer is particularly concerned about the fact that no SSA measurements were acquired and the impact of the variation of the snow grain sizes with the metamorphism is barely speaken. It is difficult to believe in an exhaustive study of the impact of surface roughness/LAP on albedo without having an idea of the impact of snow metamorphism, since these 3 parameters are strongly linked in albedo variations. Authors should carefully confirm the results by including this analysis, maybe by adding a section speaking of this parameter (or using existing values of snow grain size published in the litterature over this region). Moreover, reviewer suggest to modify the structure of the manuscript to present the results clearer. The organization of the manuscript and presentation of the data and results need some improvement. Indeed, sections are mixed, and there are many repetitions that can be easily avoid. The protocols of measurements are not enough detailed whereas it is very important to be confident on the measurement analysis. Finally, a deep revision of the english grammary has to be done to make the lecture easier.

Specific commentsÂă:

1) The english has to be carefully corrected. Reviewer tried to highlight some errors in the 'technical corrections' section but there are many english mistakes making the reading difficult.

2) Sections have to be read carefully to remove repetitions and to make the conclusion clearer. For example, it is often that in a 'snow section' authors speak about ice albedo measurements. Moreover, reviewer suggests to add small sentences at the end of each section to highlight the result. For example 'over ice cape, the albedo increases while the surface roughness increases and the LAI decreases'.

3) Explanation and results are insufficient: scatter plots presenting field observations have to be followed by physical analysis, according to what has been shown in the litterature. Deeper explanations are needed to better understand the novelty of this work.

4) Figures need to be well called in the text, it is hard to follow the analysis when the text does not refer to the right figure. Moreover, reviewer suggests small changes in the graphs to be clearer (in the 'technical corrections' section). For example, It could help the understanding if authors replot the Figure 3 with different symbols/colors associated to 1) measurements taken over snow cover 2) measurements taken over ice surface.

5) As said in the 'General comment' section, it is not clear what relationships have been highlightedin this paper. There is a strong relationship between SZA/SSA/LAI/roughness/albedo. The impact of the solar zenith angle and SSA variations needs to be analysed deeper.

6) The Protocols/Measurements sections need more explanations. For example, the accuracy of the instruments should be precise, and references presenting the instruments are missing. Reviewer has some questions about the protocol performed to acquired albedo measurements: How do you estimate the direct/diffuse part of the albedo? Measurements are acquired in clear sky conditions? At which height was located the sensor (ie what is the area actually seen by the sensor) ? What if you measure the aluminium square in addition to the snow surface?

7) There is a problem of unity in equation 2, epsilon is in cm while the h* variable is in meters.

8) Nothing is new in the discussion part, except the parametrization method. Authors insist on the fact that it is physically-based, but it is not, it is fitted over the area so it is empirical. The sections should be better organized, this is very confusing.

Moreover, the new parametrization should be investigated deeper, the associated error needs to be estimated for each type of surfaces (using control points over the snow covered surface, the ice surface, . . .).

Technical corrections

L13: 'Fluctuations in surface albedo are due primarily to variations in micro scale surface roughness ($\xi$) and light-absorbing impurities (LAIs) in this region.' => Please add the SSA+solar zenith angle L.22: by consider => by considering L.43: english => if we want to improve the accuracy of the energy budget estimate L.52: can accelerates => can accelerate L.54: which enhance => which enhances L. 60: Please rewrite: 'For the Qilian mountain glaciers, where the measured daily mean albedo decreased to the lowest of 0.13$\pm$0.06 due to the effect of LAIs for four glaciers observed during melting season'=> For example: For the Qilian mountain glaciers, the measured daily mean albedo decreased to 0.13$\pm$0.06 due to the effect of LAIs for four glaciers observed during melting season L63: Please put this sentence before, in the section above: 'As the snow melts, insoluble LAIs are retained at the snow surface, so concentrations of LAIs in surface snow increase with snow melt, further reducing snow albedo (Doherty et al., 2013).' L75: Studies have indicates => Studies indicated L80: the distribution LAIs=> the distribution of LAIs L85: Surface roughness structure => 'Surface roughness features' seems more appropriate.

L88: 'During the past 5 years the August-one ice cap has become darker due to the accumulation of LAIs' => Please add a reference. L90: has indicate => has indicated L97: we try to investigate => we investigated L100: at different altitudes and times => and resolution? Or please add the resolution. => This section needs to be rewrite to present clearly the plan of your study. For example: 'first, to study the spatial

variation.../second... Considering the following structure of your paper: 1rst objective: spatial variation, just the relationship between manual photo/lai/albedo 2nd objective: temporal variation. L100: Combine with => these measurements were combined with L108: based on => using only ... or both ... L118: It ranges in elevation => the elevation ranges from ... L125. The glacier becomes darker with years? It was said in the introduction. Please clarify. L134: Please specify if you are talking about 3D photogrammetric acquisitions or 2D photogrammetry, and the references need to be adapted (for instance, Manninen = 2D acquisitions with a board, Irvine-Fynn = 3D acquisitions) L139: of ∼1.75 m 2 => it is a very small area, is it representative ? How was the area chosen? L 139: Please clarify: 'by surrounding the target area of snow or ice surface', do you mean by turning around? If yes, what is the space between each step/picture? L143: Please specify the number of sampled areas. L147: 'Glacier surface albedo was calculated from measurements of up and downward shortwave radiation.' => how were acquired the up and downward shortwave radiation? Please add details about this protocol and sensor.

L160: 'For automatic photogrammetry': how were process the pictures? With a software? Did you use only one reference target? Did you use control points? What is the error of the final DEM? Please clarify.

Figure 2: Please rename a) and b) because they have the same caption. + remove '[Figure 2]' in L 181 => please removed this typo for all figures. L185: repetition => 'Ice surface albedo is very sensitive to the LAIs concentration over ice surface.' Please add a sentence of transition, for example 'the surface roughness features impact the distribution of the snow impurities'.

L190: by direct => by directly affecting L 192: There is a problem with the unity of equation 2. epsilon is in cm while the h* is in metersÂă? L192 : equation 2 is the first equation => it is equation 1 here. Please correct the other equation notations in the text. L 195: 'from Lettau (1969) developed aerodynamic surface roughness' => 'from Lettau (1969) who developed aerodynamic surface roughness...' L198: adopted =>

adapted L202: Please add a reference to introduce this metric. L205: This sentence needs to be rewrite: ÂńÂăFor rough ice surface means more surface area and less concentration of LAIs over ice surface,ÂăÂż I don't understand. L207: by consider => by considering L207: this is a new metricÂă? If yes, please change 'defined' by 'introduced'

L220: 'For that reason, we used only half-hour albedo data taken when zenith angles were less than 60°' => Did you test the accuracy of the sensor considering the zenith anglesÂă? Is this a known angle of limitation? Or please add a reference. L222: from 11:00 to 17:00 o'clock => what range of zenith angle it is? It strongly depends to the date of your measurements + elevation of the studied site. Please clarify. => It seems that this section should be written in the protocol of albedo measurements, not here in this 'result' section. L225: field investigation => is it based on your automatic photogrammeric measurementsÂă? Please clarify. L230: Surface roughness => replace it by 'the associated epsilon measurements decreased ...' L236: 'Surface roughness fluctuated between 1.4±0.4cm to 3.3±1.1cm; An increasing trend of surface roughness could be detected as altitude increased (Figure 3e). ' => please, inverse these two sentences to be clearer. L245: 'surface roughness and LAIs decreased as altitude increased.' => not clear because it is only the case in July. In August 3: surface roughness increased as altitude increased and there was some snow cover at the top of the ice cap. Please clarify. L247: 'There was a much higher concentration of LAIs on the uncovered ice surface than snow surface' => I don't understand the 'much', the trend is not so strong? It could help the understanding if your replot the Figure 3 with different symbols/colors associated to 1) measurements taken over snow cover 2) measurements taken over ice surface. L248: 'As a consequence, albedo tended to be low on the ice surface and higher on snow-covered surfaces.' => be careful with this sentence, this is also mainly due to SSA values that strongly impact the albedo. There is a strong relationship between SSA/LAI/roughness/albedo. Please rewrite this sentence. L251: repetition with L240-243 => this section needs to be re-structured in order to remove repetitions and to be clearer on conclusions.

L260: 'The automatic measurement setup in the middle of the ice cap' => 'The automatic measurement setup was in the middle of the ice cap' L.264: 'Intermittent snowfall decreased surface roughness and increased albedo' => you should add that it is because there is fresh snow, and fresh snow has high SSA, inducing a high albedo. Figure 4: It could be clearer if you add the intermittent snowfall with vertical lines + air temperatures (to know when it is melting). L265: 'When snow began to melt'=> when is it ? How do you measure thatÂă? Do you have air temperature or snow surface temperatureÂă? If yes, you could add it on Figure 4 (see comment above). L276: 'When the snow melted, leaving patchy snow cover, surface roughness increased and albedo decreased quickly to around 0.3±0.05 within two day'. Here again the sentence needs to be rewrite: snow melting = lower SSA = lower albedo, the decrease of albedo is not only due to the increase of surface roughness and LAI. Please clarify. L276: 'from September 4 to 15, ice surface roughness fluctuated and increased to 2.63 cm on September 13.' => not clear, is it increasing from september 4 to 15, or for september 4 to 13? L277: 'Ice surface micro scale structures of cryoconite holes hidden LAIs from direct sun light.' => this sentence needs to be detailed to highlight the observations. => Reviewer suggest to recall this section (for example '3- Field observations'), since it details all the field observations, but not clearly enter in the physical analysis of the relationships between LAI/roughness/albedo. The analysis should include SSA variations. Even if no measurements were taken, it has to be mentioned (relating to the snow type of the area for example).

Figure 5: Please rewrite the x axis 'Surface roughness (cm)' with the epsilon metric. L301: Figure 2s => I am not sure this figure provides a lot, since it is well resumed in Figure 5a. In Figure 5a, the measurements taken manually and automatically should be separated by two different symbols to be clearer. L302: 'The combined manual and automatic scatter diagrams of Figure 5a display a significant negative power function between snow surface roughness and snow albedo (Figure 5a, r=0.82).' => this is a repetition.

[Figure]

L314: Figure 3s => same as above, the Figure 5b gives approximately the same information. Authors could put different symbols for manual/automatic measurements on Figure5b. Figure 4s: snow, patchy snow and bare ice surface => In the 'data and methods' section you should defined this terms. How do you classify an area as a 'patchy snow'? Is there a quantitative measurement of the proportion of snow in the area, or is it qualitative? How do you distinguish areas with very dense snow surface to ice surface? Please clarify. L 324: 'Scatter plots of Figure 5c and 5d shows relationship between C LAIs and albedo' => Reviewer suggests to introduce a new section dedicated to LAI-albedo relationship, or to separate the section to put the analysis of the LAI-albedo relationship over snow surface in the 'snow surface section', and over ice surface in this 'ice surface melting season'. L325: 'LAIs concentration is lower than patchy snow surface' => ' LAIs concentration is lower than the one measured over patchy snow surface' L327: Figure 4c => Figure 5c, and Figure 4d => Figure5d Please a particular attention should be taken on the notation of Figures in the text, this is confusing.

L329: Please introduce a new section, or reorganise this section into the 'snow surface section' and the 'ice surface section'. L329: 'the effective LAIs concentration of C $\xi$' => 'the effective LAIs concentration (C $\xi$)' ' by considers roughness effect' => 'by considersing roughness effect' L330: C $\xi$ is estimated based on equation (4). => this is a repetition L334: 'It indicateS that the Ceps concentration IS not affectED by surface roughness.' => english L335: 'Equation (3) is more appropriate to calculate LAIs concentration than equation (4) over snow surface.' => I don't understand. Do you mean 'to retrieve LAI concentration directly from albedo measurements'? Please clarify. L339: 'by consider surface roughness effect over not consider it in Figure 5d' => english L341: 'It means equation (4) estimate C $\xi$ could explain more ice surface albedo than equation (3) calculate C LAIs .' => please clarify. Do you meanÂǎ: could explain ice surface albedo variationsÂǎ? L352: english mistakes L354: 'In accumulation season, except snow particles metamorphism process, constant blowing snow and intermittent snowfall was the main reasons which induced surface roughness fluctuation;' => More explanations are needed. In this Result section: the english has to be carefully corrected, some sections has to be restructured, the field observations and scatter plots have to be followed by physical analysis, according to the litterature, with more explanations. Figures need to be well called. Some changes in the graphs are suggested to be clearer.

L380: 'The grain size is one of the most critical factors affect snow albedo' => 'The grain size is one of the most critical factors affectING snow albedo.' L385: 'which is very similar with scatter plot of surface roughness and snow albedo provided in this study in Figure 5a and Figure 7.' => Please specify if this studies worked with the same snow type as you.

L386: 'It means surface roughness as substitute of grain size is quite suitable for snow surface albedo explanation and parameterization at millimeter scale' => Please be careful with this conclusion, the study areas are not the same. To conclude this you should have both measurements (ssa and roughness) at the same site, at the exact same time (same sza) and over a dry and clean snow (no impurities, no liquid water content). Please clarify. Several contributions affect the albedo, and it is very difficult to separate each contribution to better understand/quantify their impacts on the albedo. L388: english L406: 'We expect that different mechanism of surface roughness evolution over melting season and accumulation season are the main reasons of different statistical relationship.' => This is not the main reason, the SSA variation also plays a strong role. Please clarify.

Section 7.2: The name of the section should be changedĂă: authors speak about macro-scale surface roughness effects in this section, and in the section before it was at millimeter-scale. L415: 'more incident radiation is absorbed by the slope facing the sun than by the slope facing away from it.' => this is not the only reason (see Warren et al 1998). The main reason is that photons are trapped between cavities, which increases the chance of a photon to be absorbed. Please clarify and detail the physical analysis. L430: filed investigation => english L430-433: Please rewrite the

sentence, I do not understand L441: 'Rough surface means lower concentration of LAIs and high albedo over smooth ice surface with heavy loading of LAIs and low ice surface albedo.' => this is confusing to speak about rough surface of a smooth ice surface.. how do you define a smooth ice surface? What is the scale of roughness features that you are talking about? Please clarify. => this is very confusing between Sections 7.1 and 7.2 => when are you talking about ice surface and snow surface? For example: the first section of the section 7.2 is talking about snow surface (see the litterature), and not the ice surface. Please clarify. L447: 'Most of the established and widely used snow surface albedo parameterization ARE either based on snowpack age, snow depth, snow density, air temperature' => english => Please add the SSA and SZA (see the litterature of Kokhanovsky) => This sentence should be put in the section 'snow surface', not 'ice surface'. L450. Please add a reference L451: 'We expect the new parameterization methods provides' => english 'are more physically based than some of the studies presented.' => What new parametrization? Please refer to the equation here. (as equation 5 but for ice surface) This sentence needs to be rewritten: this is not a physically based parametrization, it is empirical. Indeed your equation 5 strongly depends of your study site and your measurements.

L466: 'For snow covered surface, the coefficient increases quickly from -0.67 to - 0.74 when 1m plot resolution increases from 333.3 mm to 200.0 mm' => but the Figure 8a shows a very flat and smooth snow surfaceÂă? Please clarify the studied surface. L485: 'This could be a practicable way to parameterize surface roughness and albedo on a whole-glacier scale' => Please add some details. It could be interesting to apply the method proposed by Smith et al 2016 here, to estimate a temporal mean albedo at the whole-glacier scale.

Section '7.3 Glacier surface albedo parameterization at whole-glacier scales based on surface roughness' => the title needs to be changed since you investigate the sensitivity of the albedo relationships at a plot-scale, but not at the whole-glacier scale in this section. => Moreover, please rely your analysis with the litterature (Irvine-Fynn et al

2014; Smith et al 2016 ..) to clarify what is the novelty of what you propose. => It could be interesting to estimate the LAI coverage with your photogrammetric acquisitions as Takeuchi et al 2018.

L497: 'we have a general understanding of the surface roughness that controls the albedo of snow and ice surface are quite different.' => english L502: 'necessity to consider' => english L504: 'Surface roughness seems play a quite' => english L535: 'field data of LAIs, surface roughness and albedo need to collect to help us to present more detailed analysis and modeling research about surface roughness and LAIs at micro scale over ice or snow surface energy and mass balance process.' => SSA measurements should be acquired too.

---

## Referee Comment (RC2) · Alvaro Ayala (Referee) · 9 Jul 2020

Review for The Cryosphere

Title: Effects of surface roughness and light-absorbing impurities on glacier surface albedo, August-one ice cap, Qilian Mountains, China. Authors: Liu, Chen, Ding, Han, Yang, Liu, Wang, Guo, Song, Qing

PAPER SUMMARY AND RECOMMENDATION

Liu et al. present a dataset consisting of field measurements of albedo, surface roughness and surface concentration of Light-Absorbing Impurities (LAIs) collected during a

melt season on the August-one ice cap in the Qilian Mountains, China. The measurements were collected from July to October 2018, spanning most of that melt season (July to September) and one month (October) of the next accumulation season. The collected data are used to propose statistical relations between the three main monitored variables (albedo, surface roughness and surface concentration of LAIs), and to discuss the physical relations between them, among other analyses. In particular, they propose that there is a combined effect of surface roughness and LAIs surface concentration on the albedo of snow and ice.

I think that the topic of this investigation is relevant for the scientific community and The Cryosphere. Although some additional details about the sampling methodology are needed, the data seem well collected, and the physical processes that are proposed to explain the observations are interesting and reasonable. Unfortunately, I don't think that the authors convincingly prove their hypothesis about the combined influence of surface roughness and LAIs, because i) they tend to ignore or minimize the role of snow metamorphism on snow albedo, and ii) the impact of snow patches on a bare ice surface. Please see my main point below for details in this regard. Importantly, the structure of the article is confusing, and the text and figures can be condensed and better organized. Finally, the use of English language should be revised, as it sometimes obscure the ideas of the authors. My recommendation is to reconsider this article after major revisions.

GENERAL COMMENTS

1. Physical connections between surface roughness, LAIs and albedo

a. Process understanding

My main comment is that the authors tend to overestimate the influence of surface roughness on albedo in their analyses and conclusions. I have no doubt that there is a statistical relation between these two variables (as shown in Figure 5), and also some physical connection (Larue et al., 2020), but the collected data does not necessarily

suggest a causal relation. As the authors do in the paper, I divide my arguments for the periods of snow melting and bare ice melting.

- Period of bare ice exposure: Figures 2b and 2c present a good summary of the main idea discussed by the authors. A rough ice surface (with a surface roughness of about 1-10 cm) hides some LAIs, raising the area-average albedo relative to surfaces with uniform debris cover (Bøggild et al., 2010). If the surface smooths over the melting season, the LAIs emerge and the albedo decreases. The authors parameterized this effect using equation (4). However, based on the data shown by the authors, I would say the positive relation between surface roughness and albedo over an ice surface (Figure 5b) can be solely explained by the high-albedo snow patches (shown in Figure 4f) resulting from the snowfall events discernible from the occasional albedo increases during August (Figure 4a). In that respect, can the authors add a figure with the meteorological variables measured at the AWS shown in Figure 1? I think that in order to demonstrate that the albedo increases are only due to changes in surface roughness, the authors would need to show data without snow patches. Otherwise, it seems very plausible that the high-albedo snow patches resulting from the occasional summer snowfall events are responsible for the positive relation between surface roughness and albedo, during the ice melt period.

- Period of snow melting. It is well known that snow albedo decreases due to snow metamorphism and large grain sizes, which affects the light scattering. The increasing grain size can have an effect on the surface roughness, but the latter is also explained by differential melting of the surface. From lines 505-507, the authors seems to suggest that the lowering of the albedo is caused only by the increasing surface roughness.

b. "Physically-based" parameterizations and their potential for large-scale albedo estimates

The statistical relations between albedo, surface roughness and concentration of LASs presented by the authors are described as "physically-based". Although it is interesting

that the authors include these variables in an albedo parameterization, these equations are not "physically-based", because they are not derived from a fundamental principle or process (e.g. energy conservation) and their parameters (obtained from a scatter plot) cannot be directly measured in the field.

In the abstract and conclusions, the authors suggest that albedo can be retrieved from surface roughness at large scales. However, this seems unrealistic because the necessary high-resolution DEMs to derive surface roughness can only be obtained by sensors carried by Unmanned Aerial Vehicles (UAVs), which are only used at the glacier-scale. Are the authors referring to that scale? Using satellite products, albedo is much easier to retrieve than surface roughness at regional scales.

2. LAIs sampling

More details are needed regarding the LAIs sampling. How did you calculate the volume of each sample? Is that the volume of the stainless steel spoon mentioned in line 150? How did you make sure that the volume was the same in every collected sample? Can you include a picture of the spoon? How did you sampled sites with a rough surface, such as one with cryoconites? Did you take the samples at the top or the bottom?

3. Paper structure and English language

Please improve the paper structure. I have the following suggestions: - Section 2.1 is neither data nor methods. Please add a new section called "Study Site". - Sections 3 and 4 consist of results. I suggest including these sections in a traditional "Results" section. - Sections 3 and 4 have a confusing structure, partly because the titles are ambiguous and repetitive. - The numbering of sections 7 and 8 is wrong. They should be 5 and 6.

I have provided many suggestions to improve the use of English language, but I think that the authors should perform a full revision of the text.

[Figure]

**SPECIFIC COMMENTS**

11: "Fluctuations in surface albedo are due primarily to variations in micro scale surface roughness ($\xi$) and light-absorbing impurities (LAIs) in this region." I guess you mean fluctuations of bare ice albedo, because snow albedo variations are very large and are due to metamorphism. See my main comment 1a.

21: "$\xi$ could explain 68%of snow surface albedo and 38%of ice surface albedo variation in melt season." When you write that surface roughness explains 68% of the snow albedo variation, I think that you need to explicitly state that this is a statistical analysis, because the physical explanation for snow albedo variations is snow metamorphism and an increasing grain size. See my main comment 1.

37: "According to Hock (2005), on average it accounts for over 70% of the net energy input to glacier surfaces." You should mention that that number was obtained for a particular glacier (Storglaciären). In any case, what do you mean exactly by "net energy input"? Net = input - output. Table 1 in Hock (2005) shows that incoming longwave radiation (L_in) is the largest energy input. In general, L_in and S_net are the largest energy inputs to glaciers at daily, and longer, time scales (Ohmura, 2001).

78-79: "poorly investigated, and snow surface albedo parameterization methods based on surface roughness are rarely reported." However, the relation between snow surface albedo and grain size has been largely analyzed. Please see my main point 1a for snow surfaces.

86: "Surface roughness structures developed during melt season such as crevasses, cyroconite holes, can increase ice surface albedo by hiding LAIs from direct sunlight have been widely reported." Although, more than "increasing" surface albedo, I would say that they "prevent", or at least "delay", a further decrease, because they hide LAis that would otherwise reduce the area-average albedo.

229: How do you calculate the average value and uncertainty of albedo, surface rough-

ness and LAIs concentration at a particular elevation?

242: "LAIs decreased from 0.04±0.03g/ cm3 at middle part to 0.003±0.002g/ cm3 at higher elevations (Figure 3i)." This is based only in one sample at 4700 m, if you delete it, you don't have any trend. Did you try any other topographic parameter?

247-249: "There was a much higher concentration of LAIs on the uncovered ice surface than snow surface. As a consequence, albedo tended to be low on the ice surface and higher on snow-covered surfaces." This is not only explained by the LAIs. Ice albedo is usually lower than snow albedo.

378-379: "Although we do not have tandem surface roughness and grain size observations, the evolution of surface roughness calculated at 1mm resolution in snow covered period should quite similar with fluctuation trend of grain size evolution." Certainly, you have a correlation between grain size and surface roughness, but surface roughness is not only explained by grain size. Differential melting of the surface could be also important.

423-424: "...a significant positive relationship rather than a negative relationship was established over ice surface based on manual and automatic measurements. We expect it is related with abundant LAIs over ice surface at the August one ice cap." This might be explained by the snow patches. See my main comment 1.

448-450: "The performances of the establishes albedo methods either based on surface roughness, LAIs or effective LAIs concentration shows a great improvement over the assumption of a constant mean ice albedo or surrogate variables, such as air temperature, accumulated melt and elevation." I don't agree that surface roughness is much better than air temperature or melt to parameterize albedo. Please see main comment 1b.

462-464: "Since surface roughness is dependent and sensitive to topography data resolution (Figure 8a), so which resolution is appropriate for snow and ice surface

albedo estimation?" Was this part of the objectives? Are you writing about the DEM resolution? Please be more explicit in this section and present this analysis earlier in the text.

496-498: "By using manual and automatic photogrammetric measurements of surface roughness, manual LAIs samples and measurement of broadband albedo at the August-one ice cap, we have a general understanding of the surface roughness that controls the albedo of snow and ice surface are quite different." Can you really conclude that surface roughness controls the albedo of snow and ice? See my main comment 1.

505-507: "For snow-covered surfaces, ice particle metamorphism and surface melting and refreezing induced grain size increasing synchronously happened surface roughness increasing, which induced decreasing snow surface albedo in melting season." The last part of the sentence suggests that is the increasing surface roughness that induces the snow albedo decrease.

SUGGESTED TECHNICAL CORRECTIONS

4: Please make the format for last names uniform (sometimes is uppercase and sometimes not).

11: influence on melt

16: the present study consisted of an intensive. . .

19: the middle

19: A detailed analysis indicates that. . .

20: a positive linear

21: for snow and ice, respectively

22: consider -> considering

25: constant mean -> uniform

33: The energy balance and resulting melt rates...

33: the meteorological conditions

34: the physical properties

34: Delete ", which determine glacier melt process"

35: the melting of glacier -> glacier melt

35: dominated-> controlled

35: in the glacier surface -> at the...

36: Shortwave radiation is the main energy input causing snow and ice melt

39: net radiation: do you mean net shortwave radiation?

45: "Glacier albedo varies much more dramatically than other land covers" -> Albedo varies much more dramatically on glacier surfaces than on other land covers.

47: "constantly changing surface characteristics"-> I think that you should explicitly mention snow metamorphism here.

47: the solar incidence angle

52: accelerate

57: Simulations

59-62: In general, I suggest improving the writing style by making sentences shorter and more fluid. Here, I would suggest something like: LAIs have decreased the surface albedo of glaciers in Qilian mountains to values as low as 0.13+-0.06 during the melting season (LIT).

59: Please introduce the Qilian mountains before this sentence.

64: I think that you should start a new paragraph when you mention surface roughness.

70: do you mean "first recognized by Kuhn"?

71-72: In which way these field campaigns advanced the influence of roughness on albedo?

75: Studies. Which ones?

75: indicated

75: the inclusion

76: measurements of albedo? Sorry, this is not clear.

76: equations

77: and sastrugi

80: heterogeneous: Please explain better what do you mean by this.

80: of LAIs.

83-85: Please improve the structure of this key sentence.

85: the melt season,

85: crevasses and

86: the direct sunlight

88: Please introduce the ice cap more formally. Where is it? You can give the details in the next section, but give at least some indication here.

90: indicated

93: LAIs on ice

95: combined effects: This is very interesting, and difficult, how do you isolate both

effects? Please see my main comment 1.

96: have been rarely

92: we investigate

93: better understand and simulate. I think that these are very wide and general objectives. Can you be more specific?

106: at the daily scale

101: we investigate

104: on the middle

104: in melting season and accumulation season of 2018 -> in the melting and accumulation season of 2018.

107-108: This is not clear at this point.

114: Study area is neither data nor methods. Create another section.

116: the Qilian Mountains

120: May to September. Mention that this is summer.

120: How short is summer?

126: has been observed.

133: This section should be called data collection and image processing.

137: In some parts of the text you write "we" and in others you write "researchers". Please make it uniform. I would suggest to use an active voice, i.e. "we".

142: several different locations. How many? Make a reference to Figure 1. You don't have any table in your article. It would be good to have one with the coordinates, number of measurements, etc.

142: the top of the ice cap

144: the micro scale topography over different altitudes

145: Physical->You mean topographical in opposition to the turbulent heat fluxes parameter (surface roughness length)?

145: was estimated. How?

146: putting->placing a Kipp and Zonen CMP11 radiation sensor

152: in an oven

156: its operation

157: geo-reference: what do you mean?

159-160: "The photography was repeated at three hour intervals from 9:00 AM to 18:00 AM, UTC+7 time." But later you work with daily means, don't you?

159: how many different locations?

168: How did you make sure that the frame stayed horizontally?

185-189: Repetition

190: that surface roughness

190: by directly affecting

195: from the aerodynamic surface roughness parameter developed by Lettau (1969)

196: except that we changed the silhouette area facing upwind to. . .

203: Is V the volume of the spoon?

210: We assumed that

220-224: This sounds like methods to me. Is this the Results section? See my minor

comments.

225: You also have temporal variability here, not only spatial. Please change the title.

226-227: Again, I suggest shorter and more fluid sentences, something like:

We found a patchy snow cover with many cryoconite holes on the glacier terminus (∼4600 m) at the date of the first field trip (July 12).

230: Be consistent with the digits, i.e. 5.49 cm = 5.5 cm.

244-245: On July 12, surface. . .

246-247: the transient snowline retreated up-glacier.

247-248: On what plot do you base that sentence

248: bare ice surface than on snow.

249-250: Albedo didn't really increased over time.

251: minimal albedo: Do you mean the season minimum?

251: "but it also did not increase"-> and it did not show a clear trend with altitude

259: This is almost the same title as the previous sub-section.

262: October 17

270: mostly bare ice with occasional snowfall events

285: was mainly induced

259: The title of this section is very similar to that of section 3.

298-300: Snow surface albedo and the corresponding surface roughness are analyzed using. . .

301: functions

301: observations

304: Please use the same coefficient in the text and the figure (either r2 or r).

321: What is a patchy ice cover? A bare ice surface covered by snow patches or by debris patches?

322-323: This is not clearly written. Better say that the patchy snow cover is the limit between the two periods that you identified.

324: the relationship

324-325: Delete "without consider surface roughness effect over LAIs concentrations."

328: Do you mean 5d?

328: manually

331: power function, do you mean linear function?

333: that considering

334: ...albedo during the snow period.

334: It indicates

365-373: All these lines can be removed, moved to the introduction, or condensed.

369: Meltwater

378: should be quite similar

380-383: This should be mentioned earlier in the text.

386: it means that

388: LAIs are another critical factor affecting snow albedo

392: "increasing surface roughness decrease albedo" But, only in your snow melting

period

396-400: This should be in results.

400: What is the explained variance of each variable?

402: During the accumulation season

403: Please explain better the observed snow metamorphism in the accumulation period

403: During the accumulation period

413-421: This belongs to the introduction

430-433: Please re-write this sentence in a clearer way.

437: over smooth ice -> than smooth ice?

447: are based

TABLES

There are no tables in the article. At least one summarizing the distributed measurements would be useful.

FIGURES

Figure 1: Please add an inset showing the ice cap and the Qilian Mountains in the Tibetan Plateau.

Figure 2: Why is important to have rain or fair weather? Provide a brief explanation.

Figure 3: I would place altitude in the x-axis. Elevation should not be a dependent variable.

Figure 3: In the text of this figure, you write several times that there is a relation with the altitude, but based on your data, this is true only for 3a and 3b.

Figure 4: Make a uniform tick spacing in the x-axis. Example, 1-07 15-07, etc.

Figure 4: From this plot is evident that there were 3 or 4 snowfall events in August. See my main comment 1.

Figure 4: Add a line in panel a to show when were the pictures taken.

Figure 4: Show the dates of the pictures in the pictures.

Figure 5a: Note that for low albedo (<0.4) you can get very different surface roughness (1-6 cm).

Figure 5: Consider the use of different markers (in a-b and c-d) to reduce the number of panels. As in Figure 4s.

Figure 5: I would strongly suggest using only one paragraph to describe one figure. This is a big help for readers.

Figure 5: How many different sites are included in the manual measurements shown in this figure?

Figure 6: Can you merge both plots in one panel by using different markers? A logarithmic scale might be useful.

Figure 7: Apart from the high albedo for low surface roughness, there is a poor relation between the two variables. Maybe move to the supplementary?

Figure 8: What is the difference between these pictures and those shown in Figure 1s?

Figure 2sa-3sa: How many different sites did you use to build this plot?

Figure 4s: This is a very nice figure, because it shows both periods in the same plot. Can you move this figure, or a similar one, to the main text?

REFERENCES

Bøggild, C. E., Brandt, R. E., Brown, K. J., and Warren, S. G.: The ablation zone

in northeast Greenland: ice types, albedos and impurities, Journal of Glaciology, 56, 10.3189/002214310791190776, 2010.

Ohmura, A.: Physical Basis for the Temperature-Based Melt-Index Method, J. Appl. Meteorol., 40(4), 753–761, doi:10.1175/1520-0450(2001)040<0753:PBFTTB>2.0.CO;2, 2001.

Larue, F., Picard, G., Arnaud, L., Ollivier, I., Delcourt, C., Lamare, M., Tuzet, F., Revuelto, J., and Dumont, M.: Snow albedo sensitivity to macroscopic surface roughness using a new ray-tracing model, The Cryosphere, 14, 1651–1672, https://doi.org/10.5194/tc-14-1651-2020, 2020.

---

## Author Comment (AC1) · 27 Jul 2020

The authors would like to sincerely thank the referee 1 for the valuable, constructive and for opening the discussion. Referee's detailed comments certainly helped to improve the manuscript. The corresponding changes and refinements have been made in the revised paper and are also summarized in our reply below.

Please also note the supplement to this comment:
https://tc.copernicus.org/preprints/tc-2020-67/tc-2020-67-AC1-supplement.pdf

[Figure]

**Supplement:**

**Effects of surface roughness and light-absorbing impurities on glacier surface albedo, August-one ice cap, Qilian Mountains, China**

**The authors would like to sincerely thank the referee for the valuable, constructive and for opening the discussion.** Referee's detailed comments certainly helped to improve the **manuscript. The corresponding changes and refinements have been made in the revised paper and are also summarized in our reply below.**

**Reply to comments from Anonymous Referee #1**

General Comments

Authors measured the spatial and temporal surface albedo variations over the Qilian mountain glacier. They analyzed the albedo fluctuation considering the presence of small-scale surface roughness and light-absorbing impurities. They proposed an appropriate parametrization of albedo with these two parameters. Reviewer gives a certain appreciation for the reasons that authors show a strong impact of these contributions on the surface albedo at different location (snow cover, ice cape, bare ice) using an extensive dataset of in situ measurements. However, unfortunately, the organization of the manuscript and presentation of the data and results are too hard to follow. In particular, it is not clear what is the novelty of this paper compared on earlier work of Warren, Cathles, Pfeffer, Lhermitte and many others. The explanations and some results are insufficients. For now, the paper is not a significant advance or contribution, even though it has the potential with interesting measurements. In addition, the structure of the paper can also be much improved and the paper can be condensed. Substantial revisions are needed before having a merit for the publication in the TC. Reviewer is particularly concerned about the fact that no SSA measurements were acquired and the impact of the variation of the snow grain sizes with the metamorphism is barely speaken. It is difficult to believe in an exhaustive study of the impact of surface roughness/LAP on albedo without having an idea of the impact of snow metamorphism, since these 3 parameters are strongly linked in albedo variations. Authors should carefully confirm the results by including this analysis, maybe by adding a section speaking of this parameter (or using existing values of snow grain size published in the literature over this region). Moreover, reviewer suggest to modify the structure of the manuscript to present the results clearer. The organization of the manuscript and presentation of the data and results need some improvement. Indeed, sections are mixed, and there are many repetitions that can be easily avoid. The protocols of measurements are not enough detailed whereas it is very important to be confident on the measurement analysis. Finally, a deep revision of the english grammary has to be done to make the lecture easier.

**Reply: Thanks for your comments. We have revised the manuscript based on your suggestions and comments. We have revised the manuscript substantially: the structure of the paper has changed to present the results of snow part and ice part clearer. The measurements of albedo are introduced in detail. We add snow metamorphism and corresponding surface roughness changed in the revised manuscript by adding detailed analysis of field observations and manual**

**photography(We carried out field observations of snow grain size and corresponding surface roughness based on you suggestion in July of 2020 in August-one ice cap). Especially the manual photogrammetry could give provide direct visual evidence of old snow have rougher surface over fresh snow. The revision of the English grammar are also done to make the lecture easier to read.**
Specific comments :

1) The english has to be carefully corrected. Reviewer tried to highlight some errors in the 'technical corrections' section but there are many english mistakes making the
reading difficult.
**Reply: Thanks for your comments. We have revised the English mistakes based on you 'technical corrections' suggestions and polished by language services.**
2) Sections have to be read carefully to remove repetitions and to make the conclusion clearer. For example, it is often that in a 'snow section' authors speak about ice albedo measurements. Moreover, reviewer suggests to add small sentences at the end of each section to highlight the result. For example 'over ice cape, the albedo increases while the surface roughness increases and the LAI decreases'.
**Reply: Thanks for your suggestions. We have revised based on your suggestions, and add small sentences at the end of each section to highlight the result.**
3) Explanation and results are insufficient: scatter plots presenting field observations have to be followed by physical analysis, according to what has been shown in the literature. Deeper explanations are needed to better understand the novelty of this work.
**Reply: We have add physical analysis according literature of similar research or relevant topics. This work not only considers surface layers snow or ice surface characteristics (surface roughness), but also considers LAI, and considers these two factors as important surrogate variables to explain more albedo variation than formerly used parameters. We have cited works such as Jonsell et al. (2003) who mentioned that uppermost surface ice layer, including its density, crystal structures, surface roughness, impurity content and stratification and SSA, are subject to continuous changes and mainly responsible for the observed variability in ice albedo. In this work we abstract the surface layers surface roughness might be an effective surrogate variable to be applied in albedo parameterizations.**
4) Figures need to be well called in the text, it is hard to follow the analysis when the text does not refer to the right figure. Moreover, reviewer suggests small changes in the graphs to be clearer (in the 'technical corrections' section). For example, It could help the understanding if authors replot the Figure 3 with different symbols/colors associated to 1) measurements taken over snow cover 2) measurements taken over ice surface.
**Reply: we have revised accordingly, in the revised manuscript, we differentiate the manual and automatic observations, we also differentiate the snow and ice surface. We also include key meteorological air temperature and precipitation information in the revised manuscript.**
5) As said in the 'General comment' section, it is not clear what relationships have been high lighted in this paper. There is a strong relationship between SZA/SSA/LAI/roughness/albedo.

The impact of the solar zenith angle and SSA variations needs to be analysed deeper.

**Reply: Thanks for your excellent suggestions. We have add SSA, LAI, SZA effect in analysis. The impact of the solar zenith angle are mentioned in the observation part. In the August-one ice cap, the ice surface is not sensitive to zenith angle because of high concentration of LAI over surface and melt water. The snow surface is sensitive to zenith angle. For that reason, we calculated daily albedo from the integrated sum of incoming and outgoing shortwave radiation of only 10min albedo data taken when zenith angles were less than 60°, in an effort to minimize solar zenith angle effects.**

6) The Protocols/Measurements sections need more explanations. For example, the accuracy of the instruments should be precise, and references presenting the instruments are missing. Reviewer has some questions about the protocol performed to acquired albedo measurements: How do you estimate the direct/diffuse part of the albedo? Measurements are acquired in clear sky conditions? At which height was located the sensor (ie what is the area actually seen by the sensor) ? What if you measure the aluminium square in addition to the snow surface?

**Reply: we have detailed the observation in the revised manuscript. We do not differentiate direct and diffused part. The observation include cloudy and clear sky conditions. The sensor is 1m above surface for manual observations, and 1.5m above surface for automatic observations. The surface roughness is measured first. After then, the aluminum square is moved and albedo was measured.**

7) There is a problem of unity in equation 2, epsilon is in cm while the h* variable is in meters.

Reply: **We have revised accordingly.**

8) Nothing is new in the discussion part, except the parametrization method. Authors insist on the fact that it is physically-based, but it is not, it is fitted over the area so it is empirical. The sections should be better organized, this is very confusing. Moreover, the new parametrization should be investigated deeper, the associated error needs to be estimated for each type of surfaces (using control points over the snow covered surface, the ice surface).

**Reply: we have revised the physically-based as surrogate variable. The validation of new parameterization method was not present because of lake snow and ice surface LAI observation in 2019.**

Technical corrections

L13: 'Fluctuations in surface albedo are due primarily to variations in micro scale surface roughness (_) and light-absorbing impurities (LAIs) in this region.' => Please add the SSA+solar zenith angle

**Reply: Thanks for your suggestion. We have add SSA and solar zenith angle in the abstract and revised as' Fluctuations in surface albedo are due primarily to variations in micro scale surface roughness ($\xi$), snow surface area, solar zenith angle and light-absorbing impurities (LAIs) in this region.'**

L.22: by consider => by considering

**Reply: we have revised accordingly.**

L.43: english => if we want to improve the accuracy of the energy budget estimate

**Reply: Thanks, we have revised accordingly**

L.52: can accelerates => can accelerate

**Thanks, we have revised accordingly.**

L.54: which enhance => which enhances

**Thanks, we have revised accordingly**

L. 60: Please rewrite: 'For the Qilian mountain glaciers, where the measured daily mean albedo decreased to the lowest of 0.13_0.06 due to the effect of LAIs for four glaciers observed during melting season'=> For example: For the Qilian mountain glaciers, the measured daily mean albedo decreased to 0.13_0.06 due to the effect of LAIs for four glaciers observed during melting season

**Reply: Thanks for your suggestions, we have revised as 'For the Qilian mountain glaciers, the measured daily mean albedo decreased to 0.07~0.13 due to the effect of LAIs for four glaciers observed during melting season (Jiang et al., 2010; Liu et al., 2014; Qing et al., 2018; Sun et al., 2018)'.**

L63: Please put this sentence before, in the section above: 'As the snow melts, insoluble LAIs are retained at the snow surface, so concentrations of LAIs in surface snow increase with snow melt, further reducing snow albedo (Doherty et al., 2013).'

**Reply: Thanks for your suggestion, we have revised and delete this sentence here accordingly. We add this sentence in the before section.**

L75: Studies have indicates => Studies indicated

**Reply: Revised.**

L80: the distribution LAIs=> the distribution of LAIs

**Reply: Revised**

L85: Surface roughness structure => 'Surface roughness features' seems more appropriate.

**Reply: Revised**

L88: 'During the past 5 years the August-one ice cap has become darker due to the accumulation of LAIs' => Please add a reference.

**Reply: Thanks for your suggestion, we have revised as' Successive time-lapse photography of the August-one ice cap has indicate that ice surface become darker due to the accumulation of LAIs in melting season (Qing et al., 2018).'**

L90: has indicate => has indicated

**Reply: revised**

L97: we try to investigate => we investigated

**Reply: revised**

L100: at different altitudes and times => and resolution? Or please add the resolution. => This section needs to be rewrite to present clearly the plan of your study. For example: 'first, to study the spatial variation.../second: : : Considering the following structure of your paper: 1rst objective: spatial variation, just the relationship between manual photo/lai/albedo 2nd objective: temporal variation.

**Reply: Thanks for your valuable suggestion, we have revised. This paragraph is revised as''**

L100: Combine with => these measurements were combined with

**Reply: we have revised it.**

L108: based on => using only : : : or both : : :

**Reply: we have revised 'either' as 'only'.**

L118: It ranges in elevation => the elevation ranges from : : :

**Reply: Revised.**

L125. The glacier becomes darker with years? It was said in the introduction. Please clarify.

**Reply: we have 3 automatic time lapse photography site distributed over the August-one ice cap from terminal to top. The ice surface LAIs (most cryoconites) concentrated over ice surface in summer. For the last 6 years, no accumulation was observed even at top of the ice cap, the concentration of cryoconites has increased because of negative annual mass balance.**

[Figure]

**Figure 1 Cryoconites covered ice surface (grey part is sun dried cryoconite, brown part is wet cryoconite).**

L134: Please specify if you are talking about 3D photogrammetric acquisitions or 2D photogrammetry, and the references need to be adapted (for instance, Manninen = 2D acquisitions with a board, Irvine-Fynn = 3D acquisitions)

**Reply: we have revised as '3-D photogrammetry'**

L139: of _1.75 m 2 => it is a very small area, is it representative ? How was the area chosen?

**Reply: The August-one ice cap is a small ice cap without obvious surface movement. The cap surface shows no relevant morphologies. Structures such as crevasses, cracks, and seracs are not formed. Only water channels formed over ice surface with very small proportion. In this study we selected all ice surface types except water channels. The selected snow or ice surface include smooth snow covered surface, partial snow covered surface, cryoconite holes, weathering crust and smooth ice surface.**

**In the revised manuscript, we have add the selection standard for manual photogrammetry on 12 and 25 July, and 28 August.**

L 139: Please clarify: 'by surrounding the target area of snow or ice surface', do you mean by turning around? If yes, what is the space between each step/picture?

Reply: Thanks for your question, we have revised as' **Seven to fifteen photos were taken at each survey site by surrounding the portable aluminum square. At each cardinal direction of aluminum square, we took 2 or 3 photos to cover the target area. Rough ice surface such as cryoconite holes need take additional photos close to nadir angle.**'

L143: Please specify the number of sampled areas.

**Reply: we have revised 'several' as '37'**

L147: 'Glacier surface albedo was calculated from measurements of up and downward

shortwave radiation.' => how were acquired the up and downard shortwave radiation? Please add details about this protocol and sensor.

Reply: we have revised it as' **After manual photogrammetry and removal of aluminum square, a Kipp & Zonen CMP11 albedometer consists of two identical pyranometers that measure the incoming global solar and the radiation reflected from the surface below. Albedo is the ratio of the two irradiances. Pyranometer is sensitive to radiation in the wavelength range 0.285 to 2.8μm. The albedometer was mounted on a camera tripod 1m above glacier surface and readings were made in a horizontal plane by means of a spirit bubble.'**

L160: 'For automatic photogrammetry': how were process the pictures? With a software? Did you use only one reference target? Did you use control points? What is the error of the final DEM? Please clarify.

**Reply: we have used Agisoft to acquire the 1mm resolution DEM data. For manual photogrammetry, the reference control frame have 4 control points and 4 check point (eight crossshaped screws on the aluminum frame). For automatic photogrammetry, we have 4 control points and 3 control points over the control frame. The error of the final DEM have provided in the revised manuscript.**

We have revised as' **For manual photogrammetry, we put the aluminum frame horizontally over the ice surface, the plot is detrended by setting 4control points and 4 check points at z axis of the same values. For automatic photogrammetry, the control field of wooden frame was also laid horizontally over the ice surface that lowered as the ice melted and maintained a horizontal position between the control field and ice surface. 4 control points and 3 check points are on the wooden frame was applied for georeference and precision check. Both manual and automatic photographs were imported into the Agisoft Photoscan Professional 1.4.0 application, which produced plot-surface point clouds and generated a detrended micro scale DEM of 1 mm resolution at plot scale. Based on the Agisoft PhotoScan processing report, automatic photogrammetry average geo-referencing errors fluctuated at around 1 millimeter. Total root mean square error (RMSE) of the automatic check points was $3.62 \pm 1.6$mm. Standard deviation of control and check point errors were all within 15mm. Manual measurements average geo-referencing errors fluctuated at around 1 millimeter, RMSE of 4 check points was $0.99 \pm 0.3$ mm, vertical accuracy was $0.66 \pm 0.3$mm. Standard deviation for x, y and z axis were all within 5mm'**

Figure 2: Please rename a) and b) because they have the same caption. + remove '[Figure 2]' in L 181 => please removed this typo for all figures.

**Reply: Thanks for your suggestions, we have revised and rename a) and b). [Figure 2] also removed in the revised manuscript.**

L185: repetition => 'Ice surface albedo is very sensitive to the LAIs concentration over ice surface.' Please add a sentence of transition, for example 'the surface roughness features impact the distribution of the snow impurities'.

**Reply: Thanks for your suggestions, we have revised based on your suggestion**

**We revised as 'The surface roughness features impact the distribution of the glacier**

**surface impurities**'

L190: by direct => by directly affecting

**Reply: revised**.

L 192: There is a problem with the unity of equation 2. epsilon is in cm while the h* is in meters?

**Reply: We have revised 'm' as 'cm' for the h*.**

L192 : equation 2 is the first equation => it is equation 1 here. Please correct the other equation notations in the text.

**Reply: we have revised accordingly.**

L 195: 'from Lettau (1969) developed aerodynamic surface roughness' => 'from Lettau (1969) who developed aerodynamic surface roughness...'

**Reply: Done**

L198: adopted => adapted

**Reply: Done**

L202: Please add a reference to introduce this metric.

**Reply:Done**

L205: This sentence needs to be rewrite: For rough ice surface means more surface area and less concentration of LAIs over ice surface, I don't understand.

**Reply: we have revised.**

L207: by consider => by considering

**Reply: Done**

L207: this is a new metric? If yes, please change 'defined' by 'introduced'

**Reply:   It is a new metric, we have revised 'defined' as ' introduced' accordingly.**

L220: 'For that reason, we used only half-hour albedo data taken when zenith angles were less than 60_' => Did you test the accuracy of the sensor considering the zenith angles? Is this a known angle of limitation? Or please add a reference.

**Reply: we analyzed the the Kipp & Zonen CNR4 10-min up and downward shortwave radiation data. When the zenith angles are larger than 60 degree, the 10 min data shows great fluctuation especially for snow covered surface. For bare ice, especially when it was smooth and dirty, the zenith angle have very limited effect over albedo. Other studies also suggest measurement under**

L222: from 11:00 to 17:00 o'clock => what range of zenith angle it is? It strongly depends to the date of your measurements + elevation of the studied site. Please clarify. => It seems that this section should be written in the protocol of albedo measurements, not here in this 'result' section.

**Reply: we have revised this part and add in the '2.2 data collection' section**

L225: field investigation => is it based on your automatic photogrammeric measurements? Please clarify.

**Reply: We have revised 'field investigation' as 'manual observation'. The sentence has revised as' On July 12, manual photogrammetry alone the main flow-line of the August-once ice capindicated that at the ice cap terminals of 4600m'**

L230: Surface roughness => replace it by 'the associated epsilon measurements decreased ...'

**Reply: Thanks for your suggestion, we have revised as' The associated ξ measurements decreased decreased from 5.49±1.5cm to 0.5±0.6cm as altitude increased'**

L236: 'Surface roughness fluctuated between 1.4_0.4cm to 3.3_1.1cm; An increasing trend of surface roughness could be detected as altitude increased (Figure 3e). ' => please, inverse these two sentences to be clearer.

**Reply: Thanks for your suggestion, we have revised accordingly.**

L245: 'surface roughness and LAIs decreased as altitude increased.' => not clear because it is only the case in July. In August 3: surface roughness increased as altitude increased and there was some snow cover at the top of the ice cap. Please clarify.

**Reply: Thanks for your suggestions. We have revised this part based on your suggestions.**

L247: 'There was a much higher concentration of LAIs on the uncovered ice surface than snow surface' => I don't understand the 'much', the trend is not so strong? It could help the understanding if your replot the Figure 3 with different symbols/colors associated to 1) measurements taken over snow cover 2) measurements taken over ice surface.

**Reply: Thanks for your suggestions, we have revised Figure 3 based on your suggestions.**

L248: 'As a consequence, albedo tended to be low on the ice surface and higher on snow-covered surfaces.' => be careful with this sentence, this is also mainly due to SSA values that strongly impact the albedo. There is a strong relationship between SSA/LAI/roughness/albedo. Please rewrite this sentence.

**Reply: we have revised accordingly.**

L251: repetition with L240-243 => this section needs to be re-structured in order to remove repetitions and to be clearer on conclusions.

**Reply: we have rewrite this part and make it more clearer.**

L260: 'The automatic measurement setup in the middle of the ice cap' => 'The automatic measurement setup was in the middle of the ice cap'

**Reply: Done.**

L.264: 'Intermittent snowfall decreased surface roughness and increased albedo' => you should add that it is because there is fresh snow, and fresh snow has high SSA, inducing a high albedo.

**Reply: we have revised based on your suggestion, Thanks.**

Figure 4: It could be clearer if you add the intermittent snowfall with vertical lines + air temperatures (to know when it is melting).

**Reply: Thanks for your excellent suggestion, we have revised Figure 4 based on your suggestion.**

L265: 'When snow began to melt'=> when is it ? How do you measure that? Do you have air temperature or snow surface temperature? If yes, you could add it on Figure 4 (see comment above).

**Reply: we have meteorological observations at top of the ice cap including precipitation, surface temperature and air temperature. We have revised Figure 4 and add air temperature accordingly.**

L276: 'When the snow melted, leaving patchy snow cover, surface roughness increased and albedo decreased quickly to around 0.3_0.05 within two day'. Here again the sentence needs to be rewrite: snow melting = lower SSA = lower albedo, the decrease of albedo is not only due to the increase of surface roughness and LAI. Please clarify.

**Reply: we have revised based on your suggestions.**

L276: 'from September 4 to 15, ice surface roughness fluctuated and increased to 2.63 cm on September 13.' => not clear, is it increasing from september 4 to 15, or for September 4 to 13?

**Reply: we have revised it.**

L277: 'Ice surface micro scale structures of cryoconite holes hidden LAIs from direct sun light.' => this sentence needs to be detailed to highlight the observations. => Reviewer suggest to recall this section (for example '3- Field observations'), since it details all the field observations, but not clearly enter in the physical analysis of the relationships between LAI/roughness/albedo. The analysis should include SSA variations. Even if no measurements were taken, it has to be mentioned (relating to the snow type of the area for example).

**Reply: Thanks for your suggestion, we have revised based on your suggestion**

Figure 5: Please rewrite the x axis 'Surface roughness (cm)' with the epsilon metric.

**Reply: we have revised accordingly.**

L301: Figure 2s => I am not sure this figure provides a lot, since it is well resumed in Figure 5a. In Figure 5a, the measurements taken manually and automatically should be separated by two different symbols to be clearer.

**Reply: thanks for your suggestion, we have revised it.**

L302: 'The combined manual and automatic scatter diagrams of Figure 5a display a significant negative power function between snow surface roughness and snow albedo (Figure 5a, r=0.82).' => this is a repetition.

**Reply: we have revised this part accordingly.**

L314: Figure 3s => same as above, the Figure 5b gives approximately the same information. Authors could put different symbols for manual/automatic measurements on Figure5b. Figure 4s: snow, patchy snow and bare ice surface => In the 'data and methods' section you should defined this terms. How do you classify an area as a 'patchy snow'? Is there a quantitative measurement of the proportion of snow in the area, or is it qualitative? How do you distinguish areas with very dense snow surface to ice surface? Please clarify.

**Reply: Thanks for your excellent questions. During filed observations, the albedo or roughness measurement is very sensitive to the snow covered and ice covered proportion on plot scale. In July of 2018, surface melt induced ice surface firstly appeared at terminal of glacier surface. The patchy snow cover formed a belt which shift upward in the melting season until all snow disappeared In August of 2018. A glacier surface is classified as patchy snow cover based on uncover of old dirty ice layers which induced very rough surface. In this study, we only have qualitative measurement based on the manual observations. In Figure 4c, we could tell it is patchy snow cover. We have add information in the 'Data and methods' section to clarify difference of snow, patchy snow and ice.**

L 324: 'Scatter plots of Figure 5c and 5d shows relationship between C LAIs and albedo' => Reviewer suggests to introduce a new section dedicated to LAI-albedo relationship, or to separate the section to put the analysis of the LAI-albedo relationship over snow surface in the 'snow surface section', and over ice surface in this 'ice surface melting season'.

**Reply: Thanks for your excellent suggestions, we have add snow surface LAI-albedo relationship to the 'snow surface in melting season' section. The ice surface LAI-albedo**

**relationship to the 'ice surface in melting season' section.**

L325: 'LAIs concentration is lower than patchy snow surface' => ' LAIs concentration is lower than the one measured over patchy snow surface'

**Reply: Done**

L327: Figure 4c => Figure 5c, and Figure 4d => Figure5d Please a particular attention should be taken on the notation of Figures in the text, this is confusing.

**Reply: We have revised accordingly.**

L329: Please introduce a new section, or reorganise this section into the 'snow surface section' and the 'ice surface section'.

**Reply: we have revised accordingly.**

L329: 'the effective LAIs concentration of C _' => 'the effective LAIs concentration (C _)' ' by considers roughness effect' => 'by considersing roughness effect'

Reply: **Done**.

L330: C _ is estimated based on equation (4). => this is a repetition

**Reply: Done.**

L334: 'It indicates that the Ceps concentration IS not affectED by surface roughness.' => english L335: 'Equation (3) is more appropriate to calculate LAIs concentration than equation (4) over snow surface.' => I don't understand. Do you mean 'to retrieve LAI concentration directly from albedo measurements'? Please clarify.

**Reply: Here we try to express the view that microcale snow surface features do not hidden snow surface LAIs like ice surface does. In the revised manuscript, we have revised it.**

L339: 'by consider surface roughness effect over not consider it in Figure 5d' => english L341: 'It means equation (4) estimate C _ could explain more ice surface albedo than equation (3) calculate C LAIs .' => please clarify. Do you mean: could explain ice surface albedo variations?

**Reply: Yes, we mean equation (4) could explain ice surface albedo variations than equation (3). In the revised manuscript, we have revised it to be clearer.**

 L352: English mistakes

**Reply: Done.**

L354: 'In accumulation season, except snow particles metamorphism process, constant blowing snow and intermittent snowfall was the main reasons which induced surface roughness fluctuation;' => More explanations are needed. In this Result section: the english has to be carefully corrected, some sections has to be restructured, the field observations and scatter plots have to be followed by physical analysis, according to the litterature, with more explanations. Figures need to be well called. Some changes in the graphs are suggested to be clearer.

**Reply: Thanks for your suggestions, we have revised this section accordingly.**

L380: 'The grain size is one of the most critical factors affect snow albedo' => 'The grain size is one of the most critical factors affect ING snow albedo.'

**Reply:Done**

L385: 'which is very similar with scatter plot of surface roughness and snow albedo provided in this study in Figure 5a and Figure 7.' => Please specify if this studies worked with the same snow type as you.

**Reply:We have revised it to wet snow type.**

L386: 'It means surface roughness as substitute of grain size is quite suitable for snow surface albedo explanation and parameterization at millimeter scale' => Please be careful with this conclusion, the study areas are not the same. To conclude this you should have both measurements (ssa and roughness) at the same site, at the exact same time (same sza) and over a dry and clean snow (no impurities, no liquid water content). Please clarify. Several contributions affect the albedo, and it is very difficult to separate each contribution to better understand/quantify their impacts on the albedo.

**Reply:Thanks for your excellent comments, we have revised it accordingly.**

L388: english L406: 'We expect that different mechanism of surface roughness evolution over melting season and accumulation season are the main reasons of different statistical relationship.' => This is not the main reason, the SSA variation also plays a strong role. Please clarify.

**Reply:We have revised this part accordingly.**

Section 7.2: The name of the section should be changed: authors speak about macro-scale surface roughness effects in this section, and in the section before it was at millimeter-scale.

**Reply: we have revised and not mentioned macro scale roughness effect in this part and only talks about micro scale surface features effects over albedo.**

L415: 'more incident radiation is absorbed by the slope facing the sun than by the slope facing away from it.' => this is not the only reason (see Warren et al 1998). The main reason is that photons are trapped between cavities, which increases the chance of a photon to be absorbed. Please clarify and detail the physical analysis.

**Reply:Thanks for your suggestions, we have revised it and add photons trapped effect for rough surface.**

L430: filed investigation => english L430-433: Please rewrite the sentence, I do not understand

**Reply: we have revised accordingly, the revised part was' Additionally, field investigations over mainly flow line across the August-one ice cap find protrusions on the ice surface visually have less LAIs over depressions. The LAIs concentrated in depressions rather than evenly distributed over rough ice surface reduced its absorption of solar radiation effect (Figure 8e).'**

L441: 'Rough surface means lower concentration of LAIs and high albedo over smooth ice surface with heavy loading of LAIs and low ice surface albedo.' => this is confusing to speak about rough surface of a smooth ice surface.. how do you define a smooth ice surface? What is the scale of roughness features that you are talking about? Please clarify.

**Reply: we have revised and clarify that we are talking about microscale surface features.**

=> this is very confusing between Sections 7.1 and 7.2 => when are you talking about ice surface and snow surface? For example: the first section of the section 7.2 is talking about snow surface (see the litterature), and not the ice surface. Please clarify.

**Reply: we have revised and add snow surface to the Section 5.1, and ice surface to the Section 5.2.**

L447: 'Most of the established and widely used snow surface albedo parameterization ARE either based on snowpack age, snow depth, snow density, air temperature' => english => Please add the SSA and SZA (see the litterature of Kokhanovsky) => This sentence should be

put in the section 'snow surface', not 'ice surface'.

**Reply: we have revised and removed snow part to the 'snow surface' section.**

L450. Please add a reference

**Reply: we have revised accordingly.**

L451: 'We expect the new parameterization methods provides' => english 'are more physically based than some of the studies presented.' => What new parametrization? Please refer to the equation here. (as equation 5 but for ice surface) This sentence needs to be rewritten:this is not a physically based parametrization, it is empirical. Indeed your equation 5 strongly depends of your study site and your measurements.

**Reply: Thanks for your suggestions, we have rewritten this park and make it more clear than before.**

L466: 'For snow covered surface, the coefficient increases quickly from -0.67 to - 0.74 when 1m plot resolution increases from 333.3 mm to 200.0 mm' => but the Figure 8a shows a very flat and smooth snow surfaceĂ˘a? Please clarify the studied surface.

**Reply: we have clarify in the revised manuscript, the snow covered surface actually include flat surface and rough partially snow covered surface. The statistical between surface roughness and albedo include partially snow cover and flat snow surface. It could be the reason the snow surface roughness and albedo still sensitive at around 100mm resolutions.**

L485: 'This could be a practicable way to parameterize surface roughness and albedo on a whole-glacier scale' => Please add some details. It could be interesting to apply the method proposed by Smith et al 2016 here, to estimate a temporal mean albedo at the whole-glacier scale.

**Reply: we have revised accordingly.**

Section '7.3 Glacier surface albedo parameterization at whole-glacier scales based on surface roughness' => the title needs to be changed since you investigate the sensitivity of the albedo relationships at a plot-scale, but not at the whole-glacier scale in this section. => Moreover, please rely your analysis with the litterature (Irvine-Fynn et al 2014; Smith et al 2016 ..) to clarify what is the novelty of what you propose. => It could be interesting to estimate the LAI coverage with your photogrammetric acquisitions as Takeuchi et al 2018.

**Reply: we have revised the title as 'Glacier surface albedo parameterization based on surface roughness'.**

**In the revised manuscript, we talk about micro scale surface roughness effect and macro scale surface roughness effect. We talked about this manuscript's merit and deficiencies of the article.**

L497: 'we have a general understanding of the surface roughness that controls the albedo of snow and ice surface are quite different.'

**Reply: we have rewrite this part accordingly.**

=> english L502: 'necessity to consider'

=> english L504: 'Surface roughness seems play a quite'

=> english L535: 'field data of LAIs, surface roughness and albedo need to collect to help us to present more detailed analysis and modeling research about surface roughness and LAIs at micro scale over ice or snow surface energy and mass balance process.'

=> SSA measurements should be acquired too.

**Reply: Thanks for your valuable advices, we have revised and revised accordingly.**

---

## Author Comment (AC2) · 1 Aug 2020

We'd like to than the referee of Alvaro Ayala for the valuable, constructive and detailed comments which greatly helped to improve the manuscript. The corresponding changes and refinements will made in the revised paper with track changes. We are intend add some new observations in melting season in 2020. New observation data has include surface roughness and snow grain size. This could give some robust evidence of snow metamorphism process connect with snow surface roughness change. So the revised paper might take another 1-2 month to add our new data and results. Reviewer comments in normal font, our reply to each comment is provide after

the comment and given in bold font.

Please also note the supplement to this comment:
https://tc.copernicus.org/preprints/tc-2020-67/tc-2020-67-AC2-supplement.pdf
* * *
[Figure]

**Supplement:**

We'd like to than the referee of Alvaro Ayala for the valuable, constructive and detailed comments which greatly helped to improve the manuscript. The corresponding changes and refinements will made in the revised paper with track changes. We are intend add some new observations in melting season in 2020. New observation data has include surface roughness and snow grain size. This could give some robust evidence of snow metamorphism process connect with snow surface roughness change. So the revised paper might take another 1-2 month to add our new data and results. Reviewer comments in normal font, our reply to each comment is provide after the comment and given in bold font.

Reply to comments from referee 2

Effects of surface roughness and light-absorbing impurities on glacier surface albedo, August-one ice cap, Qilian Mountains, China. Authors: Liu, Chen, Ding, Han, Yang, Liu, Wang, Guo, Song, Qing

PAPER SUMMARY AND RECOMMENDATION

Liu et al. present a dataset consisting of field measurements of albedo, surface rough- ness and surface concentration of Light-Absorbing Impurities (LAIs) collected during a

melt season on the August-one ice cap in the Qilian Mountains, China. The measure- ments were collected from July to October 2018, spanning most of that melt season (July to September) and one month (October) of the next accumulation season. The collected data are used to propose statistical relations between the three main mon- itored variables (albedo, surface roughness and surface concentration of LAIs), and to discuss the physical relations between them, among other

analyses. In particular, they propose that there is a combined effect of surface roughness and LAIs surface concentration on the albedo of snow and ice.

I think that the topic of this investigation is relevant for the scientific community and The Cryosphere. Although some additional details about the sampling methodology are needed, the data seem well collected, and the physical processes that are proposed to explain the observations are interesting and reasonable. Unfortunately, I don't think that the authors convincingly prove their hypothesis about the combined influence of surface roughness and LAIs, because i) they tend to ignore or minimize the role of snow metamorphism on snow albedo, and ii) the impact of snow patches on a bare ice surface. Please see my main point below for details in this regard. Importantly, the structure of the article is confusing, and the text and figures can be condensed and better organized. Finally, the use of English language should be revised, as it sometimes obscure the ideas of the authors. My recommendation is to reconsider this article after major revisions.

GENERAL COMMENTS

Physical connections between surface roughness, LAIs and albedo Process understanding

My main comment is that the authors tend to overestimate the influence of surface roughness on albedo in their analyses and conclusions. I have no doubt that there is a statistical relation between these two variables (as shown in Figure 5), and also some physical connection (Larue et al., 2020), but the collected data does not necessarily suggest a causal relation. As the authors do in the paper, I divide my arguments for the periods of snow melting and bare ice melting.

Period of bare ice exposure: Figures 2b and 2c present a good summary of the main idea discussed by the authors. A rough ice surface (with a surface roughness of about 1-10 cm) hides some LAIs, raising the area-average albedo relative to surfaces with uniform debris cover (Bøggild et al., 2010). If the surface smooths over the melting season, the LAIs emerge and the albedo decreases. The authors parameterized this effect using equation (4). However, based on the data shown by the authors, I would say the positive relation between surface roughness and albedo over an ice surface (Figure 5b) can be solely explained by the high-albedo snow patches (shown in Figure 4f) resulting from the snowfall events discernible from the occasional albedo increases during August (Figure 4a). In that respect, can the authors add a figure with the meteorological variables measured at the AWS shown in Figure 1? I think that in order to demonstrate that the albedo increases are only due to changes in surface rough- ness, the authors would need to show data without snow patches. Otherwise, it seems very plausible that the high-albedo snow patches resulting from the occasional summer snowfall events

are responsible for the positive relation between surface roughness and albedo, during the ice melt period.

**Reply : Thank you for your comments and suggestions. In the revised manuscript, we will exclude patchy snow surface in bare ice surface analysis. In the revised manuscript, we also add precipitation data and half-hour air temperature observations which also analysis snow metamorphism.**

Period of snow melting. It is well known that snow albedo decreases due to snow metamorphism and large grain sizes, which affects the light scattering. The increasing grain size can have an effect on the surface roughness, but the latter is also explained by differential melting of the surface. From lines 505-507, the authors seems to suggest that the lowering of the albedo is caused only by the increasing surface roughness.

**Reply : Thanks for your comments. We neglect the snow metamorphism process in snow albedo and roughness analysis in the manuscript since we lack snow grain size observations in 2018. In the revised manuscript, we have revised and add snow metamorphism in melting season and accumulation season and connect surface roughness with snow grain size process with new data collected in summer of 2020.**

"Physically-based" parameterizations and their potential for large-scale albedo estimates

The statistical relations between albedo, surface roughness and concentration of LASs presented by the authors are described as "physically-based". Although it is interesting

that the authors include these variables in an albedo parameterization, these equations are not "physically-based", because they are not derived from a fundamental principle or process (e.g. energy conservation) and their parameters (obtained from a scatter plot) cannot be directly measured in the field.

**Reply: Thanks for your comments. We revised 'physically-based' as 'surrogate variables'. In the revised manuscript, we add more physically based analyzed of snow metamorphism, snow surface roughness and albedo.**

In the abstract and conclusions, the authors suggest that albedo can be retrieved from surface roughness at large scales. However, this seems unrealistic because the nec- essary high-resolution DEMs to derive surface roughness can only be obtained by sen- sors carried by Unmanned Aerial Vehicles (UAVs), which are only used at the glacier- scale. Are the

authors referring to that scale? Using satellite products, albedo is much easier to retrieve than surface roughness at regional scales.

**Reply: Thanks for your comments.High-resolution DEM data can be acquired by unmanned aerial vehicle photogrammetry or terrestrial laser scanning. Right now these methods can only applied at glacier scale. We have revised the large scale as glacier scale.**

LAIs sampling

More details are needed regarding the LAIs sampling. How did you calculate the vol- ume of each sample? Is that the volume of the stainless steel spoon mentioned in line 150? How did you make sure that the volume was the same in every collected sample? Can you include a picture of the spoon? How did you sampled sites with a rough surface, such as one with cryoconites? Did you take the samples at the top or the bottom?

**Response: We revised and add more details about sampling of snow and ice surface roughness.**

**The sample volume of snow or ice is not same. The sample volume depend on surface roughness and LAIs concentration. For example, if the surface is smooth and clean snow, we sampled 3 cm depth with an area of 20*20cm. If the snow is thin and underneath is dirty ice. The collected snow sample and the dirty ice layers. For the smooth ice surface, we collect surface 2-3cm ice. Most of the LAIs basically concentrated at surface and the underneath ice is looks clean. For rough ice surface, the sample depth depend on the surface roughness. The cryoconite holes usually 6-7cm deep. The collect sample could be 9 cm. Bellow this depth, the ice is looks clean.**

**In this study, we used a 20*20cm*10cm nonrust steel box without cover and bottom to sample snow or ice. The sample procedures are: we put the steel box over sample area.   Excavate the snow or ice surrounding the steel box to the depth we needed and keep the ice or snow in the steel box intact. After then, we lift the steel box, and sampled the snow or ice sample by a flat bottom shovel. Most of the time, ice is very hard, we use chisel to chip the ice.**

[Figure]

[Figure]

[Figure]

**Figure 1 sample procedure in field.**

Paper structure and English language

Please improve the paper structure. I have the following suggestions: - Section 2.1 is neither data nor methods. Please add a new section called "Study Site". - Sections 3 and 4 consist of results. I suggest including these sections in a traditional "Results" section. - Sections 3 and 4 have a confusing structure, partly because the titles are ambiguous and repetitive. - The numbering of sections 7 and 8 is wrong. They should be 5 and 6.

**Response: Thanks for you excellent suggestions, we ill revised based on your suggestions.**

I have provided many suggestions to improve the use of English language, but I think that the authors should perform a full revision of the text.

**Response: Thanks for your suggestion, we will revised based on your suggestions and comments and referee 1's suggestions.**

SPECIFIC COMMENTS

11: "Fluctuations in surface albedo are due primarily to variations in micro scale surface roughness ($\xi$) and light-absorbing impurities (LAIs) in this region." I guess you mean fluctuations of bare ice albedo, because snow albedo variations are very large and are due to metamorphism. See my main comment 1a.

**Reply: Thanks for your suggestions and comments, we have revised it accordingly.**

21: "ξ could explain 68%of snow surface albedo and 38%of ice surface albedo variation in melt season." When you write that surface roughness explains 68% of the snow albedo variation, I think that you need to explicitly state that this is a statistical analysis, because the physical explanation for snow albedo variations is snow metamorphism and an increasing grain size. See my main comment 1.

**Reply: Thanks for your suggestions and comments, we have revised it based on your suggestions.**

37: "According to Hock (2005), on average it accounts for over 70% of the net energy input to glacier surfaces." You should mention that that number was obtained for a particular glacier (Storglaciären). In any case, what do you mean exactly by "net energy input"? Net = input - output. Table 1 in Hock (2005) shows that incoming longwave radiation (L_in) is the largest energy input. In general, L_in and S_net are the largest energy inputs to glaciers at daily, and longer, time scales (Ohmura, 2001).

**Reply: Thanks for your comments, we have revised it accordingly.**

78-79: "poorly investigated, and snow surface albedo parameterization methods based on surface roughness are rarely reported." However, the relation between snow surface albedo and grain size has been largely analyzed. Please see my main point 1a for snow surfaces.

**Reply: Thanks for your comments, we have revised this part.**

86: "Surface roughness structures developed during melt season such as crevasses, cyroconite holes, can increase ice surface albedo by hiding LAIs from direct sunlight have been widely reported." Although, more than "increasing" surface albedo, I would say that they "prevent", or at least "delay", a further decrease, because they hide LAis that would otherwise reduce the area-average albedo.

**Reply: Thanks for your excellent suggestions, we have revised based on you suggestions.**

229: How do you calculate the average value and uncertainty of albedo, surface roughness and LAIs concentration at a particular elevation?

**Reply: we have measured up and downward shortwave radiation 3 to 6 times (measured more times when weather**

**conditions is cloudy or changed from sunny to cloudy). The data is quite variable for snow surface under cloudy weather. But it is not sensitive to weather for dirty ice surface.**

242: "LAIs decreased from 0.04±0.03g/ cm3 at middle part to 0.003±0.002g/ cm3 at higher elevations (Figure 3i)." This is based only in one sample at 4700 m, if you delete it, you don't have any trend. Did you try any other topographic parameter?

**Reply: we measured elevations, latitude and longitude. Slopes and aspect are not measured. Basically, the ice cap is flat. Aspect and slope changed not so great along the main flow line.**

247-249: "There was a much higher concentration of LAIs on the uncovered ice surface than snow surface. As a consequence, albedo tended to be low on the ice surface and higher on snow-covered surfaces." This is not only explained by the LAIs. Ice albedo is usually lower than snow albedo.

**Reply: Thanks for your suggestions, we revised accordingly.**

378-379: "Although we do not have tandem surface roughness and grain size observa- tions, the evolution of surface roughness calculated at 1mm resolution in snow covered period should quite similar with fluctuation trend of grain size evolution." Certainly, you have a correlation between grain size and surface roughness, but surface roughness is not only explained by grain size. Differential melting of the surface could be also important.

**Reply:   Thanks for your excellent suggestions, we have revised and add the differential melting here accordingly.**

423-424: ". . .a significant positive relationship rather than a negative relationship was established over ice surface based on manual and automatic measurements. We expect it is related with abundant LAIs over ice surface at the August one ice cap." This might be explained by the snow patches. See my main comment 1.

**Reply: Thanks for your suggestion. In the revised manuscript, we analyzed the pure ice surface without inclusion of patchy snow surface. More detailed results will provided in the revised manuscript. We also want include the comparative measurement of snow grain size and surface roughness data acquired in July of 2020.**

448-450: "The performances of the establishes albedo methods either based on sur- face roughness, LAIs or effective LAIs

concentration shows a great improvement over the assumption of a constant mean ice albedo or surrogate variables, such as air tem- perature, accumulated melt and elevation." I don't agree that surface roughness is much better than air temperature or melt to parameterize albedo. Please see main comment 1b.

**Reply: We will take it more carefully. In here, we try to say surface roughness and LAIs as surrogate variables show improvement over ice surface. In the revised manuscript, we have add new references which talk about ice surface albedo parameterizations such as Brock (2000) or Jonsell et al. ( 2003). The upmost ice surface characteristics of this layer are mainly responsible for the observed variability in ice albedo at any given site. The surface roughness and LAIs are surface properties which might more connect to surface albedo than air temperatures or melt amount.**

462-464: "Since surface roughness is dependent and sensitive to topography data resolution (Figure 8a), so which resolution is appropriate for snow and ice surface albedo estimation?" Was this part of the objectives? Are you writing about the DEM resolution? Please be more explicit in this section and present this analysis earlier in the text.

**Reply: Thanks for your suggestions, we have revised and introduced in introductions. We also revised here accordingly.**

496-498: "By using manual and automatic photogrammetric measurements of sur- face roughness, manual LAIs samples and measurement of broadband albedo at the August-one ice cap, we have a general understanding of the surface roughness that controls the albedo of snow and ice surface are quite different." Can you really conclude that surface roughness controls the albedo of snow and ice? See my main comment 1.

**Reply: Thanks for your suggestion, we have revised as "surface roughness is a good surrogate variables can be used to parameterize surface albedo ". More detailed will proved in revised manuscript.**

505-507: "For snow-covered surfaces, ice particle metamorphism and surface melting and refreezing induced grain size increasing synchronously happened surface rough- ness increasing, which induced decreasing snow surface albedo in melting season." The last part of the sentence suggests that is the increasing surface roughness that induces the snow albedo decrease.

**Reply: Thanks for your suggestion, we make mistake here. we have revised it. Snow metamorphism which could induced snow grain size increase, snow surface roughness increase, and albedo decrease. Snow surface roughness increase is not induces the snow albedo decrease.**

SUGGESTED TECHNICAL CORRECTIONS

4: Please make the format for last names uniform (sometimes is uppercase and some- times not).

**Reply: Thanks for your suggestions, we have revised and revised and uniform the names.**

11: influence on melt

**Reply: we have revised it.**

16: the present study consisted of an intensive. . .

**Reply: we have revised based on your suggestions.**

19: the middle

**Reply: we have revised it, thanks.**

19: A detailed analysis indicates that. . .

**Reply: we have revised it accordingly.**

20: a positive linear

**Reply: revised.**

21: for snow and ice, respectively

**Reply: revised.**

22: consider -> considering

**Reply: we have revised accordingly.**

25: constant mean -> uniform

**Reply: we have revised as 'uniform' in the revised manuscript.**

33: The energy balance and resulting melt rates. . .

**Reply: we have revised it accordingly.**

33: the meteorological conditions

**Reply: we have add ' the ' in the revised manuscript.**

34: the physical properties

**Reply: we have add ' the ' in the revised manuscript.**

34: Delete ", which determine glacier melt process"

**Reply: Thanks for your suggestions, we have revised based on your suggestions.**

35: the melting of glacier -> glacier melt

**Reply: Revised.**

35: dominated-> controlled

**Reply: Revised.**

35: in the glacier surface -> at the. . .

**Reply: we have revised.**

36: Shortwave radiation is the main energy input causing snow and ice melt

**Reply: we have revised it based on your suggestions, Thank you.**

39: net radiation: do you mean net shortwave radiation?

**Reply: we have revised as net shortwave radiation.**

45: "Glacier albedo varies much more dramatically than other land covers" -> Albedo varies much more dramatically on glacier surfaces than on other land covers.

**Reply: Thanks for your suggestions. We have revised it based on your suggestions.**

47: "constantly changing surface characteristics"-> I think that you should explicitly mention snow metamorphism here.

**Reply: We have revised based on your suggestions.**

47: the solar incidence angle

**Reply: we have revised It accordingly**

52: accelerate

**Reply: Revised.**

57: Simulations

**Reply: Revised.**

59-62: In general, I suggest improving the writing style by making sentences shorter and more fluid. Here, I would suggest something like: LAIs have decreased the surface albedo of glaciers in Qilian mountains to values as low as 0.13+-0.06 during the melting season (LIT).

**Reply: Thanks for your suggestions, we have revised it based on your suggestions.**

59: Please introduce the Qilian mountains before this sentence.

**Reply: Thanks for your suggestions, we have introduced the Qilian mountains in here accordingly.**

64: I think that you should start a new paragraph when you mention surface roughness.

**Reply: Thanks for your suggestions. We have start a new paragraph to introduce surface roughness.**

70: do you mean "first recognized by Kuhn"?

**Reply: Yes, here, we have revised it based on your suggestions and make more clear.**

71-72: In which way these field campaigns advanced the influence of roughness on albedo?

**Reply: Thanks for your question. In the revised manuscript, we will make more clear.**

75: Studies. Which ones?

**Reply:   Studies by Roujean et al. (1992),   O'Rawe (1991), and Larue et al. (2019). There we have revised and accordingly.**

75: indicated

**Reply: Revised.**

75: the inclusion

**Reply: revised.**

76: measurements of albedo? Sorry, this is not clear.

**Reply: Thanks for your question, we will revised it more clear.**

76: equations

**Reply: revised.**

77: and sastrugi

**Reply:revised.**

80: heterogeneous: Please explain better what do you mean by this.

**Reply: here we try to express the uneven distribution of LAIs induced differential absorbing of shortwave and melting differences**. In the revised manuscript, we revised clearer.

80: of LAIs.

**Reply: revised.**

83-85: Please improve the structure of this key sentence.

**Reply: Thanks for your suggestions, we will revised clearer.**

85: the melt season,

**Reply: revised.**

85: crevasses and

**Reply: revised.**

86: the direct sunlight

**Reply: revised.**

88: Please introduce the ice cap more formally. Where is it? You can give the details in the next section, but give at least

some indication here.

**Reply: Thanks for your suggestion, we introduce the ice cap and add the location here**

90: indicated

**Reply: revised.**

93: LAIs on ice

**Reply: revised.**

95: combined effects: This is very interesting, and difficult, how do you isolate both

effects? Please see my main comment 1.

**Reply: we are not trying to isolate surface roughness effect and LAIs effect in this study. Studies by Larue et al.(2020) have quantify the impact of surface roughness on albedo on clean snow surface. In this study, we find surface roughness affect ice surface albedo by hide LAIs from direct sunlight. For snow surface, the surface roughness do not affect LAIs concentration. It is very difficulty try to differentiate surface roughness and LAIs effect especially in summer, since other factors such as water content, solar incidence angle, snow grain size, or snow specific area, snow metamorphism affect snow surface albedo.**

96: have been rarely

**Reply: revised.**

92: we investigate

**Reply: revised.**

93: better understand and simulate. I think that these are very wide and general objec- tives. Can you be more specific?

**Reply: thanks for your comments, we will revised based on general comments and make it more specific in the revised manuscript.**

106: at the daily scale

**Reply: revised.**

101: we investigate

**Reply: revised.**

104: on the middle

**Reply: revised.**

104: in melting season and accumulation season of 2018 -> in the melting and accu- mulation season of 2018.

**Reply:revised**

107-108: This is not clear at this point.

**Reply: we have revised and make it clearer.**

114: Study area is neither data nor methods. Create another section.

**Reply: Thanks for your suggestion, we have revised it.**

116: the Qilian Mountains

**Reply: revised**

120: May to September. Mention that this is summer.

**Reply: revised**

120: How short is summer?

**Reply: revised**

126: has been observed.

133: This section should be called data collection and image processing.

**Reply: Thanks, we have revised accordingly.**

137: In some parts of the text you write "we" and in others you write "researchers". Please make it uniform. I would suggest to use an active voice, i.e. "we".

**Reply: Thanks, we have revised as 'we' in the revised manuscript.**

142: several different locations. How many? Make a reference to Figure 1. You don't have any table in your article. It would be good to have one with the coordinates, number of measurements, etc.

**Reply: we have revised and add the number of measurements in the revised manuscript.**

142: the top of the ice cap

**Reply: revised.**

144: the micro scale topography over different altitudes

**Reply: revised**

145: Physical->You mean topographical in opposition to the turbulent heat fluxes pa- rameter (surface roughness length)?

**Reply: Here, we have revised as surface roughness. It is not surface roughness length.**

145: was estimated. How?

**Reply: The surface roughness is estimated based on equation (2). In the revised manuscript, we have revised and make it clearer.**

146: putting->placing a Kipp and Zonen CMP11 radiation sensor

**Reply: revised.**

152: in an oven

**Reply: revised.**

156: its operation

**Reply: revised.**

157: geo-reference: what do you mean?

**Reply: we have used a wooden control field with 8 control points on it as control frame to geo-referencing pictures to DEM data.**

159-160: "The photography was repeated at three hour intervals from 9:00 AM to 18:00 AM, UTC+7 time." But later you work with daily means, don't you?

**Reply: we select one of the four sets of photos and merged to produce a 1mm×1mm resolution surface topography. Not all four sets was used to produce surface DEM data, since some of the data sets was affect by fog or too dark to be applied for DEM data.**

159: how many different locations?

**Reply: we have revised and add the number of 37 locations in the revised manuscript. In the revised manuscript we will include new observations at 20 locations carried out in 2020 over snow surfaces.**

168: How did you make sure that the frame stayed horizontally?

**Reply: we put the frame over ice or snow surface. The frame is horizontally with ice or snow surface. It make the generated DEM detrended.**

185-189: Repetition

**Reply: revised.**

190: that surface roughness

**Reply: revised**

190: by directly affecting

**Reply: revised**

195: from the aerodynamic surface roughness parameter developed by Lettau (1969)

**Reply: revised**

196: except that we changed the silhouette area facing upwind to. . .

**Reply: revised**

203: Is V the volume of the spoon?

**Reply: Yes, V is calculated based on the spoon, the spoon is a 20\*20cm\*10cm box without cover and bottom. The sample area is 20\*20cm, the depth is measured ever time base on the insert depth.**

210: We assumed that

**Reply: revised**

220-224: This sounds like methods to me. Is this the Results section? See my minor

comments.

**Reply: we have revised it and add to the methods part.**

225: You also have temporal variability here, not only spatial. Please change the title.

**Reply: revised.**

226-227: Again, I suggest shorter and more fluid sentences, something like:

We found a patchy snow cover with many cryoconite holes on the glacier terminus (~4600 m) at the date of the first field trip (July 12).

**Reply: thanks for your suggestion, we have revised accordingly.**

230: Be consistent with the digits, i.e. 5.49 cm = 5.5 cm.

**Reply: revised**

244-245: On July 12, surface. . .

**Reply: revised.**

246-247: the transient snowline retreated up-glacier.

**Reply: revised**

247-248: On what plot do you base that sentence

**Reply: We have revised and differentiate snow and ice covered surface in the revised manuscript in Figure 3. In the revised manuscript, we could find snow covered surface LAIs concentration is much lower than ice surface.**

**248: bare ice surface than on snow.**

**Reply: revised**

249-250: Albedo didn't really increased over time.

**Reply: we try to express the increasing trend of albedo from terminal to top. The albedo is not increase over time. We will revised it and make clearer.**

251: minimal albedo: Do you mean the season minimum?

**Reply: The minimal is not precise, we have revised. The albedo in August is very low when the ice cap if all bare ice from terminal to top**

251: "but it also did not increase"-> and it did not show a clear trend with altitude

**Reply: revised.**

259: This is almost the same title as the previous sub-section.

**Reply: we have revised as Automatic observation of surface roughness and albedo**

262: October 17

**Reply: revised.**

270: mostly bare ice with occasional snowfall events

**Reply: revised**

285: was mainly induced

**Reply: revised**

259: The title of this section is very similar to that of section 3.

**Reply: we have revised and differentiate with section 3.**

298-300: Snow surface albedo and the corresponding surface roughness are analyzed using. . .

**Reply: revised.**

301: functions

**Reply: revised**

301: observations

**Reply: revised**

304: Please use the same coefficient in the text and the figure (either r2 or r).

**Reply: revised**

321: What is a patchy ice cover? A bare ice surface covered by snow patches or by debris patches?

**Reply: we have revised as snow patches**

322-323: This is not clearly written. Better say that the patchy snow cover is the limit between the two periods that you identified.

**Reply: Thanks for your advances, we have revised accordingly.**

324: the relationship

**Reply: revised**

324-325: Delete "without consider surface roughness effect over LAIs concentrations."

**Reply: revised.**

328: Do you mean 5d?

**Reply: we make mistakes here, we have revised.**

328: manually

**Reply: revised.**

331: power function, do you mean linear function?

**Reply: revised.**

333: that considering

**Reply: revised**

334: . . .albedo during the snow period.

**Reply: revised**

334: It indicates

**Reply: revised**

365-373: All these lines can be removed, moved to the introduction, or condensed.

**Reply: Thanks for your suggestion, we have revised accordingly.**

369: Meltwater

**Reply: revised**

378: should be quite similar

**Reply: revised**

380-383: This should be mentioned earlier in the text.

**Reply: revised.**

386: it means that

**Reply: revised**

388: LAIs are another critical factor affecting snow albedo

**Reply: revised.**

392: "increasing surface roughness decrease albedo" But, only in your snow melting

Period

**Reply: revised.**

396-400: This should be in results.

**Reply: revised**

400: What is the explained variance of each variable?

**Reply: we have add the explained variance of surface roughness and LAIs in the revised manuscript.**

402: During the accumulation season

**Reply: revised.**

403: Please explain better the observed snow metamorphism in the accumulation pe- riod

**Reply: Thanks for your comments, we have revised and add more citetations about snow metamorphism in the accumulation period in the revised manuscript.**

403: During the accumulation period

**Reply: revised.**

413-421: This belongs to the introduction

**Reply: Thanks for your suggestion, we have revised accordingly**

430-433: Please re-write this sentence in a clearer way.

**Reply: Revised, we have revised and re-write it clearer.**

437: over smooth ice -> than smooth ice?

**Reply: we have revised 'over' as 'than' .**

447: are based

**Reply: revised.**

TABLES

There are no tables in the article. At least one summarizing the distributed measure- ments would be useful.

**Reply: Thanks for your suggestion, we will add tables introduce automatic observations and manual measurements involved in this studies.**

FIGURES

Figure 1: Please add an inset showing the ice cap and the Qilian Mountains in the Tibetan Plateau.

**Reply: Thanks for your suggestions, we will revised based on your suggestions, and add the cap cap location and the Qilian Mountains**

Figure 2: Why is important to have rain or fair weather? Provide a brief explanation.

**Reply: Thanks for your suggestions, we find rough ice surface developed in cold and sunny day, and warm and cloudy or rain day favors smooth and dark ice surface. We will explain it in the revised manuscript.**

Figure 3: I would place altitude in the x-axis. Elevation should not be a dependent variable.

**Reply: Thanks for your suggestion, we will revised based on your suggestions, we also differentiate snow and ice surface in the revised manuscript based on referee 1's suggestions.**

Figure 3: In the text of this figure, you write several times that there is a relation with the altitude, but based on your data, this is true only for 3a and 3b.

**Reply: The relation with altitude is established based statistical significance.**

Figure 4: Make a uniform tick spacing in the x-axis. Example, 1-07 15-07, etc.

**Reply: Thanks, we will revised based on your suggestion.**

Figure 4: From this plot is evident that there were 3 or 4 snowfall events in August. See my main comment 1.

**Reply: Yes, there are snowfall events in August, we have add these short period snow surface roughness and albedo as snow covered surface rather than bare ice cases. During this snowfall period, the snow melt quickly, and snow surface roughness also changes fast.**

Figure 4: Add a line in panel a to show when were the pictures taken.

**Reply: Thanks for your suggestions, we have add the panel date in the revised manuscript.**

Figure 4: Show the dates of the pictures in the pictures.

**Reply: revised**

Figure 5a: Note that for low albedo (<0.4) you can get very different surface roughness (1-6 cm).

**Reply: Low albedo for snow surface are all patchy snow covered surface. Patchy snow shows great surface roughness differences. The thick snow patches usually shows larger surface roughness, and thin snow patches shows smaller surface roughness.**

Figure 5: Consider the use of different markers (in a-b and c-d) to reduce the number of panels. As in Figure 4s.

**Reply: Thanks for your advices, we will revised based on your suggestions.**

Figure 5: I would strongly suggest using only one paragraph to describe one figure. This is a big help for readers.

**Reply: Thanks for your suggestion, we will adopt your advice.**

Figure 5: How many different sites are included in the manual measurements shown in this figure?

**Reply: we have 37 sites of surface roughness observations. In the revised manuscript, we have add these information in the revised manuscript.**

Figure 6: Can you merge both plots in one panel by using different markers? A loga- rithmic scale might be useful.

**Reply: Thanks for your suggestions, in the revised manuscript, we have revised and talk snow and ice surface separately in different figures. It will be clearer.**

Figure 7: Apart from the high albedo for low surface roughness, there is a poor relation between the two variables. Maybe move to the supplementary?

**Reply: In the revised manuscript, we have discussed snow metamorphism in cold season. I think it is an interesting topic which could be a complementary for melting season.**

Figure 8: What is the difference between these pictures and those shown in Figure 1s?

**Reply: In Figure 8, we calculated surface roughness under different resolutions. Figure 8 shows rough surface are more sensitive to resolution than smooth surface. It means ice surface are more sensitive to snow surface.**

**Figure 1s calculated surface roughness at 1mm resolution for all manual and automatic observations.**

**In Figure 8, and figure 9, we try to find under what resolution, the calculated surface roughness have more significant statistical significance with albedo. We find finer resolution of than 50 mm and 100 mm resolution is recommended for ice and snow surface roughness calculations.**

Figure 2sa-3sa: How many different sites did you use to build this plot?

**Reply: we have 37 manual observation sites. The automatic observation in melt season include 65 days. We applied 63 plots of automatic observations. We have revised and include these information in the revised manuscript.**

Figure 4s: This is a very nice figure, because it shows both periods in the same plot. Can you move this figure, or a similar one, to the main text?

**Reply: Thanks for your suggestion, we will revised accordingly.**

---

## Author Comment (AC3) · 30 Aug 2020

Dear Editor and referee I would like to thanks editor and referees great and detailed comments. These comments are greatly help me to improve the manuscript. We have given some carful explanations in our replyïijŻplease see the detailed point-by-point responses below. The corresponding changes have been made in the revised paper, track changes was used in order to be easily identified. Marked-up manuscript was given at the end of the replies. We hope the revised manuscript is suitable for the journal.

[Figure]

Please also note the supplement to this comment:
https://tc.copernicus.org/preprints/tc-2020-67/tc-2020-67-AC3-supplement.pdf
* * *
**Fig. 1.**

Legend:
- LAIs samples
- Weather station
- Automatic photogrammetric camera
- Automatic albedometer observation
- Manual albedo observation
- Manual photogrammetric observation plot

a: rough ice surface

Weathering crust

Sunny

Cold, dry and windy

depressions

Cryoconite hole

Light-absorbing impurities

Ice cap ice

b: smooth ice surface

Rainfall

Cloudy

Warm, humid and windy

Ice cap ice

Light-absorbing impurities

**Fig. 2.**

Fig. 3.

[Figure]

**Fig. 4.**

[Figure]

**Fig. 5.**

[Figure]

[Figure]

**Fig. 6.**

[Figure]

**Fig. 7.**

[Figure]

$$y = 0.7091x^{-0.093}$$
$$R = 0.50$$

**Fig. 8.**

[Figure]

**Fig. 9.**

[Figure]

Fig. 10.

**Fig. 11.**

**Supplement:**

**Effects of surface roughness and light-absorbing impurities on glacier surface albedo, August-one ice cap, Qilian Mountains, China**

Junfeng Liu et al.

Dear Editor,
We have carefully revised the manuscript according to the comments from referee #1 and #2. The most important comments are that 1) organization of the manuscript and presentation of the data and results are unclear and insufficient; 2) the structure of the paper can also be much improved and the paper can be condensed; 3) no SSA measurements were acquired and the impact of the variation of the snow grain sizes with the metamorphism is not involved; 4) the impact of snow patches on a bare ice surface need exclude; 5) albedo parameterization methods are not "physically-based", because they are not derived from a fundamental principle or process.

We have given some carful explanations in our reply;please see the detailed point-by-point responses below. The corresponding changes have been made in the revised paper, track changes was used in order to be easily identified. Marked-up manuscript was given at the end of the replies. We hope the revised manuscript is suitable for the journal.
Best regards,
Junfeng Liu

**Reply to comments from Anonymous Referee #1**

General Comments

Authors measured the spatial and temporal surface albedo variations over the Qilian mountain glacier. They analyzed the albedo fluctuation considering the presence of small-scale surface roughness and light-absorbing impurities. They proposed an appropriate parametrization of albedo with these two parameters. Reviewer gives a  certain appreciation for the reasons that authors show a strong impact of these contributions on the surface albedo at different location (snow cover, ice cape, bare ice) using an extensive dataset of in situ measurements. However, unfortunately, the organization of the manuscript and presentation of the data and results are too hard to follow. In particular, it is not clear what is the novelty of this paper compared on earlier work of Warren, Cathles, Pfeffer, Lhermitte and many others. The explanations and some results are insufficients. For now, the paper is not a significant advance or contribution, even though it has the potential with interesting measurements. In addition, the

structure of the paper can also be much improved and the paper can be condensed. Substantial revisions are needed before having a merit for the publication in the TC. Reviewer is particularly concerned about the fact that no SSA measurements were acquired and the impact of the variation of the snow grain sizes with the metamorphism is barely speaken. It is difficult to believe in an exhaustive study of the impact of surface roughness/LAP on albedo without having an idea of the impact of snow metamorphism, since these 3 parameters are strongly linked in albedo variations. Authors should carefully confirm the results by including this analysis, maybe by adding a section speaking of this parameter (or using existing values of snow grain size published in the literature over this region). Moreover, reviewer suggest to modify the structure of the manuscript to present the results clearer. The organization of the manuscript and presentation of the data and results need some improvement. Indeed, sections are mixed, and there are many repetitions that can be easily avoid. The protocols of measurements are not enough detailed whereas it is very important to be confident on the measurement analysis. Finally, a deep revision of the english grammary has to be done to make the lecture easier.

**Reply: Thanks for your comments. We have revised the manuscript based on your suggestions and comments. We have revised the manuscript substantially: the structure of the paper has changed to present the results of snow part and ice part clearer. The measurements of albedo and LAIs sample are introduced in detail. We add snow metamorphism and corresponding surface roughness changed in the revised manuscript by adding detailed analysis of field observations and manual photography(We carried out field observations of snow grain size and corresponding surface roughness based on you suggestion in July of 2020 in August-one ice cap, the new data is presented in Figure 9 of the revised manuscript). Especially the manual photogrammetry could give provide direct visual evidence of old snow have rougher surface over fresh snow. The revision of the English grammar are also done to make the lecture easier to read.**

Specific comments :
1) The english has to be carefully corrected. Reviewer tried to highlight some errors in the 'technical corrections' section but there are many english mistakes making the reading difficult.
**Reply: Thanks for your comments. We have revised the English mistakes based on you 'technical corrections' suggestions and polished by language services.**
 2) Sections have to be read carefully to remove repetitions and to make the conclusion clearer. For example, it is often that in a 'snow section' authors speak about ice albedo measurements. Moreover, reviewer suggests to add small sentences at the end of each section to highlight the result. For example 'over ice cape, the albedo increases while the surface roughness increases and the LAI decreases'.
**Reply: Thanks for your suggestions. We have revised based on your suggestions, and add small sentences at the end of each section to highlight the result. The structure of the manuscript has revised and made the 'ice section' and 'snow**

**section' more clear.**

3) Explanation and results are insufficient: scatter plots presenting field observations have to be followed by physical analysis, according to what has been shown in the literature. Deeper explanations are needed to better understand the novelty of this work.

**Reply: We have add physical analysis according literature of similar research or relevant topics. This work not only considers surface layers snow or ice surface characteristics (surface roughness), but also considers LAI, and considers these two factors as important surrogate variables to explain more albedo variation than formerly used parameters. We have cited works such as Jonsell et al. (2003) who mentioned that uppermost surface ice layer, including its density, crystal structures, surface roughness, impurity content and stratification and SSA, are subject to continuous changes and mainly responsible for the observed variability in ice albedo. In this work we abstract the surface layers surface roughness might be an effective surrogate variable to be applied in albedo parameterizations. We also cited SSA related works (Taillandier et al., 2007; Gallet et al., 2009) to relate the snow metamorphism and albedo in the revised manuscript.**

**Snow surface roughness evolution include the snow metamorphism information and differential melting in melting season; In accumulation season, snow surface roughness is induced by blowing snow induce sastrugi and dry snow metamorphism.**

4) Figures need to be well called in the text, it is hard to follow the analysis when the text does not refer to the right figure. Moreover, reviewer suggests small changes in the graphs to be clearer (in the 'technical corrections' section). For example, It could help the understanding if authors replot the Figure 3 with different symbols/colors associated to 1) measurements taken over snow cover 2) measurements taken over ice surface.

**Reply: we have revised accordingly, in the revised manuscript, we differentiate the manual and automatic observations, we also differentiate the snow and ice surface. We have include key meteorological air temperature and precipitation information in the revised manuscript. Snow surface photo and corresponding DEM and DEM profile variation are also add in the revised manuscript.**

5) As said in the 'General comment' section, it is not clear what relationships have been high lighted in this paper. There is a strong relationship between SZA/SSA/LAI/roughness/albedo. The impact of the solar zenith angle and SSA variations needs to be analysed deeper.

**Reply: Thanks for your excellent suggestions. We have add SSA, LAI, SZA effect in analysis. The impact of the solar zenith angle are mentioned in the observation part. Some supplement observations include SZA carried out in July 2020 has provided in the revised manuscript. In the August-one ice cap, the ice surface is not sensitive to zenith angle because of high concentration of LAI over surface and melt water. In the revised manuscript we have cite some research work about zenith angel effects. The snow surface is sensitive to zenith angle. For that reason, we calculated daily albedo from the integrated sum of incoming and outgoing shortwave radiation of only 10min albedo data taken when zenith angles were less than 60º, in an effort to minimize solar zenith angle effects.**

6) The Protocols/Measurements sections need more explanations. For example, the accuracy of the instruments should be precise, and references presenting the instruments are missing. Reviewer has some questions about the protocol performed to acquired albedo measurements: How do you estimate the direct/diffuse part of the albedo? Measurements are acquired in clear sky conditions? At which height was located the sensor (ie what is the area actually seen by the sensor) ? What if you measure the aluminium square in addition to the snow surface?

**Reply: we have detailed the observation in the revised manuscript. We do not differentiate direct and diffused part. The observation include cloudy and clear sky conditions. The sensor is 1m above surface for manual observations, and 1.5m above surface for automatic observations. The surface roughness is measured first. After then, the aluminum square is moved and albedo was measured. LAI sample was sampled after surface roughness photography.**

**The albedo measurements part has changed as "After manual photogrammetry and the removal of the aluminum square, a Kipp & Zonen CMP11 albedometer, consisting of two identical pyranometers, measured the incoming global solar and the radiation reflected from the surface below. Albedo is the ratio of the two irradiances. The Kipp & Zonen CMP 11 pyranometer is sensitive to radiation in the wavelength range of 0.285 to 2.8μm. The albedometer was mounted on a camera tripod 1m above the glacier surface and three readings were made at each point on a horizontal plane by means of a spirit bubble. Albedo was measured within 3h of local solar noon, to minimize the effects of diffuse radiation and high zenith angles. "**

7) There is a problem of unity in equation 2, epsilon is in cm while the h* variable is in meters.

Reply: **We have revised accordingly.**

8) Nothing is new in the discussion part, except the parametrization method. Authors insist on the fact that it is physically-based, but it is not, it is fitted over the area so it is empirical. The sections should be better organized, this is very confusing. Moreover, the new parametrization should be investigated deeper, the associated error needs to be estimated for each type of surfaces (using control points over the snow covered surface, the ice surface).

**Reply: we have revised the physically-based as surrogate variable. The validation of new parameterization method was not present because of lake snow and ice surface LAI observation in 2019. The parameterization method have presented in Table 1 which include RMSE, albedo range and P-test.**

Technical corrections

L13: 'Fluctuations in surface albedo are due primarily to variations in micro scale surface
roughness (_) and light-absorbing impurities (LAIs) in this region.' => Please add the SSA+solar zenith angle

**Reply: Thanks for your suggestion. In the revised manuscript, we have considered the solar zenith angel's effect, in which bare ice surface is less affected by it than snow surface.**

**In the revised manuscript of line 210-220, we revised as' Solar zenith angle has a strong effect on snow surface albedo especially when zenith angles were larger than 60° (Gardner and Sharp, 2010; Dickinson et al., 1986). For that reason, we calculated daily albedo from the integrated sum of incoming and outgoing shortwave radiation of only 10min albedo data taken when zenith angles were less than 60°, in an effort to minimize solar zenith angle effects. Note that this weighs the albedo calculation to the middle of the day when insolation is highest and we do not consider zenith angle effects. Bare ice albedo is insensitive to the zenith angle for this impurity-rich glacier, due to LAI isotropic absorption by LAIs and water on the glacier surface (Marshall and Miller, 2020). In this case, the manual and automatic albedo observations are insensitive to the zenith angle in late July and August as the glacier surface was mainly bare ice.'**

**In the L13, we have revised as ' Fluctuations in surface albedo are due primarily to variations in the physical properties of the uppermost surface snow layers' s pecific surface area (SSA), surface roughness (ξ) and light-absorbing impurities (LAIs).'**

L.22: by consider => by considering

**Reply: we have revised accordingly.**

L.43: english => if we want to improve the accuracy of the energy budget estimate

**Reply: Thanks, we have revised accordingly**

L.52: can accelerates => can accelerate

**Reply: Thanks, we have revised accordingly.**

L.54: which enhance => which enhances

**Reply: Thanks, we have revised accordingly**

L. 60: Please rewrite: 'For the Qilian mountain glaciers, where the measured daily mean albedo decreased to the lowest of 0.13_0.06 due to the effect of LAIs for four glaciers observed during melting season'=> For example: For the Qilian mountain glaciers, the measured daily mean albedo decreased to 0.13_0.06 due to the effect of LAIs for four glaciers observed during melting season

**Reply: Thanks for your suggestions, we have revised as 'Qilian Mountains are located on the northeastern edge of the**

**Tibetan Plateau, LAIs have decreased the surface albedo of glaciers in Qilian Mountains to values as low as $0.13\pm0.06$ during the melting season (Jiang et al., 2010; Liu et al., 2014; Qing et al., 2018; Sun et al., 2018)'.**

L63: Please put this sentence before, in the section above: 'As the snow melts, insoluble LAIs are retained at the snow surface, so concentrations of LAIs in surface snow increase with snow melt, further reducing snow albedo (Doherty et al., 2013).'

**Reply: Thanks for your suggestion, we have revised and delete this sentence here accordingly. We add this sentence in the before section of Line 57-Line 60.**

L75: Studies have indicates => Studies indicated

**Reply: Revised.**

L80: the distribution LAIs=> the distribution of LAIs

**Reply: Revised**

L85: Surface roughness structure => 'Surface roughness features' seems more appropriate.

**Reply: Revised**

L88: 'During the past 5 years the August-one ice cap has become darker due to the accumulation of LAIs' => Please add a reference.

**Reply: Thanks for your suggestion, we have revised as' Successive time-lapse photography of the August-one ice cap has indicate that ice surface become darker due to the accumulation of LAIs in melting season (Qing et al., 2018).'**

L90: has indicate => has indicated

**Reply: revised**

L97: we try to investigate => we investigated

**Reply: revised**

L100: at different altitudes and times => and resolution? Or please add the resolution. => This section needs to be rewrite to present clearly the plan of your study. For example: 'first, to study the spatial variation.../second: : : Considering the following structure of your paper: 1rst objective: spatial variation, just the relationship between manual photo/lai/albedo 2nd objective: temporal variation.

**Reply: Thanks for your valuable suggestion, we have revised.**

**This paragraph is revised as' First, we investigated the spatial distribution of surface roughness via manual photogrammetry, manual LAI samples and albedo over different altitudes along the main flow-line of the August-one ice cap during the 2018 melt season. The effects of micro-scale surface roughness and LAIs on glacier snow and bare ice albedo was investigated at different altitudes and times. Second, automatic photogrammetry of surface roughness and automatic observation of glacier surface albedo was conducted on the middle of the ice cap in the melting and accumulation season of 2018. The effect of surface roughness on glacier snow and ice albedo was also analyzed at the daily scale in the**

melt and accumulation seasons separately. Third, the surface roughness effect on snow and ice surface albedo was analyzed based on combined manual and automatic photogrammetry at different altitudes, times and resolutions. Forth, snow and ice surface albedo parameterization methods are established based on only surface roughness, LAIs or both LAI concentration and surface roughness in the melt season, separately. Finally, since the surface roughness is dependent on and sensitive to topography data resolution, we calculated surface roughness under different resolution from 0.4mm to 333mm to investigate its correlations with albedo at plot scale. Appropriate resolution to surface roughness calculation was suggested for snow and ice albedo estimation, respectively.'

L100: Combine with => these measurements were combined with

**Reply: we have revised it.**

L108: based on => using only : : : or both : : :

**Reply: we have revised 'either' as 'only'.**

L118: It ranges in elevation => the elevation ranges from : : :

**Reply: Revised.**

L125. The glacier becomes darker with years? It was said in the introduction. Please clarify.

**Reply: we have 3 automatic time lapse photography site distributed over the August-one ice cap from terminal to top. The ice surface LAIs (most cryoconites) concentrated over ice surface in summer. For the last 6 years, no accumulation was observed even at top of the ice cap, the concentration of cryoconites has increased because of negative annual mass balance. We have revised as 'Influenced by climate warming and continuous accumulation of LAIs, the entire August-one ice cap surface darkens during the melt season and no accumulation area has been observed for the last 6 years based on mass balance observations (Liu et al., 2020).'**

[Figure]

**Figure 1 Cryoconites covered ice surface (grey part is sun dried cryoconite, brown part is wet cryoconite). The image of right size was captured by time lapse camera, which has captured glacier surface for the last 6 years.**

L134: Please specify if you are talking about 3D photogrammetric acquisitions or 2D photogrammetry, and the references need to be adapted (for instance, Manninen = 2D acquisitions with a board, Irvine-Fynn = 3D acquisitions)

**Reply: we have revised as '3-D photogrammetry'**

L139: of _1.75 m 2 => it is a very small area, is it representative ? How was the area chosen?

**Reply: The August-one ice cap is a small ice cap without obvious surface movement. The cap surface shows no relevant morphologies. Structures such as crevasses, cracks, and seracs are not formed. Only water channels formed over ice surface with very small proportion. In this study we selected all ice surface types except water channels. The selected snow or ice surface include smooth snow covered surface, partial snow covered surface, cryoconite holes, weathering crust and smooth ice surface.**

**In the revised manuscript, we have add the selection standard for manual photogrammetry on 12 and 25 July, and 28 August. We have revised as' The August-one ice cap is a small ice cap without obvious surface movement. The cap surface shows no relevant morphology. Structures such as crevasses, cracks and seracs have not formed. Channels account for only a small portion of the glacier surface. In this study we selected all surface types, including smooth snow-covered surface, partial snow-covered surface, cryoconite holes, weathering crust and smooth ice surface, but not water channels. These photographs allowed us to calculate the micro-scale topographic over different altitudes. Topographic surface roughness was estimated based on equation (1) and these micro-scale topography data.'**

L 139: Please clarify: 'by surrounding the target area of snow or ice surface', do you mean by turning around? If yes, what is the space between each step/picture?

**Reply: Thanks for your question, we have revised as' Seven to fifteen photos were taken at each survey site by surrounding the portable aluminum square. At each cardinal direction of aluminum square, we took 2 or 3 photos to cover the target area. Rough ice surface such as cryoconite holes need take additional photos close to nadir angle.'**

L143: Please specify the number of sampled areas.

**Reply: we have revised 'several' as '37'**

L147: 'Glacier surface albedo was calculated from measurements of up and downward shortwave radiation.' => how were acquired the up and downard shortwave radiation? Please add details about this protocol and sensor.

Reply: we have revised it as' **After manual photogrammetry and removal of aluminum square, a Kipp & Zonen CMP11 albedometer consists of two identical pyranometers that measure the incoming global solar and the radiation reflected from the surface below. Albedo is the ratio of the two irradiances. Pyranometer is sensitive to radiation in the wavelength**

**range 0.285 to 2.8μm. The albedometer was mounted on a camera tripod 1m above glacier surface and readings were made in a horizontal plane by means of a spirit bubble**.'

L160: 'For automatic photogrammetry': how were process the pictures? With a software? Did you use only one reference target? Did you use control points? What is the error of the final DEM? Please clarify.

**Reply: we have used Agisoft to acquire the 1mm resolution DEM data. For manual photogrammetry, the reference control frame have 4 control points and 4 check point (eight crossshaped screws on the aluminum frame). For automatic photogrammetry, we have 4 control points and 3 control points over the control frame. The error of the final DEM have provided in the revised manuscript.**

We have revised as' **For manual photogrammetry, we put the aluminum frame horizontally over the ice surface, the plot is detrended by setting 4control points and 4 check points at z axis of the same values. For automatic photogrammetry, the control field of wooden frame was also laid horizontally over the ice surface that lowered as the ice melted and maintained a horizontal position between the control field and ice surface. 4 control points and 3 check points are on the wooden frame was applied for georeference and precision check. Both manual and automatic photographs were imported into the Agisoft Photoscan Professional 1.4.0 application, which produced plot-surface point clouds and generated a detrended micro scale DEM of 1 mm resolution at plot scale. Based on the Agisoft PhotoScan processing report, automatic photogrammetry average geo-referencing errors fluctuated at around 1 millimeter. Total root mean square error (RMSE) of the automatic check points was 3.62$\pm$1.6mm. Standard deviation of control and check point errors were all within 15mm. Manual measurements average geo-referencing errors fluctuated at around 1 millimeter, RMSE of 4 check points was 0.99$\pm$0.3 mm, vertical accuracy was 0.66$\pm$0.3mm. Standard deviation for x, y and z axis were all within 5mm**'

Figure 2: Please rename a) and b) because they have the same caption. + remove '[Figure 2]' in L 181 => please removed this typo for all figures.

**Reply: Thanks for your suggestions, we have revised and rename a) and b). [Figure 2] also removed in the revised manuscript.**

**The revised Figure 2 is :**

[Figure]

a: rough ice surface

Sunny

Weathering crust

Cold, dry and windy

depressions

Cryoconite hole

Light-absorbing impurities

Ice cap ice

b: smooth ice surface

Rainfall

Cloudy

Warm, humid and windy

Ice cap ice

Light-absorbing impurities

**Figure 2. Schematic images of cross section of the ice surface LAI distribution over a rough and smooth ice surface, in which warm, cloudy or rainfall days tended to produce a smooth ice surface, and cold and sunny weather tended to produce a rougher ice surface. Panel (a) shows LAIs hidden beneath the weathering crust or cryoconite holes, or covered by snow, and depressions have more LAIs than protrusions; panel (b) shows LAIs distributed over a relatively smooth ice surface without being hidden or protrusion effects.**

L185: repetition => 'Ice surface albedo is very sensitive to the LAIs concentration over ice surface.' Please add a sentence of transition, for example 'the surface roughness features impact the distribution of the snow impurities'.

**Reply: Thanks for your suggestions, we have revised based on your suggestion**

**We revised as 'The surface roughness features impact the distribution of the glacier surface impurities'**

L190: by direct => by directly affecting

**Reply: revised**.

L 192: There is a problem with the unity of equation 2. epsilon is in cm while the h* is in meters?

**Reply: We have revised 'm' as 'cm' for the h*.**

L192 : equation 2 is the first equation => it is equation 1 here. Please correct the other equation notations in the text.

**Reply: we have revised accordingly.**

L 195: 'from Lettau (1969) developed aerodynamic surface roughness' => 'from Lettau (1969) who developed aerodynamic surface roughness...'

**Reply: Thanks for your suggestion, we have revised as 'The definition of ξ in this study is adopted from the aerodynamic surface roughness parameter developed by Lettau (1969)'**

L198: adopted => adapted

**Reply: Done**

L202: Please add a reference to introduce this metric.

**Reply:Done**

L205: This sentence needs to be rewrite: For rough ice surface means more surface area and less concentration of LAIs over ice surface, I don't understand.

**Reply: we have revised as 'A rough ice surface means a greater surface area and lower LAI concentrations over the smooth ice surface'.**

L207: by consider => by considering

**Reply: Done**

L207: this is a new metric? If yes, please change 'defined' by 'introduced'

**Reply:   It is a new metric, we have revised 'defined' as ' introduced' accordingly.**

L220: 'For that reason, we used only half-hour albedo data taken when zenith angles were less than 60_' => Did you test the accuracy of the sensor considering the zenith angles? Is this a known angle of limitation? Or please add a reference.

**Reply: Thanks for your questions, In the revised manuscript, we have give detailed observation of albedo and calculation of albedo method. For the zenith angles effect. We also considering its effect.**

**We have revised and moved this part to the method part of line 185-191. We have revised as 'After manual photogrammetry and the removal of the aluminum square, a Kipp & Zonen CMP11 albedometer, consisting of two identical pyranometers, measured the incoming global solar and the radiation reflected from the surface below. Albedo is the ratio of the two irradiances. The Kipp & Zonen CMP 11 pyranometer is sensitive to radiation in the wavelength range of 0.285 to 2.8μm. The albedometer was mounted on a camera tripod 1m above the glacier surface and three readings**

were made at each point on a horizontal plane by means of a spirit bubble. Albedo was measured within 3h of local solar noon, to minimize the effects of diffuse radiation and high zenith angles.' In revised manuscript, we also talk about automatic observation of albedo and zenith angle effect over snow and ice surface.

We have revised as ' A Kipp & Zonen CNR4 net radiometer was set up to record incoming and reflected solar radiation of the control field surface. The pyranometers are sensitive to radiation in the wavelength range 0.3 to 2.8μm. Samples were taken every 15s; 10min means were stored on a data logger (CR800, Campbell, USA) located at a height of 1.5m. These measurements were taken over 3-month period from July 12 to October 15 2018. Solar zenith angle has a strong effect on snow surface albedo especially when zenith angles were larger than 60 º (Gardner and Sharp, 2010; Dickinson et al., 1986). For that reason, we calculated daily albedo from the integrated sum of incoming and outgoing shortwave radiation of only 10min albedo data taken when zenith angles were less than 60º, in an effort to minimize solar zenith angle effects. Note that this weighs the albedo calculation to the middle of the day when insolation is highest and we do not consider zenith angle effects. Bare ice albedo is insensitive to the zenith angle for this impurity-rich glacier, due to LAI isotropic absorption by LAIs and water on the glacier surface (Marshall and Miller, 2020). In this case, the manual and automatic albedo observations are insensitive to the zenith angle in late July and August as the glacier surface was mainly bare ice.'

L222: from 11:00 to 17:00 o'clock => what range of zenith angle it is? It strongly depends to the date of your measurements + elevation of the studied site. Please clarify. => It seems
that this section should be written in the protocol of albedo measurements, not here
in this 'result' section.
**Reply: we have revised this part and add in the '2.2 data collection' section**
**More details can be find in Line 185-191 and Line 208-220.**
L225: field investigation => is it based on your automatic photogrammeric measurements? Please clarify.
**Reply: We have revised 'field investigation' as 'manual observation'. The sentence has revised as' We found a patchy snow cover with many cryoconite holes on the glacier terminus (~4600 m) at the date of the first field trip (July 12, 2018).'**
L230: Surface roughness => replace it by 'the associated epsilon measurements decreased ...'
**Reply: Thanks for your suggestion, we have revised as' The associated surface roughness measurements decreased from 5.5±1.5cm to 0.5±0.6cm as altitude increased (Fig. 3b)'**

L236: 'Surface roughness fluctuated between 1.4_0.4cm to 3.3_1.1cm; An increasing trend of surface roughness could be detected as altitude increased (Figure 3e). ' => please, inverse these two sentences to be clearer.
**Reply: Thanks for your suggestion, we have revised accordingly.**

**We have revised as 'An increasing trend of surface roughness could be detected as altitude increased and fluctuated between 1.4±0.4cm and 3.3±1.1cm (Fig. 3e).'**

L245: 'surface roughness and LAIs decreased as altitude increased.' => not clear because it is only the case in July. In August 3: surface roughness increased as altitude increased and there was some snow cover at the top of the ice cap. Please clarify.

**Reply: Thanks for your suggestions. We have revised this part based on your suggestions.**

**We have revised as ' LAIs decreased from 0.05±0.01g cm$^{-3}$ to around 0.01±0.005g cm$^{-3}$ as altitude increased (Fig. 3f). On August 28, the ice cap surface was entirely bare ice. Albedo showed an increasing trend with altitude and fluctuated between 0.14±0.04 at the middle part to 0.19±0.05 at the top of the ice cap (Fig. 3g). Ice surface roughness fluctuated between 1.50±0.5cm at the middle of the ice cap to 1.45±0.95cm at the top of the ice cap, and it did not significantly increase as altitude increased (Fig. 3h). LAIs decreased from 0.04±0.03g cm$^{-3}$ at the middle part to 0.003±0.002g cm$^{-3}$ at higher elevations (Fig. 3i).'**

L247: 'There was a much higher concentration of LAIs on the uncovered ice surface than snow surface' => I don't understand the 'much', the trend is not so strong? It could help the understanding if your replot the Figure 3 with different symbols/colors associated to 1) measurements taken over snow cover 2) measurements taken over ice surface.

**Reply: Thanks for your suggestions, we have revised Figure 3 based on your suggestions. The snow and ice surface was differentiated in different symbols. The sentences also revised from Line 300 to 305.**

L248: 'As a consequence, albedo tended to be low on the ice surface and higher on snow-covered surfaces.' => be careful with this sentence, this is also mainly due to SSA values that strongly impact the albedo. There is a strong relationship between SSA/LAI/roughness/albedo. Please rewrite this sentence.

**Reply: we have revised accordingly.**

**We have revised as 'There was a much higher concentration of LAIs on the uncovered ice surface than on the snow surface. Albedo tended to be lower on bare ice surface than on snow.'**

L251: repetition with L240-243 => this section needs to be re-structured in order to remove repetitions and to be clearer on conclusions.

**Reply: we have rewrite this part and make it clearer. We remove repetitions and re-structured it.**

L260: 'The automatic measurement setup in the middle of the ice cap' => 'The automatic measurement setup was in the middle of the ice cap'

**Reply: Done.**

L.264: 'Intermittent snowfall decreased surface roughness and increased albedo' => you should add that it is because there is fresh snow, and fresh snow has high SSA, inducing a high albedo.

**Reply: we have revised based on your suggestion, Thanks.**

**We have revised as 'Because fresh snow has a large specific surface area (SSA) and small surface roughness, snowfall events on July 14 and 20 inducing a high albedo (Figs. 4a, 4b).'**

Figure 4: It could be clearer if you add the intermittent snowfall with vertical lines + air temperatures (to know when it is melting).

**Reply: Thanks for your excellent suggestion, we have revised Figure 4 based on your suggestion. We add precipitation data and half-hour air temperature data in the revised manuscript.**

L265: 'When snow began to melt'=> when is it ? How do you measure that? Do you have air temperature or snow surface temperature? If yes, you could add it on Figure 4 (see comment above).

**Reply: we have meteorological observations at top of the ice cap including precipitation, surface temperature and air temperature. We have revised Figure 4 and add air temperature accordingly.**

**We also rewrite this sentence and make it clearer. We revised as 'When the air temperature increased to 0 ºC and the snow began to melt (Fig. 4b), the glacier surface roughness kept increasing and albedo kept decreasing.'**

L276: 'When the snow melted, leaving patchy snow cover, surface roughness increased and albedo decreased quickly to around 0.3_0.05 within two day'. Here again the sentence needs to be rewrite: snow melting = lower SSA = lower albedo, the decrease of albedo is not only due to the increase of surface roughness and LAI. Please clarify.

**Reply: we have revised based on your suggestions.**

**We have revised as 'When the snow melted, leaving patchy snow cover. The decreased SSA and increased surface roughness led to a fast decrease of albedo to around 0.3±0.05 within 2 days.'**

L276: 'from September 4 to 15, ice surface roughness fluctuated and increased to 2.63 cm on September 13.' => not clear, is it increasing from september 4 to 15, or for September 4 to 13?

**Reply: we have revised it.**

**We have revised as 'At the end of melt season, from September 4 to 13, ice surface roughness fluctuated and increased from 0.15 cm to 2.63 cm.'**

L277: 'Ice surface micro scale structures of cryoconite holes hidden LAIs from direct sun light.' => this sentence needs to be detailed to highlight the observations. => Reviewer suggest to recall this section (for example '3- Field observations'), since it details all the field observations, but not clearly enter in the physical analysis of the relationships between LAI/roughness/albedo. The analysis should include SSA variations. Even if no measurements were taken, it has to be mentioned (relating to the snow type of the area for example).

**Reply: Thanks for your suggestion, we have revised based on your suggestion**

**We have revised as 'The ice surface micro-scale structures of cryoconite holes hid the LAIs from direct sunlight. Refreezing processes also formed weathering crust and covered LAIs (Fig. 4g). The ice surface albedo followed an increasing trend**

**during this period and increased to 0.55 on September 13. During the following 2 days, snowfall events increased albedo sharply from 0.55±0.02 to 0.83±0.06 (Fig. 4h) due to the high SSA and decreased snow surface roughness.'**

Figure 5: Please rewrite the x axis 'Surface roughness (cm)' with the epsilon metric.

**Reply: we have revised accordingly.**

L301: Figure 2s => I am not sure this figure provides a lot, since it is well resumed in Figure 5a. In Figure 5a, the measurements taken manually and automatically should be separated by two different symbols to be clearer.

**Reply: thanks for your suggestion, we have revised it.**

L302: 'The combined manual and automatic scatter diagrams of Figure 5a display a significant negative power function between snow surface roughness and snow albedo (Figure 5a, r=0.82).' => this is a repetition.

**Reply: we have revised this part accordingly.**

L314: Figure 3s => same as above, the Figure 5b gives approximately the same information.

Authors could put different symbols for manual/automatic measurements on Figure5b. Figure 4s: snow, patchy snow and bare ice surface => In the 'data and methods' section you should defined this terms. How do you classify an area as a 'patchy snow'? Is there a quantitative measurement of the proportion of snow in the area, or is it qualitative? How do you distinguish areas with very dense snow surface to ice surface? Please clarify.

**Reply: Thanks for your excellent questions. During filed observations, the albedo or roughness measurement is very sensitive to the snow covered and ice covered proportion on plot scale. In July of 2018, surface melt induced ice surface firstly appeared at terminal of glacier surface. The patchy snow cover formed a belt which shift upward in the melting season until all snow disappeared In August of 2018. A glacier surface is classified as patchy snow cover based on uncover of old dirty ice layers which induced very rough surface. In this study, we only have qualitative measurement based on the manual observations. In Figure 4c, we could tell it is patchy snow cover. We have add information in the 'Data and methods' section to clarify difference of snow, patchy snow and ice. In the revised manuscript, we have exclude patchy snow cover to analysis the albedo and bare ice surface albedo relationship.**

L 324: 'Scatter plots of Figure 5c and 5d shows relationship between C LAIs and albedo' => Reviewer suggests to introduce a new section dedicated to LAI-albedo relationship, or to separate the section to put the analysis of the LAI-albedo relationship over snow surface in the 'snow surface section', and over ice surface in this 'ice surface melting season'.

**Reply: Thanks for your excellent suggestions, we have add snow surface LAI-albedo relationship to the 'snow surface in melting season' section. The ice surface LAI-albedo relationship to the 'ice surface in melting season' section.**

L325: 'LAIs concentration is lower than patchy snow surface' => ' LAIs concentration is lower than the one measured over patchy snow surface'

**Reply: Done**

**In the revised manuscript, we have revised in Line 380 as 'Snow surface albedo is greatly affected by LAIs. The scatter plots of Fig. 6a show the relationship between $C_{LAIs}$ and albedo without considering the surface roughness effect on LAI concentrations. A significant negative power function relationship was established between snow LAIs and albedo (Fig. 6a, r=0.91).'**

L327: Figure 4c => Figure 5c, and Figure 4d => Figure5d Please a particular attention should be taken on the notation of Figures in the text, this is confusing.

**Reply: We have revised accordingly. In the revised manuscript, Figure 5 shows snow and ice surface roughness and albedo relationship. Figure 6 shows snow surface $C_{LAIs}$, $C_\xi$ and albedo relationship. Figure 7 shows ice surface $C_{LAIs}$, $C_\xi$ and albedo relationship.**

L329: Please introduce a new section, or reorganise this section into the 'snow surface section' and the 'ice surface section'.

**Reply: we have revised accordingly. We have revised as 'snow surface in the melting season', ' ice surface in the melting season', and 'snow surface in the accumulation season' .**

L329: 'the effective LAIs concentration of C _' => 'the effective LAIs concentration (C _)' ' by considers roughness effect' => 'by considersing roughness effect'

Reply: **Done**.

L330: C _ is estimated based on equation (4). => this is a repetition

**Reply: Done.**

L334: 'It indicates that the Ceps concentration IS not affectED by surface roughness.'
=> english L335: 'Equation (3) is more appropriate to calculate LAIs concentration than equation (4) over snow surface.' => I don't understand. Do you mean 'to retrieve LAI concentration directly from albedo measurements'? Please clarify.

**Reply: Here we try to express the view that microcale snow surface features do not hidden snow surface LAIs like ice surface does. In the revised manuscript, we have revised it as 'This means that considering the surface roughness effect on LAI concentration by equation (3) does not improve the relationship between LAIs and albedo during the snow period. It indicates that the snow surface LAI concentration is not affected by micro-scale snow surface features since melt-out LAIs are concentrated over the snow surface.'**

L339: 'by consider surface roughness effect over not consider it in Figure 5d'
=> english L341: 'It means equation (4) estimate C _ could explain more ice surface albedo than equation (3) calculate C LAIs .' => please clarify. Do you mean: could explain ice surface albedo variations?

**Reply: Yes, we mean equation (3) could explain ice surface albedo variations than equation (3). In the revised manuscript,**

we have revised it to be clearer.

**We have revised as' The coefficient of determination has increased from 0.35 to 0.63 after considering the surface roughness effect. This means that equation (3), estimating the effective LAI concentration $C_\xi$, could explain more ice surface albedo variation than $C_{LAIs}$ does.'**

 L352: English mistakes

**Reply: Done.**

L354: 'In accumulation season, except snow particles metamorphism process, constant blowing snow and intermittent snowfall was the main reasons which induced surface roughness fluctuation;' => More explanations are needed. In this Result section: the english has to be carefully corrected, some sections has to be restructured, the field observations and scatter plots have to be followed by physical analysis, according to the litterature, with more explanations. Figures need to be well called. Some changes in the graphs are suggested to be clearer.

**Reply: Thanks for your suggestions, we have revised this section accordingly. We have talked about surface roughness evolution in accumulation which is mainly induced dry snow metamorphism of decreased SSA, but also by the strong wind-formed sastrugi.**

**We have revised as 'In late September and October, the glacier surface is covered with dry snowpack and the LAI concentration on the fresh snow-covered ice cap is very low. In this case, the effect of LAIs is not presented in the accumulation period. The scatter diagram of snow surface roughness and albedo shows negative power function (Fig. 8, r=0.49). Theoretical studies have indicate that the initial size distribution, vertical temperature gradient and snow density all affects albedo evolution in this period. Vapor diffusion caused by curvature differences causes rapid albedo decay in the first day following snowfall (Flanner and Zender, 2006). Studies have indicated that under either isothermal conditions or under temperature gradient conditions, the dry snow specific surface area (SSA) decreases during metamorphism (Taillandier, et al., 2007). The decay of snow SSA produces a decreasing trend in snow albedo (Domine et al., 2006; Flanner and Zender, 2006). This is in addition to the snow surface being exposed to strong winds (average 4.8m s$^{-1}$ at 4 m above the surface) over the August-one ice in the accumulation period, which could also increase the rate of the SSA decrease and accelerate albedo decrease (Cabanes et al., 2003).'**

L380: 'The grain size is one of the most critical factors affect snow albedo' => 'The grain size is one of the most critical factors affect ING snow albedo.'

**Reply:Done**

L385: 'which is very similar with scatter plot of surface roughness and snow albedo provided in this study in Figure 5a and Figure 7.' => Please specify if this studies worked with the same snow type as you.

**Reply:We have revised it to wet snow type.**

L386: 'It means surface roughness as substitute of grain size is quite suitable for snow surface albedo explanation and parameterization at millimeter scale' => Please be careful with this conclusion, the study areas are not the same. To conclude this you should have both measurements (ssa and roughness) at the same site, at the exact same time (same sza) and over a dry and clean snow (no impurities, no liquid water content). Please clarify. Several contributions affect the albedo, and it is very difficult to separate each contribution to better understand/quantify their impacts on the albedo.

**Reply:Thanks for your excellent comments, we have revised it accordingly. We add Figure to connect surface roughness and snow grain size and differential melting. Because in melting season, surface roughness is controlled by differential melting and wet snow metamorphism.**

L388: english

**Reply: Done**

L406: 'We expect that different mechanism of surface roughness evolution over melting season and accumulation season are the main reasons of different statistical relationship.' => This is not the main reason, the SSA variation also plays a strong role. Please clarify.

**Reply:We have revised this part accordingly. We add SSA evolution and cited snow research work about SSA. The snow surface albedo was directly connect with snow grain size and SSA. We have revised it as 'At the August-one ice cap, temperatures are above freezing during the day and snow is subject to rapid metamorphism and production of larger, typically rounded grains and clusters. Meltwater can refreeze at the surface forming crusts when air temperatures decrease at night. The melt and refreeze patterns lead to hardened crusts made up of aggregated rounded grains and clusters of grains that have frozen together. The snow surface roughness follows an increasing trend due to snow grain particle metamorphism and differential melting of the surface. Rapid grain growth corresponding with decreasing SSA and increasing of surface roughness. There is often a concurrent rise in albedo due to the blanketing of old snow by fresh snowfall. New fresh snow surface has reduced surface roughness by preferential deposition, smaller grains and larger SSA, which increase the albedo. Fig. 9 shows the snow grain size, fresh and old snow surface roughness measured on 14 July 2020 at the top (4820 m) and lower part (4623 m) of the ice cap. We could find old snow surface roughness is larger than**

fresh snow surface due to its larger snow grain size and greater DEM variation. Due to preferential deposition, more fresh snow deposited on the pits than peaks (Fig. 9d). The snow grain size at the pits are smaller than the peaks (Fig. 9d). The DEM variation are also larger at the peaks than pits due to its larger snow grain size (Fig. 9f). The surface roughness fluctuation synchronously happened with SSA and grain size variation. Although we do not have tandem surface roughness, grain size and SSA observations, the evolution of surface roughness calculated at 1mm resolution in the snow-covered period should be quite similar with the fluctuation trend of grain size evolution. The grain size is one of the most critical factors affecting snow albedo. As grain size increases, scattering within the snowpack decreases and the absorbing path length within grains increase, thus reducing the broadband albedo (Wiscombe and Warren, 1980; Dozier et al., 1981; Warren, 1982; Flanner and Zender, 2006). Linear or exponential relationships have been established between wet snow grain size and snow albedo (Brock et al., 2000; Adolph et al., 2016; Skiles and Painter, 2017) in the melt season, which is very similar with the scatter plot of surface roughness and snow albedo provided in this study in Fig. 5a. As discussed in the preceding sections, both increasing surface roughness and LAI concentration tended to decrease albedo in snow melting period. The hidden effect or dilute effect of surface roughness on LAI concentrations is not discerned by application of equation (3) to the snow surface, as other studies have already proven that LAIs are retained at the snow surface (Doherty et al., 2013), and scavenging by melting process is limited (Skiles and Painter, 2017).'

Section 7.2: The name of the section should be changed: authors speak about macro-scale surface roughness effects in this section, and in the section before it was at millimeter-scale.

**Reply: we have revised and not mentioned macro scale roughness effect in this part and only talks about micro scale surface features effects over albedo. In the revised manuscript, we talk about 'Snow surface albedo variability explanation', 'Ice surface albedo variability explanation' and 'Glacier surface albedo parameterization based on micro-scale surface roughness data'**

L415: 'more incident radiation is absorbed by the slope facing the sun than by the slope facing away from it.' => this is not the only reason (see Warren et al 1998). The main reason is that photons are trapped between cavities, which increases the chance of a photon to be absorbed. Please clarify and detail the physical analysis.

**Reply:Thanks for your suggestions, we have revised it and add photons trapped effect for rough surface.**

**We have revised as 'It has been shown that small or macro-scale surface roughness reduces surface albedo and enhances solar absorption due to decreasing incidence angle and light trapping'**

L430: filed investigation => english L430-433: Please rewrite the sentence, I do not understand

**Reply: we have revised accordingly, the revised part was' Additionally, we find LAIs were concentrated in depressions rather than protrusions. This heterogeneous distribution characteristics of LAIs across rough ice surface led to a reduction of shortwave radiation absorption (Fig. 10f).'**

L441: 'Rough surface means lower concentration of LAIs and high albedo over smooth ice surface with heavy loading of LAIs and low ice surface albedo.' => this is confusing to speak about rough surface of a smooth ice surface. How do you define a smooth ice surface? What is the scale of roughness features that you are talking about? Please clarify.

**Reply: we have revised and clarify that we are talking about microscale surface features.**

=> this is very confusing between Sections 7.1 and 7.2 => when are you talking about ice surface and snow surface? For example: the first section of the section 7.2 is talking about snow surface (see the litterature), and not the ice surface. Please clarify.

**Reply: we have revised and add snow surface to the Section 6.1, and ice surface to the Section 6.2.**

L447: 'Most of the established and widely used snow surface albedo parameterization ARE either based on snowpack age, snow depth, snow density, air temperature' => english => Please add the SSA and SZA (see the litterature of Kokhanovsky) => This sentence should be put in the section 'snow surface', not 'ice surface'.

**Reply: we have revised and removed snow part to the 'Snow surface albedo variability explanation' section.**

L450. Please add a reference

**Reply: we have revised accordingly.**

**We have revised as 'The performance of the established albedo methods either are based on surface roughness in Fig. 5b or effective LAI concentration in Fig. 7b shows great improvement over the assumption of a constant mean ice albedo or surrogate variables, such as air temperature, accumulated melt and elevation (Brock et al., 2000). The physical properties of uppermost surface ice layer, including its density, crystal structures, surface roughness, impurity content and stratification and SSA, are subject to continuous changes as the melt season proceeds (Jonsell et al., 2003).'**

L451: 'We expect the new parameterization methods provides' => english 'are more physically based than some of the studies presented.' => What new parametrization? Please refer to the equation here. (as equation 5 but for ice surface) This sentence needs to be rewritten:this is not a physically based parametrization, it is empirical. Indeed your equation 5 strongly depends of your study site and your measurements.

**Reply: Thanks for your suggestions, we have rewritten this park and make it more clear than before.**

**We have revised as 'In this study, surrogate variables of effective LAI concentrations consider both surface roughness and LAIs, explaining a significant amount of ice surface albedo variation, and $C_\xi$ has a higher coefficient of determination than either surface roughness or LAIs (Table 1).'**

L466: 'For snow covered surface, the coefficient increases quickly from -0.67 to - 0.74 when 1m plot resolution increases from 333.3 mm to 200.0 mm' => but the Figure 8a shows a very flat and smooth snow surfaceǎa? Please clarify the studied surface.

**Reply: we have clarify in the revised manuscript, the snow covered surface actually include flat surface and rough partially snow covered surface. The statistical between surface roughness and albedo include partially snow cover and flat snow**

**surface. It could be the reason the snow surface roughness and albedo still sensitive at around 100mm resolutions.**

L485: 'This could be a practicable way to parameterize surface roughness and albedo
on a whole-glacier scale' => Please add some details. It could be interesting to apply
the method proposed by Smith et al 2016 here, to estimate a temporal mean albedo at
the whole-glacier scale.

**Reply: we have revised accordingly.**

**We have revised as 'For working at whole-glacier scale, recent developments in terrestrial laser scanning, structure-from-motion, and multi-view stereo may also be able to provide catchment-scale high-resolution topography and surface roughness data ( Passalacqua et al., 2015; Rippin et al., 2015; Smith and Vericat, 2015; Rossini et al., 2018; Xu et al., 2019). Studies such as unmanned aerial vehicle photogrammetry create centimeter-resolution DEM and orthophotos for glacier scale. These data have already been applied successfully for glacier surface roughness and brightness (albedo) analysis (Rossini et al., 2018). Smith et al. (2016) have suggested a statistical method to extrapolate glacier-scale roughness from finer-resolution plot-scale surface roughness measurements and glacier-scale DEM data. This could be a practicable way to parameterize surface roughness at the glacier scale. This statistically derived glacier-scale surface roughness could also be applied for estimation of albedo based on the surface type and roughness–albedo statistical relationships. Additionally, LAI coverage can be captured by high-resolution images (e.g., Takeuchi et al., 2018)'**

Section '7.3 Glacier surface albedo parameterization at whole-glacier scales based on
surface roughness' => the title needs to be changed since you investigate the sensitivity
of the albedo relationships at a plot-scale, but not at the whole-glacier scale in this section.
=> Moreover, please rely your analysis with the litterature (Irvine-Fynn et al 2014; Smith et al 2016 ..) to clarify what is the
novelty of what you propose. => It could be interesting to estimate the LAI coverage with your photogrammetric acquisitions as
Takeuchi et al 2018.

**Reply: we have revised the title as 'Glacier surface albedo parameterization based on micro-scale surface roughness data'. In the revised manuscript, we talk about micro scale surface roughness effect and macro scale surface roughness effect. We talked about this manuscript's merit and deficiencies of the article.**

L497: 'we have a general understanding of the surface roughness that controls the
albedo of snow and ice surface are quite different.'

**Reply: we have rewrite this part accordingly.**

**We have revised as 'By using manual and automatic photogrammetric measurements of surface roughness, manual LAI samples and measurement of broadband albedo at the August-one ice cap, we have gained a basic understanding of the surface roughness and LAI effects over snow and ice surface.'**

=> english L502: 'necessity to consider'

=> english L504: 'Surface roughness seems play a quite'

=> english L535: 'field data of LAIs, surface roughness and albedo need to collect to help us to present more detailed analysis and modeling research about surface roughness and LAIs at

micro scale over ice or snow surface energy and mass balance process.'

=> SSA measurements should be acquired too.

**Reply: Thanks for your valuable advices, we have revised accordingly.**

**We have talk about the measurement of SSA, snow grain size and surface roughness in future.**

**We have revised as 'More complete data on LAIs, surface roughness, snow surface area, SSA, snow grain size and albedo need to be collected to help us present a more detailed analysis and theoretical-based research on surface roughness and LAIs at the micro scale over ice and the snow surface energy and mass balance processes. In particular, the inclusion of SSA, snow grain size surface roughness, and albedo measurements on snow and ice albedo will allow for better assessment of the potential response of the glacier to a changing climate in future as well as improving the modeling of the energy and mass balance, not only in the glaciers of the Qilian Mountains, but also in other glacial regions that experience severe LAI effect, such as in Central Asia and the Himalaya.'**

Liu et al. present a dataset consisting of field measurements of albedo, surface rough-ness and surface concentration of Light-Absorbing Impurities (LAIs) collected during a melt season on the August-one ice cap in the Qilian Mountains, China. The measurements were collected from July to October 2018, spanning most of that melt season (July to September) and one month (October) of the next accumulation season. The collected data are used to propose statistical relations between the three main monitored variables (albedo, surface roughness and surface concentration of LAIs), and to discuss the physical relations between them, among other analyses. In particular, they propose that there is a combined effect of surface roughness and LAIs surface concentration on the albedo of snow and ice.

I think that the topic of this investigation is relevant for the scientific community and The Cryosphere. Although some additional details about the sampling methodology are needed, the data seem well collected, and the physical processes that are proposed to explain the observations are interesting and reasonable. Unfortunately, I don't think that the authors convincingly prove their hypothesis about the combined influence of surface roughness and LAIs, because i) they tend to ignore or minimize the role of snow metamorphism on snow albedo, and ii) the impact of snow patches on a bare ice surface. Please see my main point below for details in this regard. Importantly, the structure of the article is confusing, and the text and figures can be condensed and better organized. Finally, the use of English language should be revised, as it sometimes obscure the ideas of the authors. My recommendation is to reconsider this article after major revisions.

GENERAL COMMENTS

Physical connections between surface roughness, LAIs and albedo Process understanding

My main comment is that the authors tend to overestimate the influence of surface roughness on albedo in their analyses and conclusions. I have no doubt that there is a statistical relation between these two variables (as shown in Figure 5), and also some physical connection (Larue et al., 2020), but the collected data does not necessarily suggest a causal relation. As the authors do in the paper, I divide my arguments for the periods of snow melting and bare ice melting.

Period of bare ice exposure: Figures 2b and 2c present a good summary of the main idea discussed by the authors. A rough ice surface (with a surface roughness of about 1-10 cm) hides some LAIs, raising the area-average albedo relative to surfaces with uniform debris cover (Bøggild et al., 2010). If the surface smooths over the melting season, the LAIs emerge and the albedo

decreases. The authors parameterized this effect using equation (4). However, based on the data shown by the authors, I would say the positive relation between surface roughness and albedo over an ice surface (Figure 5b) can be solely explained by the high-albedo snow patches (shown in Figure 4f) resulting from the snowfall events discernible from the occasional albedo increases during August (Figure 4a). In that respect, can the authors add a figure with the meteorological variables measured at the AWS shown in Figure 1? I think that in order to demonstrate that the albedo increases are only due to changes in surface roughness, the authors would need to show data without snow patches. Otherwise, it seems very plausible that the high-albedo snow patches resulting from the occasional summer snowfall events are responsible for the positive relation between surface roughness and albedo, during the ice melt period.

**Reply: Thank you for your comments and suggestions. In the revised manuscript, we have exclude patchy snow surface in bare ice surface analysis. Positive relationship was discerned through manual and automatic surface roughness and albedo analysis in Figure 5b of the revised manuscript. In the revised manuscript, we also add precipitation data and half-hour air temperature observations which help us analysis snow metamorphism in melting season and accumulation season.**

Period of snow melting. It is well known that snow albedo decreases due to snow metamorphism and large grain sizes, which affects the light scattering. The increasing grain size can have an effect on the surface roughness, but the latter is also explained by differential melting of the surface. From lines 505-507, the authors seems to suggest that the lowering of the albedo is caused only by the increasing surface roughness.

**Reply: Thanks for your comments. We neglect the snow metamorphism process in snow albedo and roughness analysis in the manuscript since we lack snow grain size observations in 2018. In the revised manuscript, we have revised and add snow metamorphism in melting season and accumulation season and connect surface roughness with snow grain size process with new data collected in summer of 2020 in Figure 9 of revised manuscript. In the revised manuscript, we have discussed the connection of snow specific surface area (SSA), snow grain size, and surface roughness evolution in melting season and accumulation season.**

"Physically-based" parameterizations and their potential for large-scale albedo estimates

The statistical relations between albedo, surface roughness and concentration of LASs presented by the authors are described as "physically-based". Although it is interesting that the authors include these variables in an albedo parameterization, these equations are not "physically-based", because they are not derived from a fundamental principle or process (e.g. energy conservation) and their parameters (obtained from a scatter plot) cannot be directly measured in the field.

**Reply: Thanks for your comments. We revised 'physically-based' as 'surrogate variables'. In the revised manuscript, we add more physically based analyzed of snow metamorphism, snow surface roughness and albedo. As Jonsell et al. (2003) has expressed that the physical properties of uppermost surface ice layer, including its density, crystal structures, surface roughness, impurity content and stratification and SSA, are subject to continuous changes as the melt season proceeds. Changes in characteristics of this layer are mainly responsible for the observed variability in ice albedo at any given site (Dadic et al., 2013; Jonsell et al., 2003). The surface roughness as a surrogate variables which related with snow or ice surface characteristics.**

In the abstract and conclusions, the authors suggest that albedo can be retrieved from surface roughness at large scales. However, this seems unrealistic because the nec- essary high-resolution DEMs to derive surface roughness can only be obtained by sensors carried by Unmanned Aerial Vehicles (UAVs), which are only used at the glacier- scale. Are the authors referring to that scale? Using satellite products, albedo is much easier to retrieve than surface roughness at regional scales.

**Reply: Thanks for your comments. High-resolution DEM data can be acquired by unmanned aerial vehicle photogrammetry or terrestrial laser scanning. Right now these methods can only applied at glacier scale. We have revised the large scale as glacier scale in the revised manuscript.**

LAIs sampling

More details are needed regarding the LAIs sampling. How did you calculate the vol- ume of each sample? Is that the volume of the stainless steel spoon mentioned in line 150? How did you make sure that the volume was the same in every collected sample? Can you include a picture of the spoon? How did you sampled sites with a rough surface, such as one with cryoconites? Did you take the samples at the top or the bottom?

**Response: We revised and add more details about sampling of snow and ice surface roughness.**

**The sample volume of snow or ice is not same. The sample volume depend on surface roughness and LAIs concentration. For example, if the surface is smooth and clean snow, we sampled 3 cm depth with an area of 20*20cm. If the snow is thin and underneath is dirty ice. The collected snow sample and the dirty ice layers. For the smooth ice surface, we collect surface 2-3cm ice. Most of the LAIs basically concentrated at surface and the underneath ice is looks clean. For rough ice surface, the sample depth depend on the surface roughness. The cryoconite holes usually 6-7cm deep. The collect sample could be 9 cm. Bellow this depth, the ice is looks clean.**

In this study, we used a 20*20cm*10cm nonrust steel box without cover and bottom to sample snow or ice. The sample procedures are: we put the steel box over sample area. Excavate the snow or ice surrounding the steel box to the depth we needed and keep the ice or snow in the steel box intact. After then, we lift the steel box, and sampled the snow or ice sample by a flat bottom shovel. Most of the time, ice is very hard, we use chisel to chip the ice.

[Figure]

**Figure 1 sample procedure in field.**

Paper structure and English language

Please improve the paper structure. I have the following suggestions: - Section 2.1 is neither data nor methods. Please add a new section called "Study Site". - Sections 3 and 4 consist of results. I suggest including these sections in a traditional "Results" section. - Sections 3 and 4 have a confusing structure, partly because the titles are ambiguous and repetitive. - The numbering of sections 7 and 8 is wrong. They should be 5 and 6.

**Response: Thanks for you excellent suggestions, we ill revised based on your suggestions.**

I have provided many suggestions to improve the use of English language, but I think that the authors should perform a full revision of the text.

**Response: Thanks for your suggestion, we will revised based on your suggestions and comments and referee 1's suggestions.**

SPECIFIC COMMENTS

11: "Fluctuations in surface albedo are due primarily to variations in micro scale surface roughness ($\xi$) and light-absorbing impurities (LAIs) in this region." I guess you mean fluctuations of bare ice albedo, because snow albedo variations are very large

and are due to metamorphism. See my main comment 1a.

**Reply: Thanks for your suggestions and comments, we have revised it accordingly.**

21: "ξ could explain 68% of snow surface albedo and 38%of ice surface albedo variation in melt season." When you write that surface roughness explains 68% of the snow albedo variation, I think that you need to explicitly state that this is a statistical analysis, because the physical explanation for snow albedo variations is snow metamorphism and an increasing grain size. See my main comment 1.

**Reply: Thanks for your suggestions and comments, we have revised it based on your suggestions.**

37: "According to Hock (2005), on average it accounts for over 70% of the net energy input to glacier surfaces." You should mention that that number was obtained for a particular glacier (Storglaciären). In any case, what do you mean exactly by "net energy input"? Net = input - output. Table 1 in Hock (2005) shows that incoming longwave radiation (L_in) is the largest energy input. In general, L_in and S_net are the largest energy inputs to glaciers at daily, and longer, time scales (Ohmura, 2001).

**Reply: Thanks for your comments, we have revised it accordingly.**

78-79: "poorly investigated, and snow surface albedo parameterization methods based on surface roughness are rarely reported." However, the relation between snow surface albedo and grain size has been largely analyzed. Please see my main point 1a for snow surfaces.

**Reply: Thanks for your comments, we have revised this part as 'The relation between snow surface albedo and grain size has been largely analyzed. In contrast, the relationship between snow surface roughness and its albedo has been poorly investigated, and snow surface albedo parameterization methods based on surface roughness have rarely been reported on.'**

86: "Surface roughness structures developed during melt season such as crevasses, cyroconite holes, can increase ice surface albedo by hiding LAIs from direct sunlight have been widely reported." Although, more than "increasing" surface albedo, I would say that they "prevent", or at least "delay", a further decrease, because they hide LAis that would otherwise reduce the area-average albedo.

**Reply: Thanks for your excellent suggestions, we have revised based on you suggestions. We have revised as 'Surface**

roughness features developed during the melt season such as crevasses and cryoconite holes can prevent a further decrease of ice surface albedo by hiding LAIs from the direct sunlight, and have been widely described (Lliboutry, 1964; Oerlemans, 1993; Bøggild et al., 2010; Takeuchi et al., 2014; Chandler et al., 2015; Takeuchi et al., 2018).'

229: How do you calculate the average value and uncertainty of albedo, surface roughness and LAIs concentration at a particular elevation?

**Reply: we have measured up and downward shortwave radiation 3 to 6 times (measured more times when weather conditions is cloudy or changed from sunny to cloudy). The data is quite variable for snow surface under cloudy weather. But it is not sensitive to weather for dirty ice surface.**

242: "LAIs decreased from $0.04\pm0.03$g/ cm3 at middle part to $0.003\pm0.002$g/ cm3 at higher elevations (Figure 3i)." This is based only in one sample at 4700 m, if you delete it, you don't have any trend. Did you try any other topographic parameter?

**Reply: we measured elevations, latitude and longitude. Slopes and aspect are not measured. Basically, the ice cap is flat. Aspect and slope changed not so great along the main flow line.**

247-249: "There was a much higher concentration of LAIs on the uncovered ice surface than snow surface. As a consequence, albedo tended to be low on the ice surface and higher on snow-covered surfaces." This is not only explained by the LAIs. Ice albedo is usually lower than snow albedo.

**Reply: Thanks for your suggestions, we revised accordingly. We have revised as 'Measurements indicated that albedo increased from $0.14\pm0.03$ at the terminals to $0.21\pm0.05$ at the top of the ice cap (Fig. 3d).'**

378-379: "Although we do not have tandem surface roughness and grain size observations, the evolution of surface roughness calculated at 1mm resolution in snow covered period should quite similar with fluctuation trend of grain size evolution." Certainly, you have a correlation between grain size and surface roughness, but surface roughness is not only explained by grain size. Differential melting of the surface could be also important.

**Reply:    Thanks for your excellent suggestions, we have revised and add the differential melting here accordingly.**

**We have revised as 'The snow surface roughness follows an increasing trend due to snow grain particle metamorphism and differential melting of the surface. Rapid grain growth corresponding with decreasing SSA and increasing of surface**

roughness. There is often a concurrent rise in albedo due to the blanketing of old snow by fresh snowfall. New fresh snow surface has reduced surface roughness by preferential deposition, smaller grains and larger SSA, which increase the albedo. Fig. 9 shows the snow grain size, fresh and old snow surface roughness measured on 14 July 2020 at the top (4820 m) and lower part (4623 m) of the ice cap. We could find old snow surface roughness is larger than fresh snow surface due to its larger snow grain size and greater DEM variation. Due to preferential deposition, more fresh snow deposited on the pits than peaks (Fig. 9d). The snow grain size at the pits are smaller than the peaks (Fig. 9d). The DEM variation are also larger at the peaks than pits due to its larger snow grain size (Fig. 9f). The surface roughness fluctuation synchronously happened with SSA and grain size variation. Although we do not have tandem surface roughness, grain size and SSA observations, the evolution of surface roughness calculated at 1mm resolution in the snow-covered period should be similar with the fluctuation trend of grain size evolution.'

423-424: ". . .a significant positive relationship rather than a negative relationship was established over ice surface based on manual and automatic measurements. We expect it is related with abundant LAIs over ice surface at the August one ice cap." This might be explained by the snow patches. See my main comment 1.

**Reply: Thanks for your suggestion. In the revised manuscript, we analyzed the pure ice surface without inclusion of patchy snow surface. More detailed results have provided in the revised manuscript of Figure 5b about bare ice surface roughness and albedo.**

448-450: "The performances of the establishes albedo methods either based on surface roughness, LAIs or effective LAIs concentration shows a great improvement over the assumption of a constant mean ice albedo or surrogate variables, such as air temperature, accumulated melt and elevation." I don't agree that surface roughness is much better than air temperature or melt to parameterize albedo. Please see main comment 1b.

**Reply: We will take it more carefully. In here, we try to say surface roughness and LAIs as surrogate variables show improvement over ice surface. In the revised manuscript, we have add new references which talk about ice surface albedo parameterizations such as Brock (2000) or Jonsell et al. (2003). The upmost ice surface characteristics of this layer are mainly responsible for the observed variability in ice albedo at any given site. The surface roughness and LAIs are surface properties which might more connect to surface albedo than air temperatures or melt amount.**

**We have revised it as 'The performance of the established albedo methods either are based on surface roughness in Fig. 5b or effective LAI concentration in Fig. 7b shows great improvement over the assumption of a constant mean ice albedo**

or surrogate variables, such as air temperature, accumulated melt and elevation (Brock et al., 2000). The physical properties of uppermost surface ice layer, including its density, crystal structures, surface roughness, impurity content and stratification and SSA, are subject to continuous changes as the melt season proceeds (Jonsell et al., 2003). Changes in characteristics of this layer are mainly responsible for the observed variability in ice albedo at any given site (Dadic et al., 2013; Jonsell et al., 2003).'

462-464: "Since surface roughness is dependent and sensitive to topography data resolution (Figure 8a), so which resolution is appropriate for snow and ice surface albedo estimation?" Was this part of the objectives? Are you writing about the DEM resolution? Please be more explicit in this section and present this analysis earlier in the text.

**Reply: Thanks for your suggestions, we have revised and introduced in introductions. We also revised here accordingly. In the earlier part, we have revised and add as 'In this study, we investigated the spatial and temporal variability of albedo, micro-scale surface roughness and LAIs, with the objective of better understanding and simulating surface albedo variability over the snow and ice surface of the August-one ice cap in the Qilian Mountains. First, we investigated the spatial distribution of surface roughness via manual photogrammetry, manual LAI samples and albedo over different altitudes along the main flow-line of the August-one ice cap during the 2018 melt season. The effects of micro-scale surface roughness and LAIs on glacier snow and bare ice albedo was investigated at different altitudes and times. Second, automatic photogrammetry of surface roughness and automatic observation of glacier surface albedo was conducted on the middle of the ice cap in the melting and accumulation season of 2018. The effect of surface roughness on glacier snow and ice albedo was also analyzed at the daily scale in the melt and accumulation seasons separately. Third, the surface roughness effect on snow and ice surface albedo was analyzed based on combined manual and automatic photogrammetry at different altitudes, times and resolutions. Forth, snow and ice surface albedo parameterization methods are established based on only surface roughness, LAIs or both LAI concentration and surface roughness in the melt season, separately. Finally, since the surface roughness is dependent on and sensitive to topography data resolution, we calculated surface roughness under different resolution from 0.4mm to 333mm to investigate its correlations with albedo at plot scale.**

**Appropriate resolution to surface roughness calculation was suggested for snow and ice albedo estimation, respectively. '**

496-498: "By using manual and automatic photogrammetric measurements of surface roughness, manual LAIs samples and measurement of broadband albedo at the August-one ice cap, we have a general understanding of the surface roughness that controls the albedo of snow and ice surface are quite different." Can you really conclude that surface roughness controls the albedo of snow and ice? See my main comment 1.

**Reply: Thanks for your suggestion, we have revised as 'surface roughness is a good surrogate variables can be used to parameterize surface albedo'. For snow surface, we have revised and add snow** metamorphism, **SSA, snow grain size which controls snow surface albedo evolution. For ice surface, we have exclude snow patchy in the bare ice surface roughness and albedo analysis.**

**More detailed have proved in revised manuscript.**

505-507: "For snow-covered surfaces, ice particle metamorphism and surface melting and refreezing induced grain size increasing synchronously happened surface rough- ness increasing, which induced decreasing snow surface albedo in melting season." The last part of the sentence suggests that is the increasing surface roughness that induces the snow albedo decrease.

**Reply: Thanks for your suggestion, we make mistake here. we have revised it. Snow metamorphism which could induced snow grain size increase, snow surface roughness increase, and albedo decrease. Snow surface roughness increase is not induces the snow albedo decrease.**

**We have revised it as 'For snow-covered surfaces, snow particle metamorphism and surface melting and refreezing induced an increase in grain size and decrease in SSA. Differential melting and snow metamorphism induced surface roughness increase. This process occurred synchronously with snow metamorphism. Increase in grain size and decrease in SSA induced decreasing snow surface albedo in the melt season.'**

SUGGESTED TECHNICAL CORRECTIONS

4: Please make the format for last names uniform (sometimes is uppercase and some- times not).

**Reply: Thanks for your suggestions, we have revised and revised and uniform the names.**

11: influence on melt

**Reply: we have revised it.**

16: the present study consisted of an intensive. . .

**Reply: we have revised based on your suggestions.**

19: the middle

**Reply: we have revised it, thanks.**

19: A detailed analysis indicates that. . .

**Reply: we have revised it accordingly.**

20: a positive linear

**Reply: revised.**

21: for snow and ice, respectively

**Reply: revised.**

22: consider -> considering

**Reply: we have revised accordingly.**

25: constant mean -> uniform

**Reply: we have revised as 'uniform' in the revised manuscript.**

33: The energy balance and resulting melt rates. . .

**Reply: we have revised it accordingly.**

33: the meteorological conditions

**Reply: we have add ' the ' in the revised manuscript.**

34: the physical properties

**Reply: we have add ' the ' in the revised manuscript.**

34: Delete ", which determine glacier melt process"

**Reply: Thanks for your suggestions, we have revised based on your suggestions.**

35: the melting of glacier -> glacier melt

**Reply: Revised.**

35: dominated-> controlled

**Reply: Revised.**

35: in the glacier surface -> at the. . .

**Reply: we have revised.**

36: Shortwave radiation is the main energy input causing snow and ice melt

**Reply: we have revised it based on your suggestions, Thank you.**

39: net radiation: do you mean net shortwave radiation?

**Reply: we have revised as net shortwave radiation.**

45: "Glacier albedo varies much more dramatically than other land covers" -> Albedo varies much more dramatically on glaciersurfaces than on other land covers.

**Reply: Thanks for your suggestions. We have revised it based on your suggestions.**

47: "constantly changing surface characteristics"-> I think that you should explicitly mention snow metamorphism here.

**Reply: We have revised based on your suggestions.**

47: the solar incidence angle

**Reply: we have revised It accordingly**

52: accelerate

**Reply: Revised.**

57: Simulations

**Reply: Revised.**

59-62: In general, I suggest improving the writing style by making sentences shorter and more fluid. Here, I would suggestsomething like: LAIs have decreased the surface albedo of glaciers in Qilian mountains to values as low as 0.13+-0.06 during the melting season (LIT).

**Reply: Thanks for your suggestions, we have revised it based on your suggestions.**

59: Please introduce the Qilian mountains before this sentence.

**Reply: Thanks for your suggestions, we have introduced the Qilian mountains in here accordingly.**

64: I think that you should start a new paragraph when you mention surface roughness.

**Reply: Thanks for your suggestions. We have start a new paragraph to introduce surface roughness.**

70: do you mean "first recognized by Kuhn"?

**Reply: Yes, here, we have revised it based on your suggestions and make more clear.**

71-72: In which way these field campaigns advanced the influence of roughness on albedo?

**Reply: Thanks for your question. In the revised manuscript, we will make more clear.**

75: Studies. Which ones?

**Reply:  Studies by Roujean et al. (1992),  O'Rawe (1991), and Larue et al. (2019). There we have revised and accordingly.**

75: indicated

**Reply: Revised.**

75: the inclusion

**Reply: revised.**

76: measurements of albedo? Sorry, this is not clear.

**Reply: Thanks for your question, we will revised it more clear.**

76: equations

**Reply: revised.**

77: and sastrugi

**Reply:revised.**

80: heterogeneous: Please explain better what do you mean by this.

**Reply: here we try to express the uneven distribution of LAIs induced differential absorbing of shortwave and meltingdifferences**. In the revised manuscript, we revised clearer.

80: of LAIs.

**Reply: revised.**

83-85: Please improve the structure of this key sentence.

**Reply: Thanks for your suggestions, we will revised clearer.**

85: the melt season,

**Reply: revised.**

85: crevasses and

**Reply: revised.**

86: the direct sunlight

**Reply: revised.**

88: Please introduce the ice cap more formally. Where is it? You can give the details in the next section, but give at least someindication here.

**Reply: Thanks for your suggestion, we introduce the ice cap and add the location here**

90: indicated

**Reply: revised.**

93: LAIs on ice

**Reply: revised.**

95: combined effects: This is very interesting, and difficult, how do you isolate both

effects? Please see my main comment 1.

**Reply: we are not trying to isolate surface roughness effect and LAIs effect in this study. Studies by Larue et al.(2020) have quantify the impact of surface roughness on albedo on clean snow surface. In this study, we find surface roughness affect ice surface albedo by hide LAIs from direct sunlight. For snow surface, the surface roughness do not affect LAIs concentration. It is very difficult try to differentiate surface roughness and LAIs effect especially in summer, since other factors such as water content, solar incidence angle, snow grain size, or snow specific area, snow metamorphism affect snow surface albedo.**

96: have been rarely

**Reply: revised.**

92: we investigate

**Reply: revised.**

93: better understand and simulate. I think that these are very wide and general objec- tives. Can you be more specific?

**Reply: thanks for your comments, we will revised based on general comments and make it more specific in the revised manuscript.**

106: at the daily scale

**Reply: revised.**

101: we investigate

**Reply: revised.**

104: on the middle

**Reply: revised.**

104: in melting season and accumulation season of 2018 -> in the melting and accu- mulation season of 2018.

**Reply: revised**

107-108: This is not clear at this point.

**Reply: we have revised and make it clearer.**

114: Study area is neither data nor methods. Create another section.

**Reply: Thanks for your suggestion, we have revised it.**

116: the Qilian Mountains

**Reply: revised**

120: May to September. Mention that this is summer.

**Reply: revised**

120: How short is summer?

**Reply: revised**

126: has been observed.

133: This section should be called data collection and image processing.

**Reply: Thanks, we have revised accordingly.**

137: In some parts of the text you write "we" and in others you write "researchers". Please make it uniform. I would suggest to use an active voice, i.e. "we".

**Reply: Thanks, we have revised as 'we' in the revised manuscript.**

142: several different locations. How many? Make a reference to Figure 1. You don't have any table in your article. It would be good to have one with the coordinates, number of measurements, etc.

**Reply: we have revised and add the number of measurements in the revised manuscript.**

142: the top of the ice cap

**Reply: revised.**

144: the micro scale topography over different altitudes

**Reply: revised**

145: Physical->You mean topographical in opposition to the turbulent heat fluxes pa- rameter (surface roughness length)?

**Reply: Here, we have revised as surface roughness. It is not surface roughness length.**

145: was estimated. How?

**Reply: The surface roughness is estimated based on equation (2). In the revised manuscript, we have revised and make it clearer.**

146: putting->placing a Kipp and Zonen CMP11 radiation sensor

**Reply: revised.**

152: in an oven

**Reply: revised.**

156: its operation

**Reply: revised.**

157: geo-reference: what do you mean?

**Reply: we have used a wooden control field with 8 control points on it as control frame to geo-referencing pictures to DEM data.**

159-160: "The photography was repeated at three hour intervals from 9:00 AM to 18:00 AM, UTC+7 time." But later you work with daily means, don't you?

**Reply: we select one of the four sets of photos and merged to produce a 1mm×1mm resolution surface topography. Not all four sets was used to produce surface DEM data, since some of the data sets was affect by fog or too dark to be applied for DEM data.**

159: how many different locations?

**Reply: we have revised and add the number of 37 locations in the revised manuscript. In the revised manuscript we will include new observations at 20 locations carried out in 2020 over snow surfaces.**

168: How did you make sure that the frame stayed horizontally?

**Reply: we put the frame over ice or snow surface. The frame is horizontally with ice or snow surface. It make the generated DEM detrended.**

185-189: Repetition

**Reply: revised.**

190: that surface roughness

**Reply: revised**

190: by directly affecting

**Reply: revised**

195: from the aerodynamic surface roughness parameter developed by Lettau (1969)

**Reply: revised**

196: except that we changed the silhouette area facing upwind to. . .

**Reply: revised**

203: Is V the volume of the spoon?

**Reply: Yes, V is calculated based on the spoon, the spoon is a 20\*20cm\*10cm box without cover and bottom. The sample area is 20\*20cm, the depth is measured ever time base on the insert depth.**

210: We assumed that

**Reply: revised**

220-224: This sounds like methods to me. Is this the Results section? See my minor comments.

**Reply: we have revised it and add to the methods part.**

225: You also have temporal variability here, not only spatial. Please change the title.

**Reply: revised.**

226-227: Again, I suggest shorter and more fluid sentences, something like:

We found a patchy snow cover with many cryoconite holes on the glacier terminus (~4600 m) at the date of the first field trip (July 12).

**Reply: thanks for your suggestion, we have revised accordingly.**

230: Be consistent with the digits, i.e. 5.49 cm = 5.5 cm.

**Reply: revised**

244-245: On July 12, surface. . .

**Reply: revised.**

246-247: the transient snowline retreated up-glacier.

**Reply: revised**

247-248: On what plot do you base that sentence

**Reply: We have revised and differentiate snow and ice covered surface in the revised manuscript in Figure 3. In the revised manuscript, we could find snow covered surface LAIs concentration is much lower than ice surface.**

**248: bare ice surface than on snow.**

**Reply: revised**

249-250: Albedo didn't really increased over time.

**Reply: we try to express the increasing trend of albedo from terminal to top. The albedo is not increase over time. We will revised it and make clearer.**

251: minimal albedo: Do you mean the season minimum?

**Reply: The minimal is not precise, we have revised. The albedo in August is very low when the ice cap if all bare ice from terminal to top**

251: "but it also did not increase"-> and it did not show a clear trend with altitude

**Reply: revised.**

259: This is almost the same title as the previous sub-section.

**Reply: we have revised as Automatic observation of surface roughness and albedo**

262: October 17

**Reply: revised.**

270: mostly bare ice with occasional snowfall events

**Reply: revised**

285: was mainly induced

**Reply: revised**

259: The title of this section is very similar to that of section 3.

**Reply: we have revised and differentiate with section 3.**

298-300: Snow surface albedo and the corresponding surface roughness are analyzed using. . .

**Reply: revised.**

301: functions

**Reply: revised**

301: observations

**Reply: revised**

304: Please use the same coefficient in the text and the figure (either r2 or r).

**Reply: revised**

321: What is a patchy ice cover? A bare ice surface covered by snow patches or by debris patches?

**Reply: we have revised as snow patches**

322-323: This is not clearly written. Better say that the patchy snow cover is the limit between the two periods that you identified.

**Reply: Thanks for your advances, we have revised accordingly.**

324: the relationship

**Reply: revised**

324-325: Delete "without consider surface roughness effect over LAIs concentrations."

**Reply: revised.**

328: Do you mean 5d?

**Reply: we make mistakes here, we have revised.**

328: manually

**Reply: revised.**

331: power function, do you mean linear function?

**Reply: revised.**

333: that considering

**Reply: revised**

334: . . .albedo during the snow period.

**Reply: revised**

334: It indicates

**Reply: revised**

365-373: All these lines can be removed, moved to the introduction, or condensed.

**Reply: Thanks for your suggestion, we have revised accordingly.**

369: Meltwater

**Reply: revised**

378: should be quite similar

**Reply: revised**

380-383: This should be mentioned earlier in the text.

**Reply: revised.**

386: it means that

**Reply: revised**

388: LAIs are another critical factor affecting snow albedo

**Reply: revised.**

392: "increasing surface roughness decrease albedo" But, only in your snow melting

Period

**Reply: revised.**

396-400: This should be in results.

**Reply: revised**

400: What is the explained variance of each variable?

**Reply: we have add the explained variance of surface roughness and LAIs in the revised manuscript.**

402: During the accumulation season

**Reply: revised.**

403: Please explain better the observed snow metamorphism in the accumulation period

**Reply: Thanks for your comments, we have revised and add more citetations about snow metamorphism in the accumulation period in the revised manuscript. The accumulation period snow surface roughness evolution is induced by dry snow metamorphism and blowing snow events formed** sastrugi.

403: During the accumulation period

**Reply: revised.**

413-421: This belongs to the introduction

**Reply: Thanks for your suggestion, we have revised accordingly**

430-433: Please re-write this sentence in a clearer way.

**Reply: Revised, we have revised and re-write it clearer.**

437: over smooth ice -> than smooth ice?

**Reply: we have revised 'over' as 'than' .**

447: are based

**Reply: revised.**

TABLES

There are no tables in the article. At least one summarizing the distributed measure- ments would be useful.

**Reply: Thanks for your suggestion, we have add** Table **1** to **summarize snow and ice surface albedo parameterization methods**

Table 1 Accuracy of snow and ice surface albedo parameterization

| Surface types | Equations | $R^2$ | RMSE | P< | Albedo range |
|---|---|---|---|---|---|
| Snow surface for melting season | $\alpha = 0.234 C_{LAIs}^{-0.1415} - 0.02098\xi^{1.226}$ | 0.89 | 0.0536 | 99.99 | 0.85-0.22 |
| | $\alpha = 0.1012 \times \xi^{-0.338}$ | 0.68 | 0.0852 | 99.99 | 0.92-0.22 |
| | $\alpha = 0.634 \times C_{LAIs}^{-0.342}$ | 0.84 | 0.0603 | 99.99 | 0.70-0.20 |
| Snow surface for accumulation season | $\alpha = 0.4613 \times \xi^{-0.093}$ | 0.25 | 0.0608 | 99.9 | 0.92-0.65 |
| | $\alpha = 0.0585 \times C_{LAIs}^{-0.225}$ | 0.35 | 0.0659 | 99.9 | 0.10-0.38 |
| Ice surface | $\alpha = 0.0374 \times C_{\xi}^{-0.26}$ | 0.63 | 0.0501 | 99.99 | 0.10-0.38 |
| | $\alpha = 3.3 \times \xi + 0.1212$ | 0.38 | 0.0503 | 99.9 | 0.12-0.38 |

FIGURES

Figure 1: Please add an inset showing the ice cap and the Qilian Mountains in the Tibetan Plateau.

**Reply: Thanks for your suggestions, we will revised based on your suggestions, and add the ice cap location and the Qilian Mountains**

Figure 2: Why is important to have rain or fair weather? Provide a brief explanation.

**Reply: Thanks for your suggestions, we find rough ice surface developed in cold and sunny day, and warm and cloudy or rain day favors smooth and dark ice surface. We have explain it in the revised manuscript of Figure 2 captions.**

Figure 3: I would place altitude in the x-axis. Elevation should not be a dependent variable.

**Reply: Thanks for your suggestion, we will revised based on your suggestions, we also differentiate snow and ice surface in the revised manuscript based on referee 1's suggestions.**

Figure 3: In the text of this figure, you write several times that there is a relation with the altitude, but based on your data, this is true only for 3a and 3b.

**Reply: The relation with altitude is established based statistical significance.**

Figure 4: Make a uniform tick spacing in the x-axis. Example, 1-07 15-07, etc.

**Reply: Thanks, we will revised based on your suggestion.**

Figure 4: From this plot is evident that there were 3 or 4 snowfall events in August. See my main comment 1.

45   **Reply: Yes, there are snowfall events in August, we have add these short period snow surface roughness and albedo as snow covered surface rather than bare ice cases. During this snowfall period, the snow melt quickly, and snow surface roughness also changes fast.**

Figure 4: Add a line in panel a to show when were the pictures taken.

**Reply: Thanks for your suggestions, we have add the panel date in the revised manuscript.**

50   Figure 4: Show the dates of the pictures in the pictures.

**Reply: revised**

Figure 5a: Note that for low albedo (<0.4) you can get very different surface roughness (1-6 cm).

**Reply: Low albedo for snow surface are all patchy snow covered surface. Patchy snow shows great surface roughness differences. The thick snow patches usually shows larger surface roughness, and thin snow patches**

55   **shows smaller surface roughness.**

Figure 5: Consider the use of different markers (in a-b and c-d) to reduce the number of panels. As in Figure 4s.

**Reply: Thanks for your advices, we will revised based on your suggestions.**

Figure 5: I would strongly suggest using only one paragraph to describe one figure. This is a big help for readers.

**Reply: Thanks for your suggestion, we will adopt your advice.**

60   Figure 5: How many different sites are included in the manual measurements shown in this figure?

**Reply: we have 37 sites of surface roughness observations. In the revised manuscript, we have add these information in the revised manuscript.**

Figure 6: Can you merge both plots in one panel by using different markers? A loga- rithmic scale might be useful.

**Reply: Thanks for your suggestions, in the revised manuscript, we have revised and talk snow and ice surface**

65   **separately in different figures. It will be clearer.**

Figure 7: Apart from the high albedo for low surface roughness, there is a poor relation between the two variables. Maybe move to the supplementary?

**Reply: In the revised manuscript, we have discussed snow metamorphism in cold season. I think it is an interesting topic which could be a complementary for melting season.**

70   Figure 8: What is the difference between these pictures and those shown in Figure 1s?

**Reply: In Figure 8, we calculated surface roughness under different resolutions. Figure 8 shows rough surface are more sensitive to resolution than smooth surface. It means ice surface are more sensitive to snow surface.**

**Figure 1s calculated surface roughness at 1mm resolution for all manual and automatic observations.**

**In Figure 8, and figure 10, we try to find under what resolution, the calculated surface roughness have more**

75 **significant statistical significance with albedo. We find finer resolution of than 50 mm and 100 mm resolution is recommended for ice and snow surface roughness calculations.**

Figure 2sa-3sa: How many different sites did you use to build this plot?

**Reply: we have 37 manual observation sites. The automatic observation in melt season include 65 days. We applied 63 plots of automatic observations. We have revised and include these information in the revised**
80 **manuscript.**

Figure 4s: This is a very nice figure, because it shows both periods in the same plot. Can you move this figure, or a similar one, to the main text?

**Reply: Thanks for your suggestion, we have revised and add in the main text of Figure 5 accordingly.**

**Effects of surface roughness and light-absorbing impurities on glacier surface albedo, August-one ice cap, Qilian Mountains, China.**

Junfeng Liu[1], Rensheng Chen[1,2], Yongjian Ding[1], Chuntan Han[1], Yong Yang[1], Zhangwen Liu[1], Xiqiang Wang[1], Shuhai Guo[1], Yaoxuan Song[1], Wenwu Qing[3]

90   1.   Qilian Alpine Ecology and Hydrology Research Station, Key Laboratory of Ecohydrology of Inland River Basin, Northwest Institute of Eco-Environment and Resources, Chinese Academy of Sciences, Lanzhou, China.

2.   College of Urban and Environmental Science, Northwest University, Xi'an, China

3.   Lanzhou University, Lanzhou, China.

Correspondence to: Rensheng Chen (crs2008@lzb.ac.cn)

**Abstract:** Surface albedo is the main influence on  melt of glaciers in the Qilian Mountains. Fluctuations in surface albedo are due primarily to variations in the physical properties of the uppermost surface snow layers' specific surface area (SSA), surface roughness ($\xi$) and light-absorbing impurities (LAIs). However, combined $\xi$ and LAIs effects on glacier surface albedo are rarely studied, and surface

100   roughness is rarely considered in the albedo parameterization methods in this region. The present study consisted of an intensive photogrammetric survey of glacier surface roughness, LAI samples and albedo observations along the main flow-line of the August-one ice cap during the 2018 melt season. Automatic photogrammetry of surface roughness and automatic observation of glacier surface albedo was conducted in the middle of the ice cap in 2018. A Detailed analysis indicates

105   that a negative power function and a positive linear relationship exist between $\xi$ and albedo for snow and ice , respectively. Statistical analysis indicates $\xi$ was able to explain 68% of snow surface albedo and 38% of ice surface albedo variation in the melt season. Effective LAI concentrations ($C_\xi$) calculated by considering the $\xi$ effect on LAI deposition account for more than 63% of albedo variation at the ice surface. Using either $\xi$ or $C_\xi$ to estimate ice surface albedo would be a great improvement over some

110   currently used parameterization methods, such as assuming a uniform ice surface albedo. A finer resolution smaller than 50mm and 100mm, is recommended for ice and snow $\xi$ calculations, which can explain more albedo variation than coarse resolutions above it. With advances in topographic surveys

to improve the resolution, extent and availability of topographic datasets and surface roughness, appropriate parameterizations of albedo based on ξ have exciting potential to be applied over  glacier scale.

**Keywords**: Albedo; Snow specific surface area; surface roughness; Light-absorbing impurities; August-one ice cap; Qilian Mountains.

**1. Introduction**

The energy balance and resulting melt rates at the glacier–atmosphere interface is controlled by the meteorological conditions above the glacier and by the physical properties of the glacier itself (Hock, 2005). In general, glacier melt is dominated by the air temperature and net shortwave radiation in the glacier surface. Shortwave radiation is the main energy input causing snow and ice melt (Male and Granger, 1981; Gardner and Sharp, 2010). According to Hock (2005), on average it accounts for over 70% of the net energy input to Storglaciären glacier surfaces. Recent surface energy budget (SEB) studies undertaken on glaciers in the Qilian Mountains show that in this region, melt season  shortwave radiation accounts for 80–92% of glacier melt in ablation zones (Chen et al., 2007; Sun et al., 2012; Qing et al., 2018; Sun et al., 2018) and over 70% of glacier melt in accumulation zones (Sun et al., 2014; Wu et al., 2016). Hence the rate of glacier melt is largely determined by the surface albedo in the melt season. It is of great importance to study albedo if we are to improve the accuracy of the energy budget estimate and meltwater runoff in this region.

Albedo varies much more dramatically on glacier surfaces than other land covers, within the range of 0.1 for dirty wet ice to 0.9 for fresh dry snow (Warren and Wiscombe, 1980; Bøggild et al., 2010). Its variability is attributable to constantly changing surface characteristics mainly induced by snow metamorphism, as well as clouds and the solar incidence angle (Mellor, 1977; Carroll and Fitch, 1981; Konzelmann and Ohmura, 1995; Cutler and Munro, 1996; Oerlemans and Knap, 1998; Brock et al., 2000; Kylling et al., 2000; Gardner and Sharp, 2010; Zhuravleva and Kokhanovsky, 2011; Goelles and Bøggild, 2017). Light-absorbing impurities (LAIs)

at the glacier surface, consisting of wind-blown dust particles, carbonaceous particles and colored organic matter produced by glacial organisms, can accelerate the snow aging process, and reduce glacier albedo greatly in its visible spectrum (Oerlemans, 1993; Brock et al., 2000; Bøggild et al., 2010; Hadley and Kirchstetter, 2012; Cook et al., 2017), which enhances the melting of snow and ice (Bond et al., 2013; Flanner et al., 2007; Oerlemans et al., 2009; Xu et al., 2009; Gabbi et al., 2015). As the snow melts, insoluble LAIs are retained at the snow surface, so concentrations of LAIs in the surface snow increase with snowmelt, further reducing snow albedo (Doherty et al., 2013). Glaciers in the Tibetan Plateau are greatly affected by LAIs (Takeuchi and Li, 2008; Jiang et al., 2010; Yang et al., 2011; Wang et al., 2012; Zhang et al., 2013; Liu et al., 2014; Ji, 2016; Li et al., 2018; Qing et al., 2018; Sun et al., 2018). Simulations of the effect of LAIs on the albedo of Tibetan glaciers showed that LAIs had a contribution of 34% to the albedo reduction in the late spring (Ming et al., 2012). Qilian Mountains are located on the northeastern edge of the Tibetan Plateau, LAIs have decreased the surface albedo of glaciers in Qilian Mountains to values as low as $0.13\pm0.06$ during the melting season~~For glaciers in the Qilian Mountains, the measured daily mean albedo decreased to ~ 0.07 to ~0.13 due to the effect of LAIs on four glaciers observed during the melt season~~ (Jiang et al., 2010; Liu et al., 2014; Qing et al., 2018; Sun et al., 2018). Snow surface roughness is a measurement of the variability of surface microtopographic features (Fassnacht et al., 2009a). It is a function of crystal type, deposition conditions, and metamorphism and temperature history (Fassnacht et al., 2009a). Snow surface roughness is an important factor for the scattering of light and thereby related to the surface albedo (Warren, 1982; Leroux and Fily, 1998; Warren et al., 1998; Mishchenko et al., 1999; König et al., 2001; Arnold and Rees, 2003; Kokhanovsky and Zege, 2004; Zhuravleva and Kokhanovsky, 2011; Larue et al., 2019). The importance of surface roughness regarding snow was first recognized by Kuhn (1974, 1985), who pioneered the study of its effect on the bidirectional reflectance distribution function. The theoretical influence of snow roughness on albedo has been advanced by several measurement campaigns (Grenfell et al., 1994; Warren et al., 1998; Zhuravleva and Kokhanovsky, 2011). Several theoretical models have also been developed in order to quantify the snow roughness effect on radiative characteristics (O'Rawe, 1991; Roujean et al., 1992;

165  Larue et al., 2019). Studies  indicated that the inclusion of surface roughness in radiative transfer equations has improved agreement with measurements of albedo for macro-scale snow surface features such as suncups, penitents and sastrugi (Pfeffer and Bretherton 1987; Leroux and Fily, 1998; Hudson and Warren, 2007; Kuchiki et al., 2011; Lhermitte et al., 2014). The relation between snow surface albedo and grain size has been largely analyzed. In contrast, the relationship between snow surface roughness and its

170  albedo has been poorly investigated, and snow surface albedo parameterization methods based on surface roughness have rarely been reported on.

As the ice melts, the distribution of LAIs across ice surfaces is heterogeneous (Hodson et al., 2008; Li et al., 2018), leading to differential absorption of shortwave radiation at a range of spatial scales. The heterogeneous distribution of LAIs results in the roughening of the ice surface, a process that enhances

175  turbulent heat exchange across the atmospheric boundary layer–ice interface. Furthermore,  a  roughening surface hides LAIs from direct sunlight and increasing ice surface albedo. Surface roughness features developed during the melt season such as crevasses and cryoconite holes can prevent a further decrease of  ice surface albedo by hiding LAIs from the direct sunlight, and have been widely described (Lliboutry, 1964; Oerlemans, 1993;

180  Bøggild et al., 2010; Takeuchi et al., 2014; Chandler et al., 2015; Takeuchi et al., 2018). The August-one ice cap is located in the middle of the Qilian Mountains on the northeastern edge of the Tibetan Plateau. Successive time-lapse photography of the August-one ice cap indicates that the ice surface has become darker due to the accumulation of LAIs during the melt season (Qing et al., 2018). Larger melt rates may have caused an increase in the melt-out of LAIs contained in the ice. A study at the August-one ice cap

185  has indicated that spatial and temporal surface roughness is variable during the melt season (Liu et al., 2020). This surface roughness could directly affect the concentrations of LAIs over ice surface roughness and furthermore affect the ice surface albedo. Although there has been extensive research focusing on quantifying the impact of LAIs  on ice and snow to understand the relationship between LAIs and albedo reduction (Aoki et al., 2011; Painter et al., 2012; Ginot et al., 2014; Kaspari et al., 2014),  surface

190  roughness and LAI effects over dirty ice surface albedo have  been rarely investigated together. Ice

albedo parameterization methods have been rarely established by considers both surface roughness and LAIs.

In this study, we investigated the spatial and temporal variability of albedo, micro-scale surface roughness and LAIs, with the objective of better understanding and simulating surface albedo variability over the snow and ice surface of the August-one ice cap in the Qilian Mountains. First, we investigated the spatial distribution of surface roughness via manual photogrammetry, manual LAI samples and albedo over different altitudes along the main flow-line of the August-one ice cap during the 2018 melt season. The effects of micro-scale surface roughness and LAIs on glacier snow and bare ice albedo was investigated at different altitudes and times. Second, automatic photogrammetry of surface roughness and automatic observation of glacier surface albedo was conducted at on the middle of the ice cap in the melting season and accumulation season of 2018. The effect of surface roughness on glacier snow and ice albedo was also analyzed on at the daily scale in the melt and accumulation seasons separately. Third, the surface roughness effect on snow and ice surface albedo was analyzed based on combined manual and automatic photogrammetry at different altitudes, times and resolutions. Forth, snow surface and bare ice surface albedo parameterization methods are established based on only surface roughness, LAIs or both LAI concentration and surface roughness in the melt season, separately. Finally, since the surface roughness is dependent on and sensitive to topography data resolution, we calculated surface roughness under different resolution from 0.4mm to 333mm to investigate its correlations with albedo at plot scale. Appropriate resolution to surface roughness calculation was suggested for snow and ice albedo estimation, respectively.

2. Study area

2. Data and methods

The August-one ice cap is located in the middle of the Qilian Mountains on the northeastern edge of the Tibetan Plateau (Fig 1a, 1b). The glacier is a flat-topped ice cap that is approximately 2.3 km long and 2.4 km$^2$ in area. The elevation ranges from 4550 to 4820m a.s.l. (Guo et al., 2015). The ice cap experiences

westerly winds, and is characterized by a typical continental climate with dominant precipitation in warm season from May through September. Mean monthly air temperature is > 0 °C from June through August at 4800 m a.s.l. in summer. Moreover, the Badain Jaran and Tengger deserts are located to the north, the Qaidam Basin lies to the south of the Qilian Mountains. Prevailing winds send enormous amounts of dust particles onto the glacial surface in this region (Kreutz, 2007; Dong et al., 2014a; Dong et al., 2014b). Annual dust deposition in the western mountain areas spans a range of 143.8–207.6 µg cm$^{-2}$yr$^{-1}$ (Dong et al., 2014b). Influenced by climate warming and continuous accumulation of LAIs, the entire August-one ice cap surface darkens during the melt season and no accumulation area has been observed for the last 6 years based on mass balance observations (Liu et al., 2020).

[Figure]

Figure 1. Location of ice cap and study sites. (a) Location of the August-one glacier. (b) Locations of the

**3. Data collection,  image processing and surface roughness and effective LAI calculation**
* * *
~~The August-one ice cap is located in the middle of Qilian Mountains on the northeastern edge of the Tibetan Plateau (Fig 1a, 1b). The glacier is a flat topped ice cap that is approximately 2.3 km long and 2.4 km² in area. The elevation ranges from 4550 to 4820m a.s.l. (Guo et al., 2015). The ice cap experiences westerly winds, and is characterized by a typical continental climate with dominant precipitation from May through September. Summer is short, and mean monthly air temperature is > 0 °C from June through August at 4800 m a.s.l. Moreover, the Badain Jaran and Tengger deserts are located to the north; the Qaidam Basin lies to the south of the Qilian Mountains. Prevailing winds send enormous amounts of dust particles onto the glacial surface in this region (Kreutz, 2007; Dong et al., 2014a; Dong et al., 2014b). Annual dust deposition in the western mountain areas spans a range of 143.8–207.6 µg cm⁻²yr⁻¹ (Dong et al., 2014b). Influenced by climate warming and continuous accumulation of LAIs, the entire August-one ice cap surface darkens during the melt season and no accumulation area has been observed for the last 6 years based on mass balance observations (Liu et al., 2020).~~

**3.1 Data collection and image processing**

3D photogrammetry has been widely used to measure micro-scale surface topography (Fassnacht et al., 2009a; Fassnacht et al., 2009b; Irvine-Fynn et al., 2014; Smith et al., 2016; Miles et al., 2017; Quincey

et al., 2017; Fitzpatrick et al., 2019). In this study, manual close-range photogrammetry was performed using a portable aluminum square frame delineating a 1.1m×1.1m plot.  We made high-definition photographs of the glacier surface within the frame. Photographs were taken at a distance of ~1.6 m, covering an area of ~1.75 m². At each survey site, 7 to 15 photographs were taken by surrounding the portable aluminum square. From each cardinal direction of aluminum square, we took two or three photographs to cover the target area. Rough ice surface such as cryoconite holes required the taking of additional photographs close to nadir angle. The camera used was an EOS 6D 50mm, with a fixed focal lens and an image size of 5472×3648 pixels. The f-stop was fixed at f 22 with an exposure time from 1/25 to 1/125 s. This was done for 37 different locations that ranged from the glacier terminals to the top of the ice cap. These transits were performed on 12 and 25 July, and 28 August, during the 2018 melt season. The August-one ice cap is a small ice cap without obvious surface movement. The cap surface shows no relevant morphology. Structures such as crevasses, cracks and seracs have not formed. Channels account for only a small portion of the glacier surface. In this study we selected all surface types, including smooth snow-covered surface, partial snow-covered surface, cryoconite holes, weathering crust and smooth ice surface, but not water channels. These photographs allowed  us to calculate the micro-scale topographic over different altitudes.  Topographic surface roughness was estimated based on equation (1) and these micro-scale topography data.

After manual photogrammetry and the removal of the aluminum square, a Kipp & Zonen CMP11 albedometer, consisting of two identical pyranometers, measured the incoming global solar and the radiation reflected from the surface below. Albedo is the ratio of the two irradiances. The Kipp & Zonen CMP 11 pyranometer is sensitive to radiation in the wavelength range of 0.285 to 2.8μm. The albedometer was mounted on a camera tripod 1m above the glacier surface and three readings were made at each point on a horizontal plane by means of a spirit bubble. Albedo was measured within 3h of local solar noon, to minimize the effects of diffuse radiation and high zenith angles.

Surface LAIs were sampled in a smaller 20cm×20×10 cm nonrust steel box without cover and bottom to sample snow or ice inside the aluminum frame (Fig. S1 in the Supplement), after manual

photogrammetry and albedo measurements had been taken. The samples were collected using a stainless steel spoon from the upper 0–9 cm surface at each site and stored in a transparent plastic ziplock bag. All collected samples were subsequently transported to the field laboratory and filtered. The filters were dried in  an oven at 50 °C for 24 h to eliminate the water vapor. The mass of insoluble LAIs in the snow or ice samples were calculated by weighing the filter using a microbalance (accuracy: 0.01g).

A station for automatic close-range photogrammetry was set up on a relatively flat surface in the middle of the ice cap (4700m, 98° 53.4′ E, 39° 1.1′ N. See Fig. 1), and began its operation on 12 July 2018. A 1.5m×2m plot was marked off by a wooden frame and served as a geo-reference control field. A Canon EOS 1300D camera moved along a 1.5 m long slider rail and took  7 pictures of the control field at 7 different locations of the slider rail over a period of 10 min. The photography was repeated at 3h intervals from 09:00 to 18:00 UTC+7. This apparatus took pictures over a period of 4 months (12 July to 17 October). Each daily photography series produced four sets of pictures (12h and 3h intervals). The best-exposed photo sets were manually selected and merged to produce a 1mm×1mm resolution surface topography.  A Kipp & Zonen CNR4 net radiometer was set up to record incoming and reflected solar radiation of the control field surface. The pyranometers are sensitive to radiation in the wavelength range 0.3 to 2.8μm. Samples were taken every 15s; 10min means were stored on a data logger (CR800, Campbell, USA) located at a height of 1.5m. These measurements were taken over 3-month period from July 12 to October 15 2018. Solar zenith angle has a strong effect on snow surface albedo especially when zenith angles were larger than 60 ° (Gardner and Sharp, 2010; Dickinson et al., 1986). For that reason, we calculated daily albedo from the integrated sum of incoming and outgoing shortwave radiation of only 10min albedo data taken when zenith angles were less than 60°, in an effort to minimize solar zenith angle effects. Note that this weighs the albedo calculation to the middle of the day when insolation is highest and we do not consider zenith angle effects. Bare ice albedo is insensitive to the zenith angle for this impurity-rich glacier, due to LAI isotropic absorption by LAIs and water on the glacier surface (Marshall and Miller, 2020). In this case, the manual and automatic albedo observations are insensitive to the zenith angle in late July and

August as the glacier surface was mainly bare ice.

For manual photogrammetry, we put the aluminum frame horizontally over the ice surface; the plot is detrended by setting four control points and four check points at $z$ axis of the same values. For automatic photogrammetry, the control field of wooden frame was also laid horizontally over the ice surface that lowered as the ice melted and maintained a horizontal position between the control field and ice surface. Four control points and three check points on the wooden frame were applied for georeference and precision check. Both manual and automatic photographs were imported into the Agisoft PhotoScan Professional 1.4.0 application, which produced $1\times1$ m$^2$ plot scale surface point clouds and generated a detrended micro-scale digital elevation model (DEM) of 1 mm resolution at the plot scale. Based on the Agisoft PhotoScan processing report, automatic photogrammetry average geo-referencing errors fluctuated around 1mm. The total root mean square error (RMSE) of the automatic check points was 3.62 $\pm$1.6mm. The standard deviation of control and check point errors were all within 15mm. The manual measurements' average geo-referencing errors fluctuated around 1 mm, the RMSE of the four check points was 0.99$\pm$0.3 mm and the vertical accuracy was 0.66$\pm$0.3mm. The standard deviation for the $x$, $y$ and $z$ axes were all within 5mm. More detailed information about the current research program's use of manual and automatic photogrammetry can be found in Liu et al. (2020).

**3.2 Calculation of surface roughness and effective LAI concentrations**

[Figure]

[Figure]

**Figure 2. Schematic images of cross section of the ice surface LAI distribution over a rough and smooth ice surface, in which warm, cloudy or rainfall days tended to produce a smooth ice surface, and cold and sunny weather tended to produce a rougher ice surface. Panel (a) shows LAIs hidden beneath the weathering crust or cryoconite holes, or covered by snow, and depressions have more LAIs than protrusions; panel (b) shows LAIs distributed over a relatively smooth ice surface without being hidden or protrusion effects.**

There are dozens of surface roughness-describing parameters (Dong et al., 1992; Dong et al., 1994; Smith, 2014). Metrics such as random roughness (RR) or root mean square height deviation ($\sigma$), the sum of the absolute slopes ($\Sigma S$), the microrelief index (MI), rugosity (Brasington et al., 2012) and the peak frequency (the number of elevation peaks per unit transect length) have commonly been used. The surface roughness features impact the distribution of the glacier surface impurities (Fig. 2).

 This means that surface roughness could indirectly affect albedo
by directly affecting LAI concentrations. In this study, a 3-D measure of the topographic roughness is
defined as follows:

$$\xi = h^* \times \frac{A_s}{A_p} \quad (1)$$

where $\xi$ is the surface roughness (cm), $h^*$ represents the average vertical extent of microtopographic
variations (cm), $A_s$ is the detrended 3-D surface area (cm$^2$) and $A_p$ is the planimetric or 2-D area (cm$^2$).
The definition of $\xi$ in this study is adopted from the aerodynamic surface roughness parameter developed
by Lettau (1969),  except that we
changed the silhouette area facing upwind, to the 3-D surface area in equation (2).
The $\xi$ is particularly sensitive to local variability in the surface slope and aspect and the presence of
topographic singularities. We adapted equation (1) to calculate surface roughness ($\xi$) by using the
automatic and manual photogrammetry-acquired detrended $1\times 1$m plot DEM data at 1mm resolution.
Based on the plot-scale $\xi$ and sampled LAIs, the LAI concentration ($C_{LAIs}$) over smooth  snow ($\xi$=0 cm)
and the ice surface was estimated based on the following equation:

$$C_{LAIs} = \frac{M_s}{V} \quad (2)$$

where $C_{LAIs}$ is the LAI concentration over the smooth surface (g cm$^{-3}$), $M_s$ is the weight of dried LAIs (g)
and $V$ is volume of collected snow or ice samples at the August-one ice cap (cm$^3$).

A rough ice surface means a greater surface area and lower LAI concentrations over the smooth ice surface;
also rough surface structures such as depressions and cryoconite holes hide LAIs from direct sunlight and
reduce the LAI concentrations. In this case, an effective LAI concentration ($C_\xi$) by considering surface
roughness is introduced as follows:

$$C_\xi = \frac{M_s}{V(1+\xi)} \quad (3)$$

In equation (3), we assumed that the effective LAI concentration ($C_\xi$) can be adjusted by considering the
surface roughness effects. Here, $C_\xi$ is the effective LAI concentration over the rough surface (g cm$^{-3}$). As

for the very smooth ice surface or snow surface, $\xi$ is close to 0, and the limit of $C_\xi$ is $C_{LAIs}$. For the snow surface, insoluble LAIs are retained at the snow surface, so concentrations of LAIs in the surface snow increase with snowmelt and increase with a low snow surface $\xi$, and no rough structures form in the melt season. In this case we expect equation (2) to be more appropriate for calculating snow surface LAI concentrations in the melt season. Equation (3) is appropriate for calculating ice surface LAI concentrations especially for rough ice surface.

**4. Field observation of spatial and temporal variability of albedo, surface roughness and LAI concentrations**

**4.1 Manual observation of spatial and temporal variability of surface roughness, LAIs and albedo**

We found a patchy snow cover with many cryoconite holes on the glacier terminus (~4600 m) at the date of the first field trip (July 12, 2018).  As altitude increased, there was less bare ice, more snow and fewer cryoconite holes. From 4700m to the top of glacier, there was near-complete snow coverage. Manual measurements indicated that surface albedo increased from 0.31±0.06 at the glacier terminals to 0.61±0.04 at the top of the ice cap (Fig. 3a). The associated surface roughness measurements decreased from 5.5±1.5cm to 0.5±0.6cm as altitude increased (Fig. 3b). LAIs decreased from 0.02±0.013mg g$^{-1}$ for patchy snow cover to around 0.0018±0.001g cm$^{-3}$ for snow cover as altitude increased (Fig. 3c). On August 3, the August-one ice cap surface was basically all ice, except at the top of the ice cap, where there was still some patchy snow cover. Measurements indicated that albedo increased from 0.14±0.03 at the terminals to 0.21±0.05 at the top of the ice cap (Fig. 3d). Glacier surface albedo had decreased greatly when compared with the July 12 observations. An increasing trend of surface roughness could be detected as altitude increased and fluctuated between 1.4±0.4cm and 3.3±1.1cm (Fig. 3e). LAIs decreased from 0.05±0.01g cm$^{-3}$ to around 0.01±0.005g cm$^{-3}$ as altitude increased (Fig. 3f). On August 28, the ice cap surface was entirely bare ice. Albedo showed an increasing trend with altitude and fluctuated between

0.14±0.04 at the middle part to 0.19±0.05 at the top of the ice cap (Fig. 3g). Ice surface roughness fluctuated between 1.50±0.5cm at the middle of the ice cap to 1.45±0.95cm at the top of the ice cap, and it did not significantly increase as altitude increased (Fig. 3h). LAIs decreased from 0.04±0.03g cm$^{-3}$ at the middle part to 0.003±0.002g cm$^{-3}$ at higher elevations (Fig. 3i).

395  On July 12, surface snow cover gave way to ice, and the transient snowline retreated up-glacier . On the snow-covered glacier surface, surface roughness and LAIs decreased as altitude increased (Fig. 3b, 3c). Albedo, conversely, increased as altitude increased (Fig. 3a). As the melt season progressed, the transient snowline retreated to the top of the ice cap on August 3. The partially snow-covered ice surface formed a

400 rougher surface at the top than at lower altitude. There was a much higher concentration of LAIs on the uncovered ice surface than on the snow surface. Albedo tended to be lower on bare ice surface than on snow.  After the ice cap was fully bare ice, the glacier showed season minimal albedo over the entire surface. The surface roughness of the ice was quite variable and it did not show a clear trend with altitude.

405 LAI concentrations were highly variable and showed an increasing trend as altitude increased.

[Figure]

**Figure 3. Albedo, surface roughness and LAI concentration versus altitude. (a-c) As observed on 12 July, (d-f)**
**as observed on 3 August, (g-i) as observed on 28 August.**

410

**4.2 Automatic observation of temporal variability of albedo and surface roughness**

The automatic measurement setup was in the middle of the ice cap, and it recorded daily surface roughness and albedo, operating successfully for the melt season from July 12 to September 15, and for 1 month during the accumulation period from September 16 to October 17 (Fig. 4a). Photographs captured by the automatic Canon 1300D showed that from July 12 to July 20 the glacier was snow-covered (Fig. 4c). Because fresh snow has a large specific surface area (SSA) and small surface roughness, snowfall events on July 14 and 20 inducing a high albedo (Figs. 4a, 4b). After the snow had stopped falling, surface albedo tended to decrease and surface roughness to increase (Fig. 4a). When the air temperature increased to 0 ᵒC and the snow began to melt (Fig. 4b), the glacier surface roughness kept increasing and albedo kept decreasing. From July 21 to July 24, snow cover became patchy and surface roughness increased from 1.1cm to 2.1cm (Fig. 4a and 4d). During this period, surface albedo decreased sharply from 0.72 to 0.30. The micro-scale structures of cryoconite holes formed during this period (Fig. 4d). From July 25 to September 3, the glacier surface was mostly bare ice with occasional snow fall events (Fig. 4e and 4f). Surface albedo fluctuated between 0.11±0.01 for the smooth ice surface and 0.68±0.02 for the intermittent snowfall period, and surface roughness fluctuated between 0.8 cm and 1.6 cm. There were LAIs concentrated on the ice surface (Fig. 4e and 4f). Intermittent snowfalls covered the LAIs and reduced the surface roughness and increased SSA during this period. At such times,  albedo increased sharply above 0.6. When the snow melted, leaving patchy snow cover. The decreased SSA and increased surface roughness led to a fast decrease of albedo to around 0.3±0.05 within 2 days. As soon as bare ice appeared, the ice surface albedo decreased to around 0.14±0.03. At the end of melt season, from September 4 to 13, ice surface roughness fluctuated and increased from 0.15 cm to 2.63 cm. The ice surface micro-scale structures of cryoconite holes hid the LAIs from direct sunlight. Refreezing processes also formed weathering crust and covered LAIs (Fig. 4g). The ice surface albedo followed an increasing trend during this period and increased to 0.55 on September 13. During the following 2 days, snowfall events increased albedo sharply from 0.55±0.02 to 0.83±0.06 (Fig. 4h) due to the high SSA and decreased snow surface roughness.

[Figure]

**Figure 4. (a) Variations of glacier albedo and surface roughness over time at the automatic photogrammetric surface roughness observation site. (b) Half-hour air temperature and daily precipitation at top of the ice cap. Photograph (c) shows the snow-covered surface ; photograph (d) shows the partially snow-covered surface  with cryoconite holes; (e) and (f) show the smooth ice surface ; (g) shows the rough ice with ice curst ; (h) shows the smooth snow surface ; (i) and (j) show blowing snow events inducing sastrugi .**

440

After September 15, the air temperature dropped below 0 °C and the snow-covered surface basically entered the accumulation period at the automatic photogrammetry site. Snow surface albedo fluctuated from 0.61 to 0.90. Snow surface roughness fluctuated between 0.30 cm and 1.67 cm. Based on the snow surface patterns captured by the Canon 1300D camera and meteorological data, we could tell that the surface roughness variation was mainly induced by constant blowing snow events and intermittent snowfall effects during this period (Figs. 4i and 4j). Blowing snow events not only increased snow density and decreased its SSA, but also formed sastrugi and increased snow surface roughness. These factors all decreased the dry snow albedo in the accumulation period. Basically, seasonal variations were large on the August-one ice cap, from above 0.8 for fresh, dry snow at start of the melt season, to 0.5 for aged, wet snow in mid-summer, to as low as 0.1 for impurity-rich bare ice that was exposed after snow had melted, back to 0.7 for wind-packed, dry snow. We find that Sseasonal albedo fluctuation at the August-one ice cap is due to snow metamorphism, SSA variation, water content in the snow and concentrations of impurities (Warren and Wiscombe, 1980; Conway et al., 1996; Gallet et al., 2009; Gardner and Sharp, 2010; Domine et al., 2006).

**5 Statistical relationship between Albedo albedovariability, surface roughness and LAIs**

**45.1 Snow surface in the melt season**

The 1 m² plot sSnow surface albedo and corresponding surface roughness calculated at 1mm resolution are analyzed using based on the manual and automatic observations at the August-one ice cap. A significant negative power functions is are established between the snow surface roughness and albedo for manual and automatic observations, respectively (Fig. S2 in the Supplement). The correlation coefficient reached 0.77 (Fig. S2a) for manual and 0.88 (Fig. S2b) for automatic one. The combined manual and automatic scatter diagrams of Fig. 5a display a significant negative power function, and the correlation coefficient reached 0.82 0.83 between the snow surface roughness and snow albedo.

Snow surface albedo is greatly affected by LAIs. The scatter plots of Fig. 6a show the relationship between $C_{LAIs}$ and albedo without considering the surface roughness effect on LAI concentrations. A significant negative power function relationship was established between snow LAIs and albedo (Fig. 6a,

r=0.91). The scatter diagram of Fig. 6b shows the effective LAI concentration $C_\xi$ and albedo relationships by considering the roughness effect on the snow surface. Figure 6b display a power function between the manually sampled snow $C_\xi$ and observed surface albedo (r=0.60). The correlation coefficient decreased significantly compared with Fig. 6a (r=0.91). This means that considering the surface roughness effect on

475 LAI concentration by equation (3) does not improve the relationship between LAIs and albedo during the snow period. It indicates that the snow surface LAI concentration is not affected by micro-scale snow surface features since melt-out LAIs are concentrated over the snow surface.

We calculate a coefficient of determination value of 0.88 by using a multiple nonlinear regression that includes the surface roughness and $C_{LAIs}$. This indicates that over 85% of the variability in snow albedo

480 can be accounted for using surface roughness and $C_{LAIs}$, in which LAIs explained 54% and surface roughness explain 35 % of snow albedo variance. The resulting equation is as follows:

$$\alpha = 0.234 C_{LAIs}^{-0.1415} - 0.02098 \xi^{1.226}. \quad (4)$$

where $C_{LAIs}$ is is the LAI concentration, $\xi$ is the snow surface roughness (cm).

485

[Figure]

Figure 5. (a) Combined manual and automatic measurements of snow and ice surface roughness versus albedo in the 2018 melt season; (b) combined manual and automatic measurements of bare ice surface roughness versus albedo in the 2018 melt season without inclusion of partial snow cover.

[Figure]

Figure 6. (a) Snow surface $C_{LAIs}$ versus albedo in the 2018 melt season; (b) snow surface $C_\xi$ versus albedo in the 2018 melt season based on manual observations.

495

**5.2 Ice surface in the melt season**

With inclusion of snow patches and bare ice, combined manual and automatic measurements of ice surface albedo and surface roughness displayed a significant positive linear relationship (Fig. 5a, r=0.62). Since snow patches coverage have significant effect over ice surface albedo. Without inclusion of snow

500    patches, both manual and automatic measurements of bare ice surface albedo and surface roughness displays significant positive linear relationships. The correlation coefficients reached 0.64 and 0.47 for manual and automatic estimated surface roughness and albedo, respectively (Fig. 5b). It means the positive relationship between bare ice albedo and surface roughness are due to changes in surface roughness.

505

510 Based on Figs. 5a  we were able to find that the negative power function between snow and patchy snow surface roughness and albedo changed to a positive linear relationship between  bare ice surface roughness and albedo. A tipping point of the negative relationship to a positive relationship between surface roughness and albedo appeared when a bare ice surface covered by snow patches. We identified that the patchy

515 snow cover is the limit between the two periods. During this period, the large quantity of LAIs hidden beneath the snow surface reappeared at depressions or cryoconite holes (Fig. 4d).

Fluctuations in the ice surface albedo are mainly affected by the concentrations of the uppermost surface layers' LAIs. The scatter plot of Fig. 7a shows the relationship between $C_{LAIs}$ and albedo

520 . Figure 7a displays a significant power function between the manually sampled ice $C_{LAIs}$ and observed ice surface albedo (Fig. 7a, r=0.59). The scatter diagrams of Fig. 7b display a significant power function between the effective LAI concentration $C_{\xi}$ and the observed ice surface albedo (Fig. 7b, r=0.79). The correlation coefficient in Fig. 7b between effective LAI concentration and surface albedo has greatly improved by considering the

525 surface roughness effect on LAI concentration than not consider surface roughness in Fig. 7a. The coefficient of determination has increased from 0.35 to 0.63 after considering the surface roughness effect. This means that equation (3), estimating the effective LAI concentration $C_{\xi}$, could explain more ice surface albedo variation than $C_{LAIs}$ does.

[Figure]

Figure 7. (a) Ice surface $C_{LAIs}$ versus albedo in 2018 melt season; (b) ice surface $C_{\xi}$ versus albedo in 2018 melt season based on manual observations.

**5.3 Snow surface in accumulation period**

In late September and October, the glacier surface is covered with dry snowpack and the LAI concentration on the fresh snow-covered ice cap is very low. In this case, the effect of LAIs is not presented in the accumulation period. The scatter diagram of snow surface roughness and albedo shows negative power function (Fig. 8, r=0.49). Theoretical studies have indicate that the initial size distribution, vertical temperature gradient and snow density all affects albedo evolution in this period. Vapor diffusion caused by curvature differences causes rapid albedo decay in the first day following snowfall (Flanner and Zender, 2006). Studies have indicated that under either isothermal conditions or under temperature gradient conditions, the dry snow specific surface area (SSA) decreases during metamorphism (Taillandier, et al., 2007). The decay of snow SSA produces a decreasing trend in snow albedo (Domine et al., 2006; Flanner and Zender, 2006). This is in addition to the snow surface being exposed to strong

545 winds (average 4.8m s$^{-1}$ at 4 m above the surface) over the August-one ice in the accumulation period, which could also increase the rate of the SSA decrease and accelerate albedo decrease (Cabanes et al., 2003).

In the accumulation season, except for the dry snowpack metamorphism process, constant blowing snow formed a high-density sastrugi windpack, which dominated surface roughness fluctuation (Figs. 4a, 4i

550 and 4j). Over a sastrugi field, Warren et al. (1998) mentions two causes for albedo reduction. Firstly, sastrugi lower the average incidence angle, which reduces the albedo due to the strong dependence of albedo on the incidence angle of incoming radiation (Warren, 1982). Perpendicular insolation could result in an albedo decrease between 2% and 4% relative to parallel insolation (Carroll and Fitch, 1981; Kuhn, 1974). Secondly, multiple reflections between the sastrugi causes light trapping in the trough.

555 Radiative transfer model studies have indicated that differently shaped channels and crevasses decrease in effective albedo over time due to the changing morphologies of the roughness features (Cathles et al., 2011). The field experiment in the accumulation season showed that the negative correlation between surface roughness and albedo is affected not only by the dry snow metamorphism of decreased SSA, but also by the strong wind-formed sastrugi.

[Figure]

560

Figure 8. Automatic measurements of snow surface roughness versus albedo in the accumulation period from September 16 to October 17 2018

**6. Discussion**

**6.1 Snow surface albedo variability explanation**

 At the August-one ice cap, temperatures are above freezing during the day and snow is subject to rapid metamorphism and production of larger, typically rounded grains and clusters. Meltwater can refreeze at the surface forming crusts when air temperatures decrease at night. The melt and refreeze patterns lead to hardened crusts made up of aggregated rounded grains and clusters of grains that have frozen together. The snow surface roughness follows an increasing trend due to snow grain particle metamorphism and differential melting of the surface. apid grain growth corresponding with decreasing SSA and increasing of surface roughness . There is often a concurrent rise in albedo due to the blanketing of old snow by fresh snowfall. New fresh snow surface has reduced surface roughness by preferential deposition, smaller grains and larger SSA, which increase the albedo. Fig. 9 shows the snow grain size, fresh and old snow surface roughness measured on 14 July 2020 at the top (4820 m) and lower part (4623 m) of the ice cap. We could find old snow surface roughness is larger than fresh snow surface due to its larger snow grain size and greater DEM variation. Due to preferential deposition, more fresh snow deposited on the pits than peaks (Fig. 9d). The snow grain size at the pits are smaller than the peaks (Fig. 9d). The DEM variation are also larger at the peaks than pits due to its larger snow grain size (Fig. 9f). The surface roughness fluctuation synchronously happened with SSA and grain size variation. Although we do not have tandem surface roughness, grain size and SSA observations, the evolution of surface roughness calculated at 1mm resolution in the snow-covered period should be  similar with the fluctuation trend of grain size evolution. The grain size is one of the most critical factors affecting snow albedo. As grain size increases, scattering within the snowpack decreases and the

absorbing path length within grains increase, thus reducing the broadband albedo (Wiscombe and Warren, 1980; Dozier et al., 1981; Warren, 1982; Flanner and Zender, 2006). Linear or exponential relationships have been established between wet snow grain size and snow albedo (Brock et al., 2000; Adolph et al., 2016; Skiles and Painter, 2017) in the melt season, which is very similar with the scatter plot of surface roughness and snow albedo provided in this study in Fig. 5a. As discussed in the preceding sections, both increasing surface roughness and LAI concentration tended to decrease albedo in snow melting period. The hidden effect or dilute effect of surface roughness on LAI concentrations is not discerned by application of equation (3) to the snow surface, as other studies have already proven that LAIs are retained at the snow surface (Doherty et al., 2013), and scavenging by melting process is limited (Skiles and Painter, 2017).

[Figure]

Figure 9 Manual photogrammetry photographed fresh and old snow surface, corresponding plot DEM, DEM profile variation and measured snow grain size on 14 July 2020. Figure 9a is smooth snow surface photographed 12 h after snowfall at top of the ice cap with snow grain size of 0.2 - 1.0 mm; 9b is corresponding DEM of Figure 9a; Figure 9c is 4 cm wide profile DEM variation at middle of the plot

marked out by dashed line in Figure 9a and 9b. Figure 9d was rough snow surface photographed 11 h after snowfall close to the terminal of the ice cap at 4623m. The gray area in Figure 9d is old snow peaks with coarser snow grain size of 1.0 - 3.5mm. The white part are fresh snow covered pits with smaller snow grain size of 0.3mm – 1.0mm than gray area. Figure 9e is corresponding DEM of Figure 9d, Figure 9f is 4 cm wide profile DEM variation at middle of the plot marked out by dashed line in Figure 9d and 9e.

LAI is another highly critical factor affecting snow albedo (e.g., Adolph et al., 2016; Skiles and Painter, 2017). Albedo is most strongly reduced by the presence of LAIs in the visible part (Warren and Wiscombe, 1980), and this effect is enhanced as snow grain size increases (Flanner et al., 2007; Hadley and Kirchstetter, 2012). Higher LAI content can also impact near-infrared range albedo (Hadley and Kirchstetter, 2012). As discussed in the preceding sections, both increasing surface roughness and LAI concentration tended to decrease albedo. The hidden effect or dilute effect of surface roughness on LAI concentrations is not discerned by application of equation (3) to the snow surface, as other studies have already proven that LAIs are retained at the snow surface (Doherty et al., 2013), and scavenging by melting process is limited (Skiles and Painter, 2017). We calculate a coefficient of determination value of 0.88 by using a multiple nonlinear regression that includes the surface roughness and $C_{LAIs}$. This indicates that over 85% of the variability in snow albedo can be accounted for using surface roughness and $C_{LAIs}$. The resulting equation is as follows:

$$\alpha = 0.234 C_{LAIs}{}^{-0.1415} - 0.02098\xi^{1.226}. \quad (4)$$

The statistical relationship and significance between snow surface roughness and albedo in the melt season are also different with the accumulation period. During the accumulation season, fine-scale surface roughness develops due to ice grain metamorphism and differential melting. In the accumulation period, large-scale features develop due to constant blowing snow events, and snow surface albedo and roughness are controlled by snow metamorphism and blowing snow sastrugi. There are snow surface features with microstructures in the melt season. Snow surface roughness increases and albedo decreases due to the rapid decrease of SSA, grain metamorphism, water content and LAIs. Several contributions affect the

albedo, and it is very difficult to separate each contribution and quantify their impacts on the albedo
evolution. There is a general understanding of the physics that control the albedo of snow in the melt
season and accumulation period, but different factors dominate the changes in albedo at various locations
(Doherty et al., 2013; Skiles and Painter, 2017). Albedo is governed primarily by the SSA and LAIs.
However, the variables which physically control spatial and temporal albedo are difficult to incorporate
directly into energy balance models at the glacier-wide scale (Brock et al., 2000). Therefore, surrogate
variables such as snowpack age, snow depth, snow density and air temperature are widely used in albedo
parameterizations (Amaral et al., 2017). The field experiment in this study indicated albedo also correlates
highly with surface roughness, which is related to surface layer characteristics of the SSA and the snow
grain size surface layer. We expect that surface roughness might be a proxy variable of SSA and needs to
be confirmed by comparative observations of SSA and roughness over dry and clean snow in later studies.
Considering its advantage of being easier for measurement on a glacier-wide scale, it could be used in
snow albedo parameterization in other places.

**56.2 Ice surface albedo variability explanation**

It has been shown that small or macro-scale surface roughness reduces surface albedo and enhances solar
absorption due to decreasing incidence angle and light trapping (Warren et al., 1998; König et al., 2001;
Arnold and Rees, 2003; Rees and Arnold, 2006; Cathles et al., 2011; König et al., 2001; Zhuravleva and
Kokhanovsky, 2011; Lhermitte et al., 2014; Larue et al., 2019). Conversely, in this study a significant
positive relationship rather than a negative relationship was established between ice surface albedo and
surface roughness over the ice surface based on manual and automatic measurements. We suspect that it
is related with abundant LAIs over the ice surface of the August-one ice cap. Firstly, a rough ice surface
means more protection of LAIs from sunlight and a smooth ice surface means more exposure of LAIs to
direct sunlight. These hidden effects such as cryoconite holes, which could increase surface albedo, have
been widely reported (e.g., Bøggild et al., 2010). Manual observations on August 3 and 28 indicate that
micro-scale structures of ice surface cryoconite holes protect LAIs from direct radiation (Figs. 4d and 4g,
Fig. 8e). Secondly, a rough ice surface means greater surface area than a smooth ice surface, which might

have decreased the concentration of LAIs over the rough ice surface compared to the smooth ice surface (Fig. 10a). Additionally, we find LAIs were concentrated in depressions rather than protrusions. This heterogeneous distribution characteristics of LAIs across rough ice surface led to a reduction of shortwave radiation absorption (Fig. 10f). . Figure  10 displays photographs captured during field LAI measurements that display LAI distribution characteristics over a partially snow- or ice-covered surface (Fig. 10c), smooth dark bare ice (Fig. 10d), cryoconite holes (Fig. 10e), protrusions (Fig. 10f) and combined hidden effect of LAIs covered by granular ice and protrusions affecting the ice surface (Fig. 10g). Obviously, cryoconite holes or protrusions all have larger ice surface area and surface roughness over smooth ice, and decrease the LAI effect on the absorption of shortwave radiation. A smooth ice surface roughness features a maximized LAI effect on the absorption shortwave radiation and induces the lowest albedo of 0.1 (Fig. 10d). The positive linear relationship between the ice surface roughness and albedo in Fig. 5b reflect the surface roughness effect on LAI concentration. For dirty ice with heavy loading of LAIs, micro-scale rough ice means an ice surface featured with cryoconite holes, protrusions and depressions or ice crust, and induces higher albedo than with a smooth ice surface, which corresponds with a high concentration of LIAs and lower albedo. We believe the ice surface roughness and albedo might have a negative power function similar with snow surfaces provided ice surface LAI concentrations are very low.

As Brock et al. (2000) suggested, to calculate albedo variation accurately in numerical models, parameterizations must be as physically based as possible. The performance of the established albedo methods either are based on surface roughness in Fig. 5b or effective LAI concentration in Fig. 7b shows great improvement over the assumption of a constant mean ice albedo or surrogate variables, such as air temperature, accumulated melt and elevation (Brock et al., 2000). The physical properties of uppermost surface ice layer, including its density, crystal structures, surface roughness, impurity content and

stratification and SSA, are subject to continuous changes as the melt season proceeds (Jonsell et al., 2003). Changes in characteristics of this layer are mainly responsible for the observed variability in ice albedo at any given site (Dadic et al., 2013; Jonsell et al., 2003). In this study, surrogate variables of effective LAI concentrations consider both surface roughness and LAIs, explaining a significant amount of ice surface albedo variation, and $C_\xi$ has a higher coefficient of determination than either surface roughness or LAIs (Table 1).

[Figure]

Fig. 910. (a) Sensitivity of surface roughness under different resolutions. (b) Smooth snow surface with

an albedo of 0.52±0.02; (c) rough partially snow-covered surface with an albedo of 0.38±0.04; (d) bare ice that is smooth, dark and rich in LAIs, with an albedo of 0.1±0.01; (e) cryoconite hole hidden effect with an albedo of 0.36±0.02; (f) protrusion effect with an albedo of 0.18±0.02; (g) protrusion and hidden effects with an albedo of 0.23±0.01. (b), (c) and (e) were photographed on July 12 2018, (d) and (f) on August 3 2018 and (g) on August 28 2018.

**6.3 Glacier surface albedo parameterization based on micro-scale surface roughness data**

Snow and ice surface albedo parameterization methods are established based on only surface roughness, LAIs or both LAI concentration and surface roughness in the melt season in Table 1. In this study, we have applied 1×1mm resolution DEM data to calculate surface roughness. Since the surface roughness is dependent on and sensitive to topography data resolution (Fig. 10a), it is important to understand which resolution is appropriate for snow and ice surface albedo estimation. Based on the manual photogrammetric topographic data, Fig. 11 shows correlation coefficients between surface roughness and albedo under different resolutions of the snow and ice surface. For snow-covered surfaces, the coefficient increases quickly from -0.67 to -0.74 when 1m plot resolution improves from 333.3 mm to 200.0 mm, after which the coefficient stabilized at -0.72±0.04 with the increase of resolution. In this study, nine manual photogrammetric snow surfaces are flat and smooth and three plots are partially snow-covered rough surfaces. The partially snow-covered surface roughness is more sensitive to the changes of resolution than the flat snow surface (Fig. 10a), because most of manual photogrammetric snow-covered surfaces are flat. Consequently, the correlation coefficient between albedo and snow surface roughness is not sensitive either. Figure 11 shows that the correlation coefficient increases much slower under the finer resolution of 100mm. A resolution of finer than 100 mm is recommend for surface roughness calculations for snow surfaces in the melt season.

Table 1 Accuracy of snow and ice surface albedo parameterization

| Surface types | Equations | $R^2$ | RMSE | $P<$ | Albedo range |
|---|---|---|---|---|---|
| Snow surface for | $\alpha = 0.234 C_{LAIs}^{-0.1415} - 0.02098\xi^{1.226}$ | 0.89 | 0.0536 | 99.99 | 0.85-0.22 |

| | | | | | |
|---|---|---|---|---|---|
| melting season | $\alpha = 0.1012 \times \xi^{-0.338}$ | 0.68 | 0.0852 | 99.99 | 0.92-0.22 |
| | $\alpha = 0.634 \times C_{LAIs}^{-0.342}$ | 0.84 | 0.0603 | 99.99 | 0.70-0.20 |
| Snow surface for accumulation season | $\alpha = 0.4613 \times \xi^{-0.093}$ | 0.25 | 0.0608 | 99.9 | 0.92-0.65 |
| | $\alpha = 0.0585 \times C_{LAIs}^{-0.225}$ | 0.35 | 0.0659 | 99.9 | 0.10-0.38 |
| Ice surface | $\alpha = 0.0374 \times C_{\xi}^{-0.26}$ | 0.63 | 0.0501 | 99.99 | 0.10-0.38 |
| | $\alpha = 3.3 \times \xi + 0.1212$ | 0.38 | 0.0503 | 99.9 | 0.12-0.38 |

For ice surfaces, the correlation coefficient between surface roughness and albedo increases from 0.17 to 0.84 with the increase of the plot resolution from 333.3 mm to 0.4 mm. The calculated surface roughness is very sensitive to resolution, especially for those rough ice surfaces such as cryoconite holes and protrusions (Fig. 8a10a). A coarse resolution reduces surface roughness differences of the different ice surface features. Consequently, the correlation coefficient reduces as resolution worsens. Figure 10 11 shows that the correlation coefficient increases much slower under finer resolution (<50mm). For albedo studies, finer resolution than 50 mm resolution is recommended for topography and surface roughness data during the melt season.

For working at whole-glacier scale, recent developments in terrestrial laser scanning, structure-from-motion, and multi-view stereo may also be able to provide catchment-scale high-resolution topography and surface roughness data (Xu et al., 2019; Rossini et al., 2018; Passalacqua et al., 2015; Rippin et al., 2015; Smith and Vericat, 2015; Rippin et al., 2015 Rossini et al., 2018; Xu et al., 2019; Rossini et al., 2018;). Studies such as unmanned aerial vehicle photogrammetry create centimeter-resolution DEM and orthophotos for glacier scale. These data have already been applied successfully for glacier surface roughness and brightness (albedo) analysis (Rossini et al., 2018). Smith et al. (2016) have suggested a statistical method to extrapolate glacier-scale roughness from finer-resolution plot-scale surface roughness measurements and glacier-scale DEM data. This could be a practicable way to parameterize

surface roughness at the glacier scale. This statistically derived glacier-scale surface roughness could also be applied for estimation of albedo based on the surface type and roughness–albedo statistical relationships. Additionally, LAI coverage can be captured by high-resolution images (e.g., Takeuchi et al., 2018). LAI coverage might be a good substitute for LAI concentration in albedo parameterization, especially for dark and LAI-rich bare ice. Caution must be taken when surface roughness is applied for albedo estimation, since surface roughness is affected not only by melt-induced micro-scale surface features, but also by movement-induced macro-scale fractures, crevasses and water channels (Rossini et al., 2018). Rough surfaces are expected to produce great brightness reductions due to an increase in light trapping from multiple reflections within the rough features. The albedo of glacial surfaces is affected by LAIs, and macro- and micro-scale features, especially for bare ice surface. Bare ice surface albedo parameterization must consider these different scale roughness features and LAIs.

[Figure]

Figure 1011. Correlation coefficients between albedo and surface roughness under different resolutions. Snow and ice surface roughness is estimated based on manual photogrammetric data under different

resolutions at the 1 m plot scale.

**67. Conclusion**

By using manual and automatic photogrammetric measurements of surface roughness, manual LAI samples and measurement of broadband albedo at the August-one ice cap, we have gained a basic understanding of the surface roughness and LAI effects over snow and ice surface. The negative power function statistical relationships between snow surface roughness and albedo changed to positive linear relationship for the ice surface. The LAIs are retained at the snow surface and their concentrations increased with snowmelt. LAIs over the ice surface can increase due to melt concentration effects, be removed by rainfall and meltwater runoff or be hidden in melt-formed features such as cryoconite holes or weathering crust. A positive linear relationship between ice surface roughness and albedo indicates that micro-scale smooth ice surface increased LAI concentrations over rough surfaces and decreased ice surface albedo. Detailed analysis of albedo, LAI concentrations and surface roughness leads us to conclude that surface roughness is a good surrogate variable, allowing the calculation of snow and ice albedo variation and estimation of the net shortwave radiation energy budget, especially in the melt season. Snow and ice surface albedo parameterization methods are developed based on high-resolution surface roughness or both LAI and surface roughness.

Surface roughness seems to play a different role in snow surfaces albedo than it does for ice surfaces. For snow-covered surfaces, ice snow particle metamorphism and surface melting and refreezing induced an increase in grain size and decrease in SSA,. Differential melting and snow metamorphism induced surface roughness increase. This process occurred synchronously with snow metamorphism.and occurred synchronously with surface roughness increase, . which Increase in grain size and decrease in SSA induced decreasing snow surface albedo in the melt season. For dirty ice surfaces, the increasing surface roughness increased its surface albedo through the combination of the hidden LAI effect, a greater surface area diluting the LAI effect and protrusion-induced uneven distribution of the LAI effect. Surface roughness over the LAI effect can be simplified by considering surface roughness during calculation of LAI concentrations in equation (3). This will give respectable results even if only surface roughness data

are used. LAI concentration data are rarely available over the whole glacier except when sampled manually. The ice surface albedo parameterization method based on $C_\xi$ is not practicable. However, considering the proportion of the LAI-covered area estimated at the plot scale based on high-resolution images might be a good substitute for LAI concentrations in the future. Correlation coefficient analysis between the surface roughness and albedo at different resolutions indicate that resolutions finer than 50 mm and 100 mm resolution are recommend for ice and snow surface roughness calculations, respectively. Surface roughness develops due to local melt inhomogeneities in the melt season. The study by Liu et al. (2020) indicates that a relatively high and positive turbulent heat proportion induces a smooth ice surface and a lower or negative turbulent heat proportion induces a rough ice surface. In this study, a positive linear relationship was established between ice surface roughness and albedo. Based on these two studies, we could conclude that ice surface roughness played a delicate role in turbulent heat flux and net shortwave radiation over the ice surface. Smooth ice usually developed during warm and cloudy days, and a smooth ice surface means a greater concentration of LAIs over the ice surface and more efficient absorption of inward shortwave radiation, but less efficiency in turbulent heat exchange. Under sunny and cloudy days, the smooth ice surface developed into a rough ice surface because of the high proportion of shortwave radiation. Consistent shifting of the weather pattern from cloudy to sunny or vise versa induced a relatively small fluctuation of ice surface roughness during the melt season. Ice surface albedo also remained around $0.15 \pm 0.05$ except during intermittent snowfall, which increased ice surface albedo. At the end of melt season, the roughest ice surface usually developed under consistent cold and sunny days. The rough ice surface not only induced loss of turbulent heat loss from ice surface, but also hid LAIs and increased the ice surface albedo to around 0.4. Consequently, the efficiency of net shortwave absorption was also reduced to its minimum and ice surface melt shut down. In this study, we did not intend to present a more quantified and detailed analysis about surface roughness's role in adjusting the shortwave ratio and turbulent heat flux. More complete data on LAIs, surface roughness, snow surface area, SSA, snow grain size and albedo need to be collected to help us present a more detailed analysis and theoretical-based research on surface roughness and LAIs at the micro scale over ice and the snow

surface energy and mass balance processes. In particular, the inclusion of SSA, snow grain size surface roughness, and albedo measurements on snow and ice albedo will allow for better assessment of the

800    potential response of the glacier to a changing climate in future as well as improving the modeling of the energy and mass balance, not only in the glaciers of the Qilian Mountains, but also in other glacial regions that experience severe LAI effect, such as in Central Asia and the Himalaya.

*Data availability*. All of the observational data presented in this study are available upon request to the

805    author (Junfeng Liu, liujfzyou@lzb.ac.cn).

[revised manuscript text omitted]

Doherty, B. J., Grenfell, T. C., Forsstrom, S., Hegg, D. L., Brandt, R. E., and Warren, S. G.: Observed vertical redistribution

of black carbon and other insoluble light-absorbing particles in melting snow, Journal of Geophysical Research: Atmospheres,

118, 5553–5569, doi:10.1002/jgrd.50235, 2013.

865   Domine, F., Salvatori, R., Legagneus, L., and Salzano, R.: Correlation between the specific surface area and the short wave infrared (SWIR) reflectance of snow: Preliminary investigation, Cold Regions Science and Technology, 46, 60– 68, doi:10.1016/j.coldregions.2006.06.002, 2006.

Dong, W. P., Sullivan, P. J., and Stout, K. J.: Comprehensive study of parameters for characterising three-dimensional surface topography IV: parameters for characterising spatial and hybrid properties, Wear, 178, 45–60, doi: 10.1016/0043-
870   1648(94)90128–7, 1994.

Dong, W. P., Sullivan, P. J., and Stout, K. J.: Comprehensive study of parameters for characterizing three-dimensional surface topography I: Some inherent properties of parameter variation, Wear, 159, 161–171, 1992.

Dong, Z., Qin, D., Chen, J, Qin, X., Ren, J., Cui, X., Du, Z., and Kang, S.: Physicochemical impacts of dust particles on alpine glacier meltwater at the Laohugou Glacier basin in western Qilian Mountains, China, Science of the Total Environment, 493, 930–942, 2014a.

Dong, Z., Qin, D., Kang, S., Ren, J., Chen, J., Cui, X., Du, Z., and Qin, X.: Physicochemical characteristics and sources of atmospheric dust deposition in snow packs on the glaciers of western Qilian Mountains, China, Tellus B: Chemical and Physical Meteorology, 66, 20956, doi:10.3402/tellusb.v66.20956, 2014b.

Dozier, J., Schneider, S. R., and McGinnis, D. F.: Effect of grain size and snowpack water equivalence on visible and near-infrared satellite-observations of snow, Water Resources Research, 17(4), 1213–1221, doi:10.1029/WR017i004p01213, 1981.

Fassnacht, S. R., Stednick, J. D., Deems, J. S., and Corrao, M. V.: Metrics for assessing snow surface roughness from digital imagery, Water Resources Research, 45, W00D31, 10.1029/2008wr006986, 2009a.

Fassnacht, S. R., Williams, M., and Corrao, M.: Changes in the surface roughness of snow from millimetre to metre scales, Ecological Complexity, 6, 221–229, 10.1016/j.ecocom.2009.05.003, 2009b.

Fitzpatrick, N., Radić, V., and Menounous, B.: A multi-season investigation of glacier surface roughness lengths through in situ and remote observation, The Cryosphere, 13, 1051–1071, doi:10.5194/te-13-1051-2019, 2019.

Flanner, M. G., and Zender, C. S.: Linking snowpack microphysics and albedo evolution, Journal Geophysical Research, 111, D12208, doi:10.1029/2005jd006834, 2006.

Flanner, M. G., Zender, C.S., Randerson, J.T., and Rasch, P. J.: Present-day climate forcing and response from black carbon in snow. Journal of Geophysical Research: Atmosphere. 112. D11202, doi:10.1029/2006JD008003, 2007.

Gabbi, J., Huss, M., Bauder, A., Cao, F., Schwikowski, M.: The impact of Saharan dust and black carbon on albedo and long-term mass balance of an Alpine glacier, The Cryosphere 9, 1385–1400, doi:10.5194/tcd-9-1133-2015, 2015.

Gallet JC., Domine F, Zender CS, and Picard G.: Measurement of the specific surface area of snow using infrared reflectance in an integrating sphere at 1310 and 1550 nm. Cryosphere, 3, 167-182, doi: 10.5194/tc-3-167-2009 2009.

Gardner, A., and Sharp, M.: A review of snow and ice albedo and the development of a new physically based broadband albedo parameterization, Journal of Geophysical Research: Earth Surface, 115, 10.1029/2009JF001444, 2010.

Ginot, P., Dumont, M., Lim, S., Patris, N., Taupin, J.-D., Wagnon, P., Gilbert, A., Arnaud, Y., Marinoni, A., Bonasoni, P., and Laj, P.: A 10 year record of black carbon and dust from a Mera Peak ice core (Nepal): variability and potential impact on melting of Himalayan glaciers, The Cryosphere, 8, 1479–1496, doi:10.5194/tc-8-1479-2014, 2014.

Goelles, T., and Bøggild, C.: Albedo reduction of ice caused by dust and black carbon accumulation: a model applied to the K-transect, West Greenland, Journal of Glaciology, 63, 1061–1076, doi:10.1017/jog.2017.74, 2017.

Grenfell, T. C., Warren, S. G., and Mullen, P. C.: Reflection of solar radiation by the Antarctic snow surface at ultraviolet, visible and near-infrared wavelengths, Journal of Geophysical Research, 99, 18669–18684, doi:10.1029/94JD01484, 1994.

Guo, W., Liu, S., Xu, J., Wu, L., Shangguan, D., Yao, X., Wei, J., Bao, W., Yu, P., Liu, Q., and Jiang, Z.: The second Chinese glacier inventory: data, methods and results, Journal of Glaciology, 61, 357–372, 10.3189/2015jog14j209, 2015.

Hadley, O. L., and Kirchstetter, T. W.: Black-carbon reduction of snow albedo, Nature Climate Change, 2(6), 437–440, doi:10.1038/ nclimate1433, 2012.

Hock, R., and Holmgren, B.: A distributed surface energy-balance model for complex topography and its application to Storglaciären, Sweden, Journal of Glaciology, 51, 25–36, doi:10.3189/172756505781829566, 2005.

Hock, R.: Glacier melt: a review of processes and their modeling, Progress in Physical Geography: Earth and Environment, 29, 362–391, doi:10.1191/0309133305pp453ra, 2005.

Hodson, A., Anesio, A. M., Tranter, M., Fountain, A., Osborn, M., Priscu, J., Laybourn-Parry, J., and Sattler, B.: Glacial Ecosystems, Ecological Monographs, 78,  doi: 10.1890/07-0187.1, 2008.

Hudson, S. R., and Warren, S. G.: An explanation for the effect of clouds over snow on the top-of-atmosphere bidirectional reflectance, Journal of Geophysical Research, 112, D19202, doi: 10.1029/2007JD008541, 2007.

[revised manuscript text omitted]

22118, doi:10.1073/pnas.0910444106, 2009.

1055  Xu, C., Li, Z., Li, H., Wang, F., and Zhou, P.: Long-range terrestrial laser scanning measurements of annual and intra-annual

mass balance for Urumqi Glacier No. 1, eastern Tien Shan, China, Cryosphere, 13, 2361-2383, doi: 10.5194/tc-13-2361-2019,

2019.

Yang, W., Guo, X., Yao, T., Yang, K., Zhao, L., Li, S., and Zhu, M.: Summertime surface energy budget and ablation modeling
in the ablation zone of a maritime Tibetan glacier, Journal of Geophysical Research, 116, D14, doi:10.1029/2010JD015183,
1060  2011.
Zhang, G., Kang, S., Fujita, K., Huintjes, E., Xu, J., Yamazaki, T., Haginova, S., Yang, W., Schere, D., Schneider, C., and
Yao, Y.: Energy and mass balance of Zhadang glacier surface, central Tibetan Plateau, Journal of Glaciology, 59, 137–148,
doi.org/10.3189/2013JoG12J152 , 2013.
Zhuravleva, T. B., and Kokhanovsky, A. A.: Influence of surface roughness on the reflective properties of snow, Journal of
1065  Quantitative Spectroscopy and Radiative Transfer, 112, 1353–1368, doi:10.1016/j.jqsrt.2011.01.004, 2011.

[Figure]

Figure S1. LAI samples over different snow and ice surface roughness types. From left to right, and from top to bottom, the snow and ice surface types are clean snow surface, snow surface rich in LAIs, cryoconite holes, smooth bare ice surface, protrusions and a very rough ice surface at both macro and micro scales.

[Figure]

Figure S2. (a) Manual photogrammetry-estimated snow surface roughness versus albedo in the 2018 melt season; (b) automatic photogrammetry-estimated snow surface roughness versus albedo in the 2018 melt season.

1075